# Learning the Infinitesimal Generator of Stochastic Diffusion Processes

**Vladimir R. Kostic**
CSML, Istituto Italiano di Tecnologia
University of Novi Sad
vladimir.kostic@iit.it

**Karim Lounici**
CMAP-Ecole Polytechnique
karim.lounici@polytechnique.edu

**Hélène Halconruy**
SAMOVAR, Télécom Sud-Paris
MODAL'X, Université Paris Nanterre
helene.halconruy@telecom-sudparis.eu

**Timothée Devergne**
CSML & ATSIM, Istituto Italiano di Tecnologia
timothee.devergne@iit.it

**Massimiliano Pontil**
CSML, Istituto Italiano di Tecnologia
AI Centre, University College London
massimiliano.pontil@iit.it

## Abstract

We address data-driven learning of the infinitesimal generator of stochastic diffusion processes, essential for understanding numerical simulations of natural and physical systems. The unbounded nature of the generator poses significant challenges, rendering conventional analysis techniques for Hilbert-Schmidt operators ineffective. To overcome this, we introduce a novel framework based on the energy functional for these stochastic processes. Our approach integrates physical priors through an energy-based risk metric in both full and partial knowledge settings. We evaluate the statistical performance of a reduced-rank estimator in reproducing kernel Hilbert spaces (RKHS) in the partial knowledge setting. Notably, our approach provides learning bounds independent of the state space dimension and ensures non-spurious spectral estimation. Additionally, we elucidate how the distortion between the intrinsic energy-induced metric of the stochastic diffusion and the RKHS metric used for generator estimation impacts the spectral learning bounds.

## 1 Introduction

Continuous-time processes are often modeled using ordinary differential equations (ODEs), assuming deterministic dynamics. However, real systems in science and engineering that are modeled by ODEs are subject to unfeasible-to-model influences, necessitating the extension of deterministic models through *stochastic differential equations* (SDEs), see [40, 42] and references therein. SDEs are advantageous for modeling inherently random phenomena. For instance, in finance, they specify the stochastic process governing asset behavior, a crucial step in constructing pricing models for financial derivatives [41]. Another compelling application arises in atomistic simulations, where SDEs are used to model the evolution of atomic systems subjected to thermal fluctuations through the Boltzmann distribution [37].

A diverse range of SDEs can be represented as $dX_t = a(X_t)dt + b(X_t)dW_t$, where $X_0 = x$. Here, $W$ denotes a (possibly multi-dimensional) Brownian motion, and the functions $a$ and $b$ are commonly

38th Conference on Neural Information Processing Systems (NeurIPS 2024).

| Aspect | [1] | [19] | [43] | Our work |
|---|---|---|---|---|
| Covers many SDEs | ✗ (only Laplacian) | ✓ | ✗ (only Langevin) | ✓ |
| Risk metric | $\mathcal{L}_\pi^2(\mathcal{X})$ metric | $\mathcal{L}_\pi^2(\mathcal{X})$ metric | $\mathcal{L}_\pi^2(\mathcal{X})$ metric | energy (8) |
| Physics-informed method | ✗ | ✓(full info. needed) | ✗ | ✓ (partial info. needed) |
| Avoids spurious eigenvalues | ✗ | ✗ | ✗ | ✓ |
| IG error bound | $\mathcal{O}(n^{-\frac{d}{2(d+1)}})$ | $\mathrm{Var} = \mathcal{O}(\frac{d^2}{\gamma^2\sqrt{n}})$ | $\mathcal{O}(n^{-\frac{1}{4}})$ | $\mathcal{O}(n^{-\frac{\alpha}{2(\alpha+\beta)}}), \alpha \geq \tau$ $\mathcal{O}(n^{-\frac{\alpha}{2(\beta+\tau)}}), \alpha < \tau$ |
| Spectral rates | ✗ | ✗ | ✗ | ✓ |
| Time complexity | $\mathcal{O}(n^2 + n^{3/2}d)$ | $\mathcal{O}(n^3 d^3)$ | $\mathcal{O}(n^3 d^3)$ | $\mathcal{O}(r\, n^2 d^2)$ |

Table 1: Comparison to previous kernel-based works on generator learning. Sample size is $n$, state-space dimension is $d$, $\gamma$ is the regularization parameter of KKR and RRR and $r$ is RRR rank parameter. Our learning bounds are derived in Theorem 2 where the parameters $\alpha, \beta, \tau$ quantify the intrinsic difficulty of the problem and impact of kernel choice on learning IG.

known as the *drift* and *diffusion* coefficients, respectively. Determining these coefficients from one or more trajectories, whether discretized or continuous, has been a key pursuit in "diffusion statistics" since the 1980s, as seen in works like [22, 31, 30]. However, uncovering the drift and diffusion coefficients alone does not reveal all the intrinsic properties of a complex system, such as the metastable states of Langevin dynamics in atomistic simulations [see e.g. 49, and references therein]. Consequently, there has been a shift and growing interest in the Infinitesimal Generator (IG) of an SDE, as its spectral decomposition offers a more in-depth understanding of system dynamics and behavior, thus providing a comprehensive picture beyond the mere identification of coefficients.

Recovering the spectral decomposition of the IG can theoretically be achieved by exploiting the well-studied Transfer Operators (TO) [see 27, and references therein]. TOs represent the average evolution of state functions (observables) over time and, being linear and amenable to spectral decomposition under certain conditions, they offer a valuable means of interpreting and analyze nonlinear systems. However, they require evenly sampled data at a high rate, which may be impractical. Additionally, TO approaches are purely data-driven, complicating the incorporation of partial or full knowledge of an SDE into the learning process. Thus, there is growing interest in learning the IG directly from data, as it can handle uneven sampling and integrate SDE knowledge. The challenge lies in the IG being an unbounded operator, unlike TO which are often well-approximated by Hilbert-Schmidt operators with comprehensive statistical theory [28]. Unfortunately, the existing statistical theory collapses when applied to unbounded operators, prompting the need to completely rethink the problem.

**Related work**  Extensive research has explored learning dynamical systems from data [see the monographs 7, 32, and references therein]. Analytical models for dynamics are often unavailable, motivating the need for data-driven methods. Two prominent approaches have emerged: deep learning-based methods [5, 12, 36], effective for learning complex data representations but lacking statistical analysis, and kernel methods [2, 6, 10, 26, 21, 24, 27, 28, 50], offering solid statistical guarantees for the estimation of the TO but requiring careful selection of the kernel function. A related question, tackling the challenging problem of learning invariant subspaces of the TO, has recently led to the development of several methodologies [21, 34, 38, 46, 52, 51], some of which are based on deep canonical correlation analysis [3, 29]. In comparison, there have been significantly fewer works on learning the IG and only in very specific settings. In [55] a deep learning approach is developed for Langevin diffusion, while [23] presents an extended dynamical mode decomposition method for learning the generator and clarifies its connection to Galerkin's approximation. However, neither of these two works provides any learning guarantees. In this respect the only previous work we are aware of are [1], revising the Galerkin method for Laplacian diffusion, [19], presenting a kernel approach for general diffusion SDE with full knowledge of drift and diffusion coefficients, and [43] addressing Langevin diffusion. As highlighted in Table 1, the bounds and analysis in these works are either restricted to specific SDEs or are incomplete. Notably, none of these works proposed an adequate framework to handle the unboundedness of the IG, resulting in an incomplete theoretical analysis and suboptimal rates, sometimes explicitly depending on state space dimension. Moreover, the estimators proposed in these works are prone to spurious eigenvalues and do not offer guarantees on the accurate estimation of eigenvalues and eigenfunctions.

**Contributions**  In summary, our main contributions are: **1)** Proposing a fundamentally new idea to estimate the spectrum of self-adjoint generators of stable Itô SDE from a single trajectory. In contrast

to all existing works, we exploit the geometry of the process via a novel energy (risk) functional; **2)** In a certain sense we "*fight fire (resolvent) with fire (generator)*" to derive a new efficient learning method that is able to infer the best approximation of the resolvent of IG on the RKHS, independently of the time-sampling; **3)** We prove the first IG spectral estimation finite sample bounds using (imperfect) partial knowledge, which notably, overcome the curse of dimensionality present in classical numerical methods; **4)** Each important aspect of our learning method, especially in relation to the most relevant existing works, is empirically demonstrated to complement our theoretical analysis.

## 2 Background and the problem

Our drive to estimate the eigenvalues of the infinitesimal generator for an SDE like (1) stems from its crucial role in characterizing dynamics in physical systems. This operator's closed form (3), relies on the drift $a$ and diffusion coefficient $b$, where we have partial knowledge: $b$ is assumed to be known, but $a$ is not. To compensate for the lack of prior knowledge about $a$, we introduce the system's *energy* as an additional known quantity. Below, we detail the mathematical concepts framing the problem (generator, spectrum, energy) exemplified through the Langevin and Cox-Ingersoll-Ross processes. For detailed list of notation used throughout the paper we refer Table 7 in the appendix.

**Transfer operator and infinitesimal generator** A variety of physical, biological, and financial systems evolve through stochastic processes $X = (X_t)_{t \in \mathbb{R}^+}$, where $X_t \in \mathcal{X} \subset \mathbb{R}^d$ denotes the system's state at time $t$. A commonly employed model for such dynamics is captured by *stochastic differential equations* (SDEs) of the form

$$dX_t = a(X_t)dt + b(X_t)dW_t \quad \text{and} \quad X_0 = x, \tag{1}$$

where $x \in \mathcal{X}$, $W = (W_t^1, \ldots, W_t^p)_{t \in \mathbb{R}_+}$ is a $\mathbb{R}^p$-dimensional ($p \in \mathbb{N}$) standard Brownian motion, the *drift* $a : \mathcal{X} \to \mathbb{R}^d$, and the *diffusion* $b : \mathcal{X} \to \mathbb{R}^{d \times p}$ are assumed to be globally Lipschitz and sub-linear, so that the SDE (1) admits an unique solution $X = (X_t)_{\geqslant 0}$ with values in $(\mathcal{X}, \mathcal{B}(\mathcal{X}))$. Processes akin to equations like (1) are diverse, spanning models like Langevin and Cox-Ingersoll-Ross processes (see examples below), with broad applications in science and engineering. The process $X$ is a continuous-time Markov process with almost surely continuous sample paths whose dynamics is described by a family of probability densities $(p_t)_{t \in \mathbb{R}_+}$ and *transfer operators* $(A_t)_{t \in \mathbb{R}_+}$ such that for all $t \in \mathbb{R}_+$, $E \in \mathcal{B}(\mathcal{X})$, $x \in \mathcal{X}$ and measurable function $f : \mathcal{X} \to \mathbb{R}$,

$$\mathbb{P}(X_t \in E | X_0 = x) = \int_E p_t(x, y)dy \text{ and } A_t f = \int_{\mathcal{X}} f(y)p_t(\cdot, y)dy = \mathbb{E}\big[f(X_t) \,|\, X_0 = \cdot\big]. \tag{2}$$

Evaluating $A_t f$ at $x$ yields the expectation of $f$ starting from $x$ and evolving until time $t$, making the transfer operator crucial for understanding $X$ dynamics. The family $(A_t)_{t \in \mathbb{R}_+}$ satisfies the fundamental semigroup equation $A_{t+s} = A_t \circ A_s$, for $s, t \in \mathbb{R}_+$. Here, we focus on the transfer operator's effect on the set $\mathcal{L}_\pi^2(\mathcal{X})$, a choice driven by the existence of an invariant measure $\pi$ for $A_t$ on $(\mathcal{X}, \mathcal{B}(\mathcal{X}))$ which satisfies $A_t^* \pi = \pi$ for all $t \in \mathbb{R}_+$. Then, the process $X$ is characterized by the infinitesimal generator $L$ defined for every $f \in \mathcal{L}_\pi^2(\mathcal{X})$ such that the limit $Lf = \lim_{t \to 0^+} (A_t f - f)/t$ exists in $\mathcal{L}_\pi^2(\mathcal{X})$. The operator $L$ is closed on its domain $\mathrm{dom}(L)$ which is equal to the Sobolev space

$$\mathcal{W}_\pi^{1,2}(\mathcal{X}) = \{f \in \mathcal{L}_\pi^2(\mathcal{X}) \mid \|f\|_{\mathcal{W}}^2 = \|f\|_{\mathcal{L}_\pi^2} + \|\nabla f\|_{\mathcal{L}_\pi^2} < \infty\}.$$

For SDE dynamics of the form (1), we can prove (see A.2 for details) that $L$ is the second-order differential operator given, for any $f \in \mathcal{L}_\pi^2(\mathcal{X})$, $x \in \mathcal{X}$, by

$$Lf(x) = \nabla f(x)^\top a(x) + \frac{1}{2}\mathrm{Tr}\big[b(x)^\top (\nabla^2 f(x))b(x)\big], \tag{3}$$

where $\nabla^2 f = (\partial_{ij}^2 f)_{i \in [d], j \in [p]}$ denotes the Hessian matrix of $f$.

**Spectral decomposition** Knowing only the drift $a$ and diffusion $b$ is not enough to compute (2) or to understand quantitative aspects of dynamical phase transitions, such as time scales and metastable states. The eigenvalues and eigenvectors of the generator are crucial for capturing these effects. To address the possible unbounded nature of $L$, one can turn to an auxiliary operator, the resolvent, which, under certain conditions, shares the same eigenfunctions as $L$ and becomes compact. When it exists and is continuous for some $\mu \in \mathbb{C}$, the operator $L_\mu = (\mu I - L)^{-1}$ is the *resolvent* of $L$ and the corresponding *resolvent set* is defined by

$$\rho(L) = \big\{\mu \in \mathbb{C} \mid (\mu I - L) \text{ is bijective and } L_\mu \text{ is continuous}\big\}.$$

We assume that $L$ has a *compact resolvent*, meaning $\rho(L) \neq \emptyset$ and there exists $\mu_0 \in \rho(L)$ such that $(\mu_0 I - L)^{-1}$ is compact. Under this assumption, and given that $(L, \mathrm{dom}(L))$ is self-adjoint, we can prove the spectral decomposition of the generator (see A.1 for details) as follows:

$$L = \sum_{i \in \mathbb{N}} \lambda_i \, f_i \otimes f_i, \tag{4}$$

where $(\lambda_i)_{i \in \mathbb{N}}$ are the eigenvalues of $L$, and the corresponding eigenfunctions $f_i \in \mathcal{L}^2_\pi(\mathcal{X})$, forming an orthonormal basis $(f_i)_{i \in \mathbb{N}}$, are also eigenfunctions of the transfer operator $A_t$.

**Dirichlet forms and energy** To handle the initial lack of knowledge about the drift $a$, we assume to have access to another quantity, called the *energy*, defined as $\mathfrak{E}(f) = \lim_{t \to 0} \int_{\mathcal{X}} (f(f - A_t f))/t d\pi$ for all functions $f \in \mathcal{L}^2_\pi(\mathcal{X})$ for which this limit exists, defining in the way the domain $\mathrm{dom}(\mathfrak{E})$. The associated *Dirichlet form* is the bilinear form defined by polarization for any $f, g \in \mathrm{dom}(\mathfrak{E})$ by

$$\mathfrak{E}(f, g) = -\int_{\mathcal{X}} f(Lg)d\pi = \int_{\mathcal{X}} (-Lf)g d\pi. \tag{5}$$

For every $f \in \mathrm{dom}(\mathfrak{E})$, we have $\mathfrak{E}(f) = \mathfrak{E}(f, f)$. As for every $i \in \mathbb{N}$, $0 \leq \mathfrak{E}(f_i, f_i) = -\int_{\mathcal{X}} f_i(\lambda_i f_i)d\pi = -\lambda_i$, we check that the eigenvalues of $L$ are negative. To relate $L$ to Dirichlet form, we assume there exists a *Dirichlet operator* $B = s^\top \nabla$ where $s = [s_1 \,|\, \dots \,|\, s_p] : x \in \mathcal{X} \mapsto s(x) = [s_1(x) \,|\, \dots \,|\, s_p(x)] \in \mathbb{R}^{d \times p}$ is a smooth function s.t. $Lf = s(s^\top \nabla f) = s(Bf)$ and so that

$$\int_{\mathcal{X}} (-Lf)g d\pi = \int_{\mathcal{X}} (s(x)s(x)^\top \nabla f(x))^\top \nabla g(x) \pi(dx) = \int_{\mathcal{X}} (Bf(x))^\top (Bg(x)) \pi(dx).$$

We get that for any $f \in \mathrm{dom}(\mathfrak{E})$

$$\mathfrak{E}(f) = \int_{\mathcal{X}} \|s^\top \nabla f\|^2 d\pi = \mathbb{E}_{x \sim \pi}[\|Bf(x)\|^2], \tag{6}$$

which is reminiscent of the expected value of the *kinetic energy* in quantum mechanics [17].

In the following, while we discuss in detail the examples of Overdamped Langevin and the Cox-Ingersoll-Ross processes, we briefly mention other ones that have a Dirichlet form: the Wright-Fisher diffusion (in dimension one), which can be defined in the context of population genetics and can be adapted to model interest rates, see [16], the geometric Brownian motion which models the price process of a financial asset, the multi-dimensional Brownian motion ($a=0$) that corresponds to the heat equation, the transport processes associated with advection-diffusion equation, see [33], and the process associated with Poisson's equation in electrostatics, see [25].

**Example 1** (Langevin). *Let $k_b, T \in \mathbb{R}^*_+$. The* overdamped Langevin equation *driven by a* potential $V : \mathbb{R}^d \to \mathbb{R}$ *is given by* $dX_t = -\nabla V(X_t)dt + \sqrt{2(k_b T)}dW_t$ *and* $X_0 = x$, *where $k_b$ and $T$ respectively represent the coefficient of friction and the temperature of the system. Its infinitesimal generator $L$ is defined by $Lf = -\nabla V^\top \nabla f + (k_b T)\Delta f$, for $f \in \mathcal{W}^{1,2}_\pi(\mathcal{X})$. Since $\int (-Lf)g \, d\pi = -\int \left[ \nabla \left( (k_b T)\nabla f(x) \frac{e^{-(k_b T)^{-1}V(x)}}{Z} \right) \right] g(x)dx = (k_b T) \int \nabla f^\top \nabla g \, d\pi = \int f(-Lg) \, d\pi$, generator $L$ is self-adjoint and associated to a gradient Dirichlet form with $s(x) = (k_b T)^{1/2}(\delta_{ij})_{i \in [d], j \in [p]}$.*

**Example 2** (Cox-Ingersoll-Ross process). *Let $d = 1$, $a, b \in \mathbb{R}$, $\sigma \in \mathbb{R}^*_+$. The* Cox-Ingersoll-Ross *process is solution of the SDE $dX_t = (a + bX_t)dt + \sigma\sqrt{X_t}dW_t$ and $X_0 = x$. Its infinitesimal generator $L$ is defined for $f \in \mathcal{L}^2_\pi(\mathcal{X})$ by $Lf = (a + bx)\nabla f + \frac{\sigma^2 x}{2}\Delta f$. By integration by parts, we can check that the generator $L$ satisfies $\int (-Lf)g \, d\pi = \int \frac{\sigma^2 x}{2} f'(x)g'(x) \pi(dx) = \int f(-Lg) \, d\pi$, and it is associated to a gradient Dirichlet form with $s(x) = \sigma\sqrt{x}/\sqrt{2}$.*

**Learning in reproducing kernel Hilbert spaces (RKHSs)** Throughout the paper we let $\mathcal{H}$ be an RKHS and let $k : \mathcal{X} \times \mathcal{X} \to \mathbb{R}$ be the associated kernel function. We let $\phi : \mathcal{X} \to \mathcal{H}$ be a *feature map* [45] such that $k(x, x') = \langle \phi(x), \phi(x') \rangle$ for all $x, x' \in \mathcal{X}$. We consider RKHSs satisfying $\mathcal{H} \subset \mathcal{L}^2_\pi(\mathcal{X})$ [45, Chapter 4.3], so that one can approximate $L : \mathcal{L}^2_\pi(\mathcal{X}) \to \mathcal{L}^2_\pi(\mathcal{X})$ with an operator $G : \mathcal{H} \to \mathcal{H}$. Notice that despite $\mathcal{H} \subset \mathcal{L}^2_\pi(\mathcal{X})$, the two spaces have different metric structures, that is for all $f, g \in \mathcal{H}$, one in general has $\langle f, g \rangle_\mathcal{H} \neq \langle f, g \rangle_{\mathcal{L}^2_\pi}$. In order to handle this ambiguity, we introduce the *injection operator* $S_\pi : \mathcal{H} \to \mathcal{L}^2_\pi(\mathcal{X})$ such that for all $f \in \mathcal{H}$, the object $S_\pi f$ is the element of $\mathcal{L}^2_\pi(\mathcal{X})$ which is pointwise equal to $f \in \mathcal{H}$, but endowed with the appropriate $\mathcal{L}^2_\pi$ norm.

Then, the infinitesimal generator restricted to $\mathcal{H}$ is simply $LS_\pi$ which can be estimated by $S_\pi G$ for some $G \in \mathrm{HS}(\mathcal{H})$. This approach is based on the embedding $\ell\phi\colon \mathcal{X} \to \mathcal{H}$ of the generator in the RKHS that can be defined for kernels $k \in \mathcal{C}^2(\mathcal{X} \times \mathcal{X})$ whenever one knows drift and diffusion coefficients, see App. B, so that the reproducing property $\langle \ell\phi(x), h\rangle_{\mathcal{H}} = [LS_\pi h](x)$ holds true. Based on this observation, [19] developed empirical estimators of $LS_\pi$ that essentially minimize the risk $\|LS_\pi - S_\pi G\|^2_{\mathrm{HS}(\mathcal{H}, \mathcal{L}^2_\pi)} = \mathbb{E}_{x\sim\pi}\|\ell\phi(x) - G^*\phi(x)\|^2_{\mathcal{H}}$. In scenarios where drift and diffusion coefficients are not known, then $\ell\phi$ becomes non-computable. However if the process has the Dirichlet form (6), one can still empirically estimate the Galerkin projection $(S^*_\pi S_\pi)^\dagger S^*_\pi LS_\pi$ onto $\mathcal{H}$, as considered in [1], which in fact minimizes the same risk. Yet this approach is problematic due to the unbounded nature of the generator and the associated estimators typically suffer from a large number of spurious eigenvalues around zero, making the estimation of physically most relevant eigenfunctions unreliable even in the self-adjoint case. Conversely, classical numerical methods can compute the leading part of a spectrum without spuriousness issues, suggesting that data-driven approaches should achieve similar reliability. In this paper, we address this problem by designing a novel notion of risk, leading to principled estimators designed to surmount these challenges.

## 3   Novel statistical learning framework

In this section, we tackle the challenges in developing suitable generator estimators highlighted earlier. To this end, we introduce a risk metric for resolvent estimation that can be efficiently minimized empirically, leading to good spectral estimation. Since $L$ and $(\mu I - L)^{-1}$ share the same eigenfunctions, the main idea is to learn the (compact) resolvent, which can be effectively approximated by finite-rank operators, instead of learning the generator directly. However, this approach is challenging due to the lack of closed analytical forms for the action of the resolvent.

First, given $\mu > 0$, in order to approximate the action of the resolvent on the RKHS by some operator $G\colon \mathcal{H} \to \mathcal{H}$, we introduce its embedding $\chi_\mu\colon \mathcal{X} \to \mathcal{H}$ via the reproducing property $\langle \chi_\mu(x), h\rangle_{\mathcal{H}} = [(\mu I - L)^{-1}S_\pi h](x)$, formally given by $\chi_\mu(x) = \int_0^\infty \mathbb{E}[\phi(X_t)e^{-\mu t}\,|\,X_0 = x]dt$, see App. B for details. Using this notation, we aim to estimate $[(\mu I - L)^{-1}S_\pi h](x) \approx [Gh](x)$, $h \in \mathcal{H}$, i.e. the objective is to estimate $\chi_\mu(x) \approx G^*\phi(x)$ $\pi$-a.e.

An obvious metric for the risk would be the mean square error (MSE) w.r.t. distribution $\pi$ of the data. However, this becomes intractable since in general $\chi_\mu$ is not computable in closed form when either full or partial knowledge of the process is at hand. To mitigate this issue, we introduce a different ambient space in which we study the resolvent,

$$\mathcal{W}^\mu_\pi(\mathcal{X}) = \{f \in \mathrm{dom}(L) \mid \|f\|^2_{\mathcal{W}^\mu_\pi(\mathcal{X})} \equiv \mathfrak{E}_\mu[f] = \langle f, (\mu I - L)f\rangle_{\mathcal{L}^2_\pi} < \infty\},$$

where the norm now balances the energy of an observable $f\colon \mathcal{X} \to \mathbb{R}$ w.r.t. the invariant distribution $\|f\|^2_{\mathcal{L}^2_\pi}$ and its energy w.r.t. the transient dynamics $-\langle f, Lf\rangle_{\mathcal{L}^2_\pi}$. Indeed

$$\mathfrak{E}_\mu[f] = \mathbb{E}_{x\sim\pi}[\mu|f(x)|^2 - f(x)[Lf](x)] = \mathbb{E}_{x\sim\pi}[\mu\,|f(x)|^2 + \|s(x)^\top \nabla f(x)\|^2], \qquad (7)$$

where the last equality holds for Dirichlet gradient form (6), in which case $\mathcal{W}^\mu_\pi(\mathcal{X})$ is simply a weighted Sobolev space. Importantly, this energy functional can be empirically estimated from data sampled from $\pi$, whenever full knowledge, that is drift and diffusion coefficients of the SDE (1), or partial knowledge, i.e. the diffusion coefficient and Dirichlet operator $B$ in (6), is at hand. With that in mind, instead of the standard MSE, we introduce the energy-based risk functional as

$$\min_{G\colon \mathcal{H}\to\mathcal{H}} \mathcal{R}(G) = \mathfrak{E}_\mu[\|\chi_\mu(\cdot) - G^*\phi(\cdot)\|_{\mathcal{H}}]. \qquad (8)$$

Denoting by $Z_\mu\colon \mathcal{H} \to \mathcal{W}^\mu_\pi(\mathcal{X})$ the canonical injection, (8) can be equivalently written as

$$\mathcal{R}(G) = \|(\mu I - L)^{-1}Z_\mu - Z_\mu G\|^2_{\mathrm{HS}(\mathcal{H},\mathcal{W})} = \|(\mu I - L)^{-1/2}S_\pi - (\mu I - L)^{1/2}S_\pi G\|^2_{\mathrm{HS}(\mathcal{H},\mathcal{L}^2_\pi)}, \quad (9)$$

where we abbreviated $\mathcal{W} = \mathcal{W}^\mu_\pi(\mathcal{X})$ and used $Z^*_\mu = S^*_\pi(\mu I - L)$, recalling that Hilbert-Schmidt and spectral norms for operators $A\colon \mathcal{H} \to \mathcal{W}$, are $\|A\|_{\mathrm{HS}(\mathcal{H},\mathcal{W})} = \sqrt{\mathrm{tr}(A^*(\mu I - L)A)}$ and $\|A\|_{\mathcal{H}\to\mathcal{W}} = \sqrt{\lambda_1(A^*(\mu I - L)A)} \geq \mu^{-1/2}\|A\|_{\mathcal{H}\to\mathcal{L}^2_\pi}$.

Therefore, (9) implies that the regularized energy norm, while dominating the classical $\mathcal{L}^2_\pi$ norm, exerts a *balancing effect* on the estimation of the resolvent. This leads us to the first general result regarding the well-posedness of this framework.

**Proposition 1.** *Given $\mu > 0$, let $\mathcal{H} \subseteq \mathcal{W}_\pi^\mu(\mathcal{X})$ be the RKHS associated to kernel $k \in \mathcal{C}^2(\mathcal{X} \times \mathcal{X})$ such that $Z_\mu \in \mathrm{HS}\,(\mathcal{H}, \mathcal{W}_\pi^\mu(\mathcal{X}))$, and let $P_\mathcal{H}$ be the orthogonal projector onto the closure of $\mathrm{Im}(Z_\mu) \subseteq \mathcal{W}_\pi^\mu(\mathcal{X})$. Then, for every $\varepsilon > 0$, there exists a finite rank operator $G \colon \mathcal{H} \to \mathcal{H}$ such that $\mathcal{R}(G) \leq \|(I - P_\mathcal{H})(\mu I - L)^{-1} Z_\mu\|_{\mathrm{HS}(\mathcal{H}, \mathcal{W})}^2 + \varepsilon$. Consequently, when $k$ is universal, $\mathcal{R}(G) \leq \varepsilon$.*

The previous proposition reveals that whenever the hypothetical domain is dense in the true domain and the injection operator is Hilbert-Schmidt, there is no irreducible risk and one can find arbitrarily good finite rank approximations of the generator's resolvent. Note that $Z_\mu \in \mathrm{HS}\,(\mathcal{H}, \mathcal{W}_\pi^\mu(\mathcal{X}))$ is equivalent to $Z_\mu^* Z_\mu = S_\pi^*(\mu I - L) S_\pi$ being a trace class operator, which is assured for our Examples 1 and 2, see the discussion in App. E.

Now, to address how minimization of the risk impacts the estimation of the spectral decomposition, let us define the *operator norm error* and the *metric distortion* functional, respectively, as

$$\mathcal{E}(G) = \|(\mu I - L)^{-1} Z_\mu - Z_\mu G\|_{\mathcal{H} \to \mathcal{W}}, \; G \in \mathrm{HS}\,(\mathcal{H}), \quad \text{and} \quad \eta(h) = \|h\|_\mathcal{H} / \|h\|_\mathcal{W}, \; h \in \mathcal{H}. \quad (10)$$

**Proposition 2.** *Let $\widehat{G} = \sum_{i \in [r]} (\mu - \widehat{\lambda}_i)^{-1} \widehat{h}_i \otimes \widehat{g}_i$ be the spectral decomposition of $\widehat{G} \colon \mathcal{H} \to \mathcal{H}$, where $\widehat{\lambda}_i \geq \widehat{\lambda}_{i+1}$ and let $\widehat{f}_i = S_\pi \widehat{h}_i / \|S_\pi \widehat{h}_i\|_{\mathcal{L}_\pi^2}$, for $i \in [r]$. Then for every $\mu > 0$ and $i \in [r]$*

$$\frac{|\lambda_i - \widehat{\lambda}_i|}{|\mu - \lambda_i||\mu - \widehat{\lambda}_i|} \leq \mathcal{E}(\widehat{G})\eta(\widehat{h}_i) \quad \text{and} \quad \|\widehat{f}_i - f_i\|_{\mathcal{L}_\pi^2}^2 \leq \frac{2\,\mathcal{E}(\widehat{G})\eta(\widehat{h}_i)}{\mu\,[\mathrm{gap}_i - \mathcal{E}(\widehat{G})\eta(\widehat{h}_i)]_+}, \quad (11)$$

*where $\mathrm{gap}_i$ is the difference between $i$-th and $(i+1)$-th eigenvalue of $(\mu I - L)^{-1}$.*

Note that the estimation of the eigenfunctions is first obtained in the norm with respect to the energy space, and then transformed to the $\mathcal{L}_\pi^2$-norm, as $\|f\|_\mathcal{W} \geq \sqrt{\mu}\|f\|_{\mathcal{L}_\pi^2}$, $f \in W_{\mu,\gamma}$. Therefore, by controlling the operator norm error and metric distortion (see App. C), we can guarantee accurate spectral estimation. Consequently, this allows us to approximately solve the SDE (1) starting from an initial condition

$$\mathbb{E}[h(X_t) \mid X_0 = x] = [e^{Lt} S_\pi h](x) \approx \sum_{i \in [r]} e^{\widehat{\lambda}_i t} \langle \widehat{g}_i, h \rangle_\mathcal{H} \widehat{h}_i(x). \quad (12)$$

## 4 Empirical risk minimization

In this section we address empirical risk minimization (ERM), deriving two main estimators. The first one minimizes empirical risk with Tikhonov regularization, while the second introduces additional regularization in the form of rank constraints.

To present the estimator, we denote the covariance operator w.r.t. $\mathcal{L}_\pi^2$ by $C = S_\pi^* S_\pi = \mathbb{E}_{x \sim \pi}[\phi(x) \otimes \phi(x)]$, the cross-covariance operator $T = S_\pi^* L S_\pi$, capturing correlations between input and the outputs of the generator, and the covariance operator $W_\mu = Z_\mu^* Z_\mu = S_\pi^*(\mu I - L) S_\pi$ w.r.t. energy space $\mathcal{W}$. All operators can be estimated from data, depending on the available prior knowledge. In this work we focus on the case when Dirichlet gradient form is known, i.e. we can define the embedding of the Dirichlet operator $B = s^\top \nabla \colon \mathcal{L}_\pi^2(\mathcal{X}) \to [\mathcal{L}_\pi^2(\mathcal{X})]^p$ into RKHS $d\phi \colon \mathcal{X} \to \mathcal{H}^p$ via the reproducing property as $\langle d\phi(x), h \rangle_\mathcal{H} = [BS_\pi h](x) = s(x)^\top Dh(x) \in \mathbb{R}^p$, $h \in \mathcal{H}$. More precisely, we have that $d\phi(x)$ is a $p$-dimensional vector with components $d_k\phi(x)$, where $d_k\phi \colon \mathcal{X} \to \mathcal{H}$ is given via reproducing property $\langle d_k\phi(x), h \rangle_\mathcal{H} = s_k(x)^\top Dh(x)$, $k \in [p]$. Hence, in this case we have that

$$T = -\mathbb{E}_{x \sim \pi}[d\phi(x) \otimes d\phi(x)] = -\sum_{k \in [p]} \mathbb{E}_{x \sim \pi}[d_k\phi(x) \otimes d_k\phi(x)]. \quad (13)$$

Moreover, defining $w\phi \colon \mathcal{X} \to \mathcal{H}^{p+1}$ by $w\phi(x) = [\sqrt{\mu}\phi(x), d_1\phi(x), \dots d_p\phi(x)]^\top \in \mathbb{R}^{p+1}$, we get

$$W_\mu = \mathbb{E}_{x \sim \pi}[\mu\phi(x) \otimes \phi(x) + d\phi(x) \otimes d\phi(x)] = \mathbb{E}_{x \sim \pi}[w\phi(x) \otimes w\phi(x)]. \quad (14)$$

**Regularized risk** Let us first introduce the regularized risk defined for some $\gamma > 0$ by

$$\mathcal{R}_\gamma(G) = \|(\mu I - L)^{-1/2} S_\pi - (\mu I - L)^{1/2} S_\pi G\|_{\mathrm{HS}(\mathcal{H}, \mathcal{L}_\pi^2)}^2 + \gamma\mu\|G\|_{\mathrm{HS}(\mathcal{H})}^2, \quad G \in \mathrm{HS}\,(\mathcal{H}), \quad (15)$$

which, after some algebra, can be written as

$$\begin{aligned}
\mathcal{R}_\gamma(G) &= \mathrm{tr}\left[G^*(\mu C - T + \mu\gamma I)G - 2CG + S_\pi^*(\mu I - L)^{-1} S_\pi\right] \\
&= \underbrace{\|W_{\mu,\gamma}^{-1/2} C - W_{\mu,\gamma}^{1/2} G\|_{\mathrm{HS}(\mathcal{H})}^2}_{\text{variance term}} + \underbrace{\mathfrak{E}_\mu[\chi_\mu(\cdot)] - \|W_{\mu,\gamma}^{-1/2} C\|_{\mathrm{HS}(\mathcal{H})}^2}_{\text{bias term}}
\end{aligned}$$

where $W_{\mu,\gamma} = W_\mu + \mu\gamma I$ is the regularized covariance w.r.t. $\mathcal{W}$.

Hence, assuming the access to the dataset $\mathcal{D}_n = (x_i)_{i\in[n]}$ made of i.i.d. samples from the invariant distribution $\pi$, replacing the regularized energy $\mathfrak{E}_\mu$ with its empirical estimate $\widehat{\mathfrak{E}}_\mu$ leads to the regularized empirical risk functional expressed as

$$\widehat{\mathcal{R}}_\gamma(G) = \|\widehat{W}_{\mu,\gamma}^{-1/2}\widehat{C} - \widehat{W}_{\mu,\gamma}^{1/2}G\|_{\mathrm{HS}(\mathcal{H})}^2 + \widehat{\mathfrak{E}}_\mu[\chi_\mu(\cdot)] - \|\widehat{W}_{\mu,\gamma}^{-1/2}\widehat{C}\|_{\mathrm{HS}(\mathcal{H})}^2, \tag{16}$$

where $W_{\mu,\gamma}$ and $C$ are estimated by their empirical counterparts $W_{\mu,\gamma}$ and $C$, respectively, via (13).

Therefore, our regularized empirical risk minimization approach reduces to

$$\min_G \|\widehat{W}_{\mu,\gamma}^{-1/2}\widehat{C} - \widehat{W}_{\mu,\gamma}^{1/2}G\|_{\mathrm{HS}(\mathcal{H})}^2, \tag{17}$$

and we analyze two different estimators, the first one $\widehat{G}_{\mu,\gamma}$ is obtained by minimizing regularized empirical risk (17) over all $G \in \mathrm{HS}(\mathcal{H})$, and, hence, the name *Kernel Ridge Regression* (KRR) of the generators resolvent. The second one $\widehat{G}_{\mu,\gamma}^r$ minimizes (17) subject to the (hard) constraint that $G$ is at most of (a priori fixed) rank $r \in \mathbb{N}$ and, hence, is named *Reduced Rank Regression* (RRR) of the generator's resolvent. Notice that when $r = n$, the two estimators coincide. After some algebra, one sees that both minimization problems have closed form solutions

$$\widehat{G}_{\mu,\gamma} = \widehat{W}_{\mu,\gamma}^{-1}\widehat{C} \qquad \text{and} \qquad \widehat{G}_{\mu,\gamma}^r = \widehat{W}_{\mu,\gamma}^{-1/2}[\![\widehat{W}_{\mu,\gamma}^{-1/2}\widehat{C}]\!]_r, \tag{18}$$

where $[\![\cdot]\!]_r$ denotes the $r$-truncated SVD of a compact operator.

To conclude this section, we show how to compute the eigenvalue decomposition of (18). To this end, we define the sampling operators $\widehat{S}\colon \mathcal{H} \to \mathbb{R}^n$ and $\widehat{Z}_\mu\colon \mathcal{H} \to \mathbb{R}^{(1+p)n}$ by

$$(\widehat{S}h)_i = \tfrac{1}{\sqrt{n}}h(x_i), i\in[n], \quad \text{and} \quad (\widehat{Z}_\mu h)_{kn+i} = \begin{cases} \frac{\sqrt{\mu}}{\sqrt{n}}h(x_i), & k=0, i\in[n], \\ \frac{1}{\sqrt{n}}s_k(x_i)^\top Dh(x_i), & k\in[p], i\in[n]. \end{cases}$$

Further, let $\mathrm{K} = n^{-1}[k(x_i,x_j)]_{i,j\in[n]} \in \mathbb{R}^{n\times n}$ be kernel Gram matrix, and introduce the Gram matrices $\mathrm{N}\in\mathbb{R}^{n\times pn}$ and $\mathrm{M}\in\mathbb{R}^{pn\times pn}$ whose elements, for $k\in[1+p], i,j\in[n]$ are

$$\mathrm{N}_{i,(k-1)n+j} = n^{-1}\langle\phi(x_i), d_k\phi(x_j)\rangle_{\mathcal{H}} \text{ and } \mathrm{M}_{(k-1)n+i,(\ell-1)n+j} = n^{-1}\langle d_k\phi(x_i), d_\ell\phi(x_j)\rangle_{\mathcal{H}}. \tag{19}$$

We note that although we have introduced the above matrices via inner products in $\mathcal{H}$, they can be readily computed via the kernel and its gradients knowing the Dirichlet form, see D.

**Theorem 1.** *Given $\mu > 0$ and $\gamma > 0$, let $\mathrm{J}_{\mu,\gamma} = \mathrm{K} - \mathrm{N}(\mathrm{M}+\gamma\mu I)^{-1}\mathrm{N}^\top + \gamma I$. Let $(\widehat{\sigma}_i^2, v_i)_{i\in[r]}$ be the leading eigenpairs of the following generalized eigenvalue problem*

$$\mu^{-1}(\mathrm{J}_{\mu,\gamma} - \gamma I)\mathrm{K}v_i = \widehat{\sigma}_i^2\mathrm{J}_{\mu,\gamma}v_i, \quad v_i^\top \mathrm{K}v_j = \delta_{ij}, \ i,j\in[r]. \tag{20}$$

*Denoting $\mathrm{V}_r = [v_1|\ldots|v_r]\in\mathbb{R}^{n\times r}$ and $\Sigma_r = \mathrm{diag}(\widehat{\sigma}_1,\ldots,\widehat{\sigma}_r)$, if $(\nu_i, w_i^\ell, w_i^r)_{i\in[r]}$ are eigentriplets of matrix $\mathrm{V}_r^\top\mathrm{V}_r\Sigma_r^2 \in \mathbb{R}^{r\times r}$, then the eigenvalue decomposition the RRR estimator $\widehat{G}_{\mu,\gamma}^r = \widehat{Z}_\mu\mathrm{U}_r\mathrm{V}_t^\top\widehat{S}$ is given by $\widehat{G}_{\mu,\gamma}^r = \sum_{i\in[r]}(\mu-\widehat{\lambda}_i)^{-1}\widehat{h}_i \otimes \widehat{g}_i$, where $\widehat{\lambda}_i = \mu - 1/\nu_i$, $\widehat{g}_i = \nu_i^{-1/2}\widehat{S}^*\mathrm{V}_r w_i^\ell$ and $\widehat{h}_i = \widehat{Z}_\mu^*\mathrm{U}_r w_i^r$ for $\mathrm{U}_r = (\mu\gamma)^{-1}[\mu^{-1/2}I \,|\, -\mathrm{N}(\mathrm{M}+\gamma\mu I)^{-1}]^\top(\mathrm{KV}_r - \mu\mathrm{V}_r\Sigma_r^2)\in\mathbb{R}^{(1+p)n\times r}$.*

The main computational cost of our method, in view of (12), to solve SDE (1) lies in the implicit inversion of $\mathrm{J}_{\mu,\gamma}$ when solving (20). When computed with direct solvers this inversion is of the order $\mathcal{O}(n^3 p^3)$, however leveraging on the fact that $\mu\mathrm{J}_{\mu,\gamma}$ is Schur's complement of the $(1+p)n\times(1+p)n$ symmetric positive definite matrix and using classical iterative solvers, like Lanczos or the generalized Davidson method, when $r \ll n$ this cost can significantly be reduced to $\mathcal{O}(r\,n^2 p^2)$, c.f. [18].

## 5 Spectral learning bounds

Recalling Prop. 2, in order to obtain the bounds on eigenvalues and eigenfunctions of the generator, it suffices to analyze the learning rates for the operator norm error $\mathcal{E}$ and metric distortion $\eta$. For this purpose, we analyze the operator norm error of empirical estimator $\widehat{G}_{\mu,\gamma}^r$ using the decomposition

$$\mathcal{E}(\widehat{G}) \leq \underbrace{\|(\mu I - L)^{-1}Z_\mu - Z_\mu G_{\mu,\gamma}\|_{\mathcal{H}\to\mathcal{W}}}_{\text{regularization bias}} + \underbrace{\|Z_\mu(G_{\mu,\gamma} - G_{\mu,\gamma}^r)\|_{\mathcal{H}\to\mathcal{W}}}_{\text{rank reduction bias}} + \underbrace{\|Z_\mu(G_{\mu,\gamma}^r - \widehat{G}_{\mu,\gamma}^r)\|_{\mathcal{H}\to\mathcal{W}}}_{\text{estimator's variance}},$$

where $G_{\mu,\gamma} = W_{\mu,\gamma}^{-1}C$ is the minimizer of the full (i.e. without rank constraint) Tikhonov regularized risk and $G_{\mu,\gamma}^r = W_{\mu,\gamma}^{-1/2}[\![W_{\mu,\gamma}^{-1/2}C]\!]_r$ is the population version of the empirical estimator $\widehat{G}_{\mu,\gamma}^r$.

Note that, while the last two terms in the above decomposition depend on the estimator, the first term depends only on the choice of $\mathcal{H}$ and the regularity of $L$ w.r.t. $\mathcal{H}$. In this work we focus on the classical kernel-based learning where one chooses a universal kernel [45, Chapter 4] and controls the regularization bias with a regularity condition. For details see Rem. 2 of App. E.2. Let $\mu > 0$ be a prescribed parameter, we make following assumptions to quantify the difficulty of the learning problem:

**(BK)** *Boundedness.* There exists $c_{\mathcal{W}} > 0$ such that $\operatorname{ess\,sup}_{x \sim \pi} \|w\phi(x)\|^2 \leq c_{\mathcal{W}}$, i.e. $w\phi \in \mathcal{L}_\pi^\infty(\mathcal{X}, \mathcal{H}^{1+p})$;

**(RC)** *Regularity.* For some $\alpha \in (0, 2]$ there exists $c_\alpha > 0$ such that $C^2 \preceq c_\alpha^2 W_\mu^{1+\alpha}$;

**(SD)** *Spectral Decay.* There exists $\beta \in (0, 1]$ and $c_\beta > 0$ s.t. $\lambda_j(W_\mu) \leq c_\beta\, j^{-1/\beta}$, for all $j \in J$.

The above assumptions, discussed in more details in App E.1, are in the spirit of state-of-the-art analysis of statistical learning theory of classical regression in RKHS spaces [14], recently extended to regression of transfer operators [35, 28]. In our setting, instead of relying on the injection into $\mathcal{L}_\pi^2$ as in the case of transfer operators, the relevant object is the injection $Z_\mu$ into the energy space $\mathcal{W}$.

The first assumption **(BK)** is the main limiting factor of our approach. Indeed, since $\|w\phi(x)\|^2 = \mu\|\phi(x)\|^2 + \sum_{k \in [p]}\|d_k\phi(x)\|^2$, apart from needing the kernel to be bounded, we also need the Dirichlet form embedding to be bounded. Essentially, this means that the Dirichlet coefficients are not growing too fast w.r.t. the kernel's gradients decay. Having this, we assure that $Z_\mu \in \mathrm{HS}\,(\mathcal{H}, \mathcal{W}_\pi^\mu(\mathcal{X}))$ which implies the operator norm error can be controlled.

Another key difference between generator and transfer operator regression is that the covariance w.r.t. the domain of the operator becomes $W_\mu = Z_\mu^* Z_\mu$ instead of $C = S_\pi^* S_\pi$. On the other hand, the "cross-covariance" that captures now RKHS cross-correlations of the resolvent w.r.t domain is simply $Z_\mu^*(\mu I - L)^{-1}Z_\mu = S_\pi^*(\mu I - L)(\mu I - L)^{-1}S_\pi = C$. With this in mind, **(RC)** corresponds to the regularity condition in [28] and it quantifies the relationship between the hypothesis class (bounded operators in $\mathcal{H}$) and the object of interest $(\mu I - L)^{-1}$. Indeed, if $L$ has eigenfunctions that belong to $\alpha$-interpolation space between $\mathcal{H}$ and $\mathcal{W}_\pi^\mu(\mathcal{X})$, **(RC)** holds true. In particular, if $f_i \in \mathcal{H}$ for all $i \geq 2$ (constant eigenfunction excluded), one has that $\alpha \geq 1$ (c.f. Proposition 7 in App. E.1).

Finally, **(SD)** quantifies the "size" of the hypothetical domain $\mathcal{H}$ within the true domain $\mathcal{W}_\pi^\mu(\mathcal{X})$ via the *effective dimension* $\mathrm{tr}(W_{\mu,\gamma}^{-1}W_\mu) \leq c_\beta(\mu\gamma)^{-\beta}$, which, due to **(BK)**, leads to another notion, known as the embedding property, quantifying the relationship between $\mathcal{H}$ and essentially bounded functions in the domain of the operator

**(KE)** *Kernel Embedding.* There exists $\tau \in [\beta, 1]$ and such that
$$c_\tau = \operatorname{ess\,sup} \sum_{j \in \mathbb{N}} \sigma_j^{2\tau}[\mu|z_j(x)|^2 - z_j(x)[Lz_j](x)] < +\infty, \tag{21}$$
where $Z_\mu = \sum_{j \in J} \sigma_j z_j \overset{x \sim \pi}{\otimes} h_j$ is the SVD of the injection operator $Z_\mu : \mathcal{H} \to \mathcal{W}_\pi^\mu(\mathcal{X})$.

Using the above assumptions, we prove generalization bound for the RRR estimator, notably addressing the general case when the prior knowledge of the Dirichlet coefficient is inexact, i.e.

**(DF)** *Dirichlet form.* For some $\epsilon \in [0, 1)$ there exists $s_\epsilon : \mathbb{R}^d \to \mathbb{R}^p$ so that $(1-\epsilon)s^\top s \preceq s_\epsilon^\top s_\epsilon \preceq (1+\epsilon)s^\top s$ holds $\pi$-a.e., where $s : \mathbb{R}^d \to \mathbb{R}^p$ is such that $L = B^*B$ for $B = s^\top\nabla : \mathcal{L}_\pi^2(\mathcal{X}) \to [\mathcal{L}_\pi^2(\mathcal{X})]^p$.

Recalling the form of IG in Equation 3, the Dirichlet form in **(DF)** of a self-adjoint generator exists whenever the positive definite diffusion part satisfies uniform ellipticity conditions and the drift term allows integration by parts, leading to $s(x) = b(x)/\sqrt{2}$. Thus, to obtain the partial knowledge we need, it is enough to estimate the diffusion function $s_\epsilon$ with the relative error bound in **(DF)**, see App. E.1.

**Theorem 2.** *Let* **(DF)***,* **(RC)***,* **(SD)***, and* **(KE)** *hold for some* $\epsilon \in [0, 1)$*,* $\alpha \in (0, 2]$*,* $\beta \in (0, 1]$ *and* $\tau \in [\beta, 1]$*, respectively, and let* $\mathrm{cl}(\mathrm{Im}(S_\pi)) = \mathcal{L}_\pi^2(\mathcal{X})$*. Denoting* $\lambda_k^\star = \lambda_k(S_\pi^*(\mu I - L)^{-1}S_\pi)$*,* $k \in \mathbb{N}$*, and given* $\delta \in (0, 1)$ *and* $r \in [n]$*, if RRR estimator is built from the Dirichlet coefficients* $s_\epsilon : \mathbb{R}^d \to \mathbb{R}^p$ *and*
$$\gamma \asymp n^{-\frac{1}{\alpha+\beta}} \text{ and } \varepsilon_n^\star = n^{-\frac{\alpha}{2(\alpha+\beta)}} \text{ when } \alpha \geq \tau, \text{ or } \gamma \asymp n^{-\frac{1}{\beta+\tau}} \text{ and } \varepsilon_n^\star = n^{-\frac{\alpha}{2(\beta+\tau)}} \text{ when } \alpha < \tau, \tag{22}$$
*then there exists a constant* $c > 0$*, depending only on* $\mathcal{H}$ *and gap* $\lambda_r^\star - \lambda_{r+1}^\star > 0$*, such that for large enough* $n \geq r$ *with probability at least* $1 - \delta$ *in the i.i.d. draw of* $\mathcal{D}_n$ *from* $\pi$ *it holds that*
$$\mathcal{E}(\widehat{G}_{\mu,\gamma}^r) \leq (\widehat{\sigma}_{r+1} \wedge \sqrt{\lambda_{r+1}^\star}) + c\left(\varepsilon_n^\star \ln\delta^{-1} + \epsilon\right). \tag{23}$$

*Proof sketch.* The *regularization bias* is bounded by $c_\alpha \gamma^{\alpha/2}$ by Prop. 9 of App. E.2, the *rank reduction bias* is upper bounded by $\lambda_{r+1}(S_\pi^*(\mu I - L)^{-1}S_\pi)$, while in the exact knowledge case ($\epsilon=0$) the bounds on the variance terms critically rely on the well-known perturbation result for spectral projectors reported in Prop. 4, App. A. The latter is then chained to Pinelis-Sakhanenko's inequality and Minsker's inequality for self-adjoint HS-operators, Props. 12 and 13 in App. E.3.1, respectively. When the knowledge is not exact, that is $\epsilon > 0$ in **(DF)**, this relative bound implies that $\epsilon \widehat{W}_\mu \preceq \widehat{W}_\mu - \widehat{W}_\mu^\epsilon \preceq \epsilon \widehat{W}_\mu$, where the empirical covariance with the inexact Dirichlet coefficient $s_\epsilon$ is denoted by $\widehat{W}_\mu^\epsilon$. This allows one to control the additional approximation error in the analysis of variance, paying the price of additive term $\epsilon$. Combining the bias and variance terms, we obtain the balancing equations for the regularization parameter and then the next result follows. $\square$

First, note that the learning rate (23) implies the $\mathcal{L}_\pi^2$-norm learning rate. Moreover, for $\alpha \geq \tau$, it matches information theoretic lower bounds for transfer operator learning upon replacing parameters $\alpha$, $\beta$ and $\tau$ related to the space $\mathcal{W}$ with their $\mathcal{L}_\pi^2$ analogues [28], see App. E.6. This motivates the development of the first mini-max optimality for the IG learning, for which our results are an important first step. Next, remark that Theorem 2 guarantees the reliability of fully data-driven methods when the diffusion coefficients are not known but estimated. Furthermore, when $b$ is constant or linear (e.g. Overdamped Langevin and CIR), the classical estimation bounds coincide with the relative error bound of assumption **(DF)**.

To conclude this section, we address the spectral learning bounds stemming from the Prop. 2. The main task to achieve this is to control the metric distortions, which we demonstrate in App. E.5. In this context, an additional assumption $\alpha \geq 1$ is needed, since otherwise the metric distortions can blow-up due to eigenfunctions being out of the RKHS space. Importantly, our analysis reveals that

$$\frac{|\lambda_i - \widehat{\lambda}_i|}{|\mu - \lambda_i||\mu - \widehat{\lambda}_i|} \leq \left(\widehat{s}_i \wedge 2\sqrt{\lambda_{r+1}^\star/\lambda_r^\star}\right) + c\left(\varepsilon_n^\star \ln \delta^{-1} + \epsilon\right), \ i \in [r], \tag{24}$$

where $\widehat{s}_i = \widehat{\sigma}_i \widehat{\eta}_i$ is the empirical spectral bias that informs how good is the estimation of the particular eigenpair is (see Fig. 1 a)), $\widehat{\sigma}_i$ being given in (20) and $\widehat{\eta}_i = \|\widehat{h}_i\|_\mathcal{H}/\|(\widehat{W}_\mu^\epsilon)^{1/2}\widehat{h}_i\|_\mathcal{H}$. Importantly, (24) reveals that our data-driven method for spectral decomposition of differential operator $L$ does not suffer from the curse of dimensionality as present in the classical numerical methods, see App. E.6.

## 6 Experiments

In this section, we showcase the key features of our method outlined in Table 1. We demonstrate that our approach: (1) avoids the spurious effects noted in other IG methods [19, 1], (2) is more effective than transfer operator methods [27], and (3) validates our bounds in a prediction task for a model with a non-constant diffusion term. Further details are available in App. F.

**One dimensional four well potential** We first investigate the overdamped Langevin dynamics in a potential that presents four different wells, two principal wells and then in each of them two smaller wells, given by $V(x) = 4(x^8 + 0.8\exp(-80(x^2)) + 0.2\exp(-80(x-0.5)^2) + 0.5\exp(-40(x+0.5)^2))$. This leads to three relevant eigenpairs: the slowest mode corresponds to the transition between the two principal wells, while the others two capture transitions between the smaller wells. In Fig. 1 panel **a)**, we show that the empirical bias $\widehat{s}_1 = \widehat{\sigma}_1 \widehat{\eta}_1$ allows us to choose the hyperparameters of the model, that is higher empirical bias coincides with unreliable estimation of the operator's eigenfunction. In panel **b)** we observe how it varies w.r.t. land-scale (y-axis) and regularization $\gamma$ (x-axis) hyperparameters showcasing the robustness of the model, see also Fig. 3 of App. F for hyperparameter $\mu$. Further, in panel **c)** we show the consistency of our model with the true Boltzmann distribution. Namely, we use our model to forecast the conditional probability density function (pdf) of the system. We perform the same procedure with prefect knowledge and imperfect diffusion coefficient estimated from data. We also report that if the same approach is used with the method described in [19, 43, 1], no dynamical quantity can be forecast due to the presence of numerous spurious eigenpairs, which prevent the system from relaxing towards the Boltzmann distribution. This issue is further illustrated in panel **d)** where we show how, contrary to KRR method of [19, 1], we avoid spuriousness in the estimation of eigenvalues.

**CIR model** Next, with the CIR model we show that our method is not limited to Langevin process with constant diffusion. For this process, the conditional expectation of the state $X_t$ is analytically known. We can thus compare the prediction of our model with respect to this expectation using

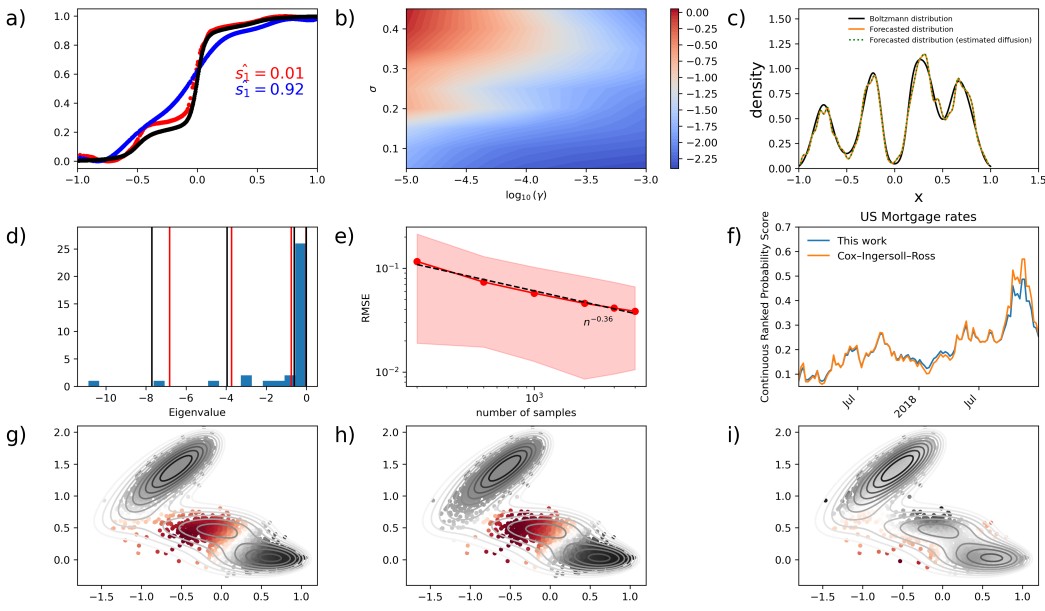

Figure 1: **a)** Empirical biases $\hat{s}_1 = \hat{\sigma}_1 \hat{\eta}_1$ and estimation of the first (nontrivial) eigenfunction of the IG of a Langevin process under a four-well potential. The ground truth is shown in black, our method RRR is red and blue for two different kernel lengthscales. **d)** Estimation by our method (black) of the eigenvalues for the same process (red) compared to the methods in [19, 1], for which eigenvalue histogram in blue shows spuriousness. **e)** Prediction RMSE for the CIR model w.r.t. number of samples. **f)** Performance of our data-driven method and fitted CIR model on the real data of US mortgage rates. **g)** The second eigenfunction of a Langevin process under Muller brown potential (white level lines) with its estimation by RRR **h)** and Transfer Operator (TO) in **i)**. Observe that TO fails to recover the metastable state.

root mean squared error (RMSE) and compute it for different number of samples to validate our bounds. Conditional expectation were computed on 100 different simulations at $t = \ln(2)/a$ which corresponds to the half life of the mean reversion. Results are shown in panel **e)** of Fig. 1.

**US mortgage rates** We have trained our method on a real 30-year US mortgage rates dataset and contrasted it with the fitted CIR model using continuous ranked probability scores that are estimated from the forecasts obtained by of each of them, see panel **f)** of Fig. 1. Each model has been trained using data from January 2009 to December 2016. The initial condition was the last week of December 2016 and the predictions were made for the years 2017 and 2018. Since the dataset is real, we used the imperfect partial knowledge, that is, for our method, we estimated the diffusion coefficient only via a least squares calibration of a CIR model over the training set. This allows more flexibility on the drift term in our model.

**Muller-Brown potential** We next study Langevin dynamics under more challenging conditions: the Muller-Brown potential. Panels **g)**-**h)**-**i)** of Fig. 1 depict the second eigenfunction obtained by our method compared to the ground truth one, as well as the one found by the transfer operator approach, with the same number of samples. Notably, our physics informed approach outperforms transfer operator learning for this task. Note that with different lag times, we were able to recover this second eigenfunction.

## 7 Conclusion

We developed a novel energy-based framework for learning the Infinitesimal Generator of stochastic diffusion SDEs using kernel methods. Our approach integrates physical priors, achieves fast error rates, and provides the first spectral learning guarantees for generator learning. A limitation is its computational complexity, scaling as $n^2 d^2$. Future work will explore alternative methods to enhance computational efficiency and investigate a broader suite of SDEs beyond stochastic diffusion.

## Acknowledgements

This work was partially funded by EU Project ELIAS under grant agreement No. 101120237, and by the European Union - NextGenerationEU initiative and the Italian National Recovery and Resilience Plan (PNRR) from the Ministry of University and Research (MUR), under Project PE0000013 CUP J53C22003010006 "Future Artificial Intelligence Research (FAIR)". We also extend our gratitude to the anonymous reviewers for their valuable feedback, which inspired us to enhance the quality and depth of our results.

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

# Supplementary Material

This appendix includes additional background on SDE and RKHS, proofs of the results omitted in the main body and information about the numerical experiments.

| notation | meaning | notation | meaning |
|---|---|---|---|
| $\wedge$ | minimum | $\vee$ | maximum |
| $[\![\cdot]\!]_r$ | $r$-truncated SVD of an operator | $I$ | identity operator |
| $\mathrm{HS}\,(\mathcal{H},\mathcal{G})$ | space of Hilbert-Schmidt operators $\mathcal{H}\to\mathcal{G}$ | $\mathrm{B}_r(\mathcal{H})$ | set of rank-$r$ Hilbert-Schmidt operators on $\mathcal{H}$ |
| $\|A\|$ | operator norm of an operator $A$ | $\|A\|_{\mathrm{HS}}$ | Hilbert-Schmidt norm of operator $A$ |
| $\sigma_i(\cdot)$ | $i$-th singular value of an operator | $\lambda_i(\cdot)$ | $i$-th eigenvalue of an operator |
| $\mathcal{X}$ | state space of the Markov process | $(X_t)_{t\geq 0}$ | time-homogeneous Markov process |
| $p$ | transition kernel of the Markov process | $\pi$ | invariant measure of the Markov process |
| $a$ | drift of the Itô process | $b$ | diffusion of the Itô process |
| $\mathcal{L}^2_\pi(\mathcal{X})$ | L2 space of functions on $\mathcal{X}$ w.r.t. measure $\pi$ | $A$ | transfer operator on $\mathcal{L}^2_\pi(\mathcal{X})$ |
| $\mathcal{W}^{1,2}_\pi(\mathcal{X})$ | Sobolev space w.r.t. measure $\pi$ on $\mathcal{X}$ | $L$ | generator of the semigroup on $\mathcal{W}^{1,2}_\pi(\mathcal{X})$ |
| $s$ | Dirichlet form diffusion | $B$ | Dirichlet form operator on $\mathcal{W}^{1,2}_\pi(\mathcal{X})$ |
| $\mu$ | shift parameter | $\mathcal{W}^{\mu}_\pi(\mathcal{X})$ | regularized energy space |
| $k(x,y)$ | kernel | $\phi$ | canonical feature map |
| $\mathcal{H}$ | reproducing kernel Hilbert space | $S_\pi$ | canonical injection $\mathcal{H}\hookrightarrow\mathcal{L}^2_\pi(\mathcal{X})$ |
| $\ell\phi$ | generator embedding | $Z_\mu$ | canonical injection $\mathcal{H}\hookrightarrow\mathcal{W}^{\mu}_\pi(\mathcal{X})$ |
| $D^v$ | $v$-directional derivative | $D$ | derivative |
| $D^v\phi$ | embedding of the $v$-directional derivative | $D\phi$ | derivative embedding |
| $w_k\phi$ | $k$-th component of Dirichlet operator embedding | $w\phi$ | Dirichlet operator embedding |
| $\sigma_j$ | $j$-th singular value of $Z_\mu$ | $J$ | countable index set of singular values of $Z_\mu$ |
| $z_j$ | $j$-th left singular function of $Z_\mu$ | $h_j$ | $j$-th right singular function of $Z_\mu$ |
| $\mathbb{1}$ | function in $\mathcal{L}^2_\pi(\mathcal{X})$ with the constant output 1 | $\gamma$ | regularization parameter |
| $\mathcal{R}$ | true risk | $\mathcal{E}$ | operator norm error |
| $\mathcal{E}_{\mathrm{HS}}$ | excess risk, i.e. HS norm error | $\mathcal{R}_0$ | irreducible risk |
| $\mathcal{D}_n$ | dataset $(x_i)_{i\in[n]}$ | $\widehat{\mathcal{R}}$ | empirical risk |
| $\widehat{S}$ | sampling operator w.r.t. $\mathcal{L}^2_\pi(\mathcal{X})$ | $\widehat{Z}_\mu$ | sampling operator w.r.t. $\mathcal{W}^{\mu}_\pi(\mathcal{X})$ |
| $C$ | covariance operator w.r.t. $\mathcal{L}^2_\pi(\mathcal{X})$ | $\widehat{C}$ | empirical covariance operator w.r.t. $\mathcal{L}^2_\pi(\mathcal{X})$ |
| $C_\gamma$ | regularized covariance operator w.r.t. $\mathcal{L}^2_\pi(\mathcal{X})$ | $\widehat{C}_\gamma$ | regularized empirical covariance operator w.r.t. $\mathcal{L}^2_\pi(\mathcal{X})$ |
| $W_\mu$ | covariance operator w.r.t. $\mathcal{W}^{\mu}_\pi(\mathcal{X})$ | $\widehat{W}_\mu$ | empirical covariance operator w.r.t. $\mathcal{W}^{\mu}_\pi(\mathcal{X})$ |
| $W_{\mu,\gamma}$ | regularized covariance operator w.r.t. $\mathcal{W}^{\mu}_\pi(\mathcal{X})$ | $\widehat{W}_{\mu,\gamma}$ | regularized empirical covariance operator w.r.t. $\mathcal{W}^{\mu}_\pi(\mathcal{X})$ |
| $T$ | derivative-covariance operator | $\widehat{T}$ | empirical derivative-covariance operator |
| $\mathrm{K}$ | kernel Gram matrix w.r.t. $\mathcal{L}^2_\pi(\mathcal{X})$ | $\mathrm{K}_\gamma$ | regularized kernel Gram matrix w.r.t. $\mathcal{L}^2_\pi(\mathcal{X})$ |
| $\mathrm{F}_\mu$ | kernel Gram matrix w.r.t. $\mathcal{W}^{\mu}_\pi(\mathcal{X})$ | $\mathrm{F}_{\mu,\gamma}$ | regularized kernel Gram matrix w.r.t. $\mathcal{W}^{\mu}_\pi(\mathcal{X})$ |
| $\mathrm{M}$ | derivative-derivative kernel Gram matrix | $\mathrm{N}$ | feature-derivative kernel Gram matrix |
| $G$ | population estimator of $(\mu I-L)^{-1}$ on $\mathcal{H}$ | $\widehat{G}$ | empirical estimator of $(\mu I-L)^{-1}$ on $\mathcal{H}$ |
| $G_{\mu,\gamma}$ | population KRR estimator | $\widehat{G}_{\mu,\gamma}$ | empirical KRR estimator |
| $G^r_{\mu,\gamma}$ | population RRR estimator | $\widehat{G}^r_{\mu,\gamma}$ | empirical RRR estimator |
| $P$ | spectral projector | $\widehat{P}$ | empirical spectral projector |
| $\eta$ | metric distortion | $\widehat{\eta}$ | empirical metric distortion |
| $\lambda$ | generator eigenvalue | $\widehat{\lambda}$ | eigenvalue of the empirical estimator |
| $f$ | generator eigenfunction in $\mathcal{L}^2_\pi(\mathcal{X})$ | $\widehat{f}$ | empirical eigenfunction in $\mathcal{L}^2_\pi(\mathcal{X})$ |
| $\widehat{h}$ | right empirical eigenfunction | $\widehat{g}$ | left empirical eigenfunction |
| $c_{\mathcal{W}}$ | boundness constant | $P_{\mathcal{H}}$ | orthogonal projector in $\mathcal{W}^{\mu}_\pi(\mathcal{X})$ onto $\mathrm{Im}(Z_\mu)$ |
| $\alpha$ | regularity parameter | $c_\alpha$ | regularity constant |
| $\beta$ | spectral decay parameter | $c_\beta$ | spectral decay constant |
| $\tau$ | embedding parameter | $c_\tau$ | embedding constant |

Table 2: Summary of used notations.

# A  Background

## A.1  Basics on operator theory for Markov processes.

We provide here some basics on operator theory for Markov processes. Let $\mathcal{X} \subset \mathbb{R}^d$ $(d \in \mathbb{N})$ and $(X_t)_{t \in \mathbb{R}_+}$ be a $\mathcal{X}$-valued time-homogeneous Markov process defined on a filtered probability space $(\Omega, \mathcal{F}, (\mathcal{F}_t)_{t \in \mathbb{R}_+}, \mathbb{P})$ where $\mathcal{F}_t = \sigma(X_s, s \leq t)$ is the natural filtration of $(X_t)_{t \in \mathbb{R}_+}$. The dynamics of $X$ can be described through of a family of probability densities $(p_t)_{t \in \mathbb{R}_+}$ such that for all $t \in \mathbb{R}_+$, $E \in \mathcal{B}(\mathcal{X})$,

$$\mathbb{P}(X_t \in E | X_0 = x) = \int_E p_t(x, y) dy.$$

Let $\mathcal{G}$ be a set of real valued and measurable functions on $\mathcal{X}$. For any $t \in \mathbb{R}_+$ the *transfer operator* (TO) $A_t : \mathcal{G} \to \mathcal{G}$ maps any measurable function $f \in \mathcal{G}$ to

$$(A_t f)(x) = \int_{\mathcal{X}} f(y) p_t(x, y) dy. \tag{25}$$

In theory of Markov processes, the family $(A_t)_{t \in \mathbb{R}_+}$ is referred to as the *Markov semigroup* associated to the process $X$.

**Remark 1.** *A possible choice is $\mathcal{G} = \mathcal{L}^{\infty}(\mathcal{X})$. Here, we are interested in another choice of $\mathcal{G}$ related to the existence of an invariant measure for $A_t$ i.e., a $\sigma$-finite measure $\pi$ on $(\mathcal{X}, \mathcal{B}(\mathcal{X}))$ such that $P_t^* \pi = \pi$ for any $t \in \mathbb{R}_+$. In that case, we can choose $\mathcal{G} = \mathcal{L}_{\pi}^2(\mathcal{X})$ so that, for all $f \in \mathcal{L}_{\pi}^2(\mathcal{X})$, $P_t f$ converges to $f$ in $\mathcal{L}_{\pi}^2(\mathcal{X})$ as $t$ goes to $0$. Note that $P_0 f = f$ and $P_{\infty} f = \int f d\mathcal{X}$ for a suitable integrable function $f : \mathcal{X} \to \mathbb{R}$.*

Within the existence of this invariant measure $\pi$, the process $X$ is then characterized by the *infinitesimal generator* (IG) $L : \mathcal{L}_{\pi}^2(\mathcal{X}) \to \mathcal{L}_{\pi}^2(\mathcal{X})$ of the family $(A_t)_{t \in \mathbb{R}_+}$ defined by

$$L = \lim_{t \to 0^+} \frac{A_t - I}{t}. \tag{26}$$

In other words, $L$ characterizes the linear differential equation $\partial_t A_t f = L A_t f$ satisfied by the transfer operator. The domain of $L$ denoted $\mathrm{dom}(L)$ coincides with the the Sobolev space $\mathcal{W}_{\pi}^{1,2}(\mathcal{X})$

$$\mathcal{W}_{\pi}^{1,2}(\mathcal{X}) = \{ f \in \mathcal{L}_{\pi}^2(\mathcal{X}) \mid \|f\|_{\mathcal{W}}^2 = \|f\|_{\mathcal{L}_{\pi}^2} + \|\nabla f\|_{\mathcal{L}_{\pi}^2} < \infty \}.$$

The spectrum of the IG can be difficult to capture due to the potential unboundedness of $L$. To circumvent this problem, one can focus on an auxiliary operator, the resolvent, which, under certain conditions, shares the same eigenfunctions as $L$ and becomes compact. The following result can be found in Yosida's book ([53], Chap. IX): For every $\mu > 0$, the operator $(\mu I - L)$ admits an inverse $L_{\mu} = (\mu I - L)^{-1}$ that is a continuous operator on $\mathcal{X}$ and

$$(\mu I - L)^{-1} = \int_0^{\infty} e^{-\mu t} A_t dt.$$

The operator $L_{\mu}$ is the *resolvent* of $L$ and the corresponding *resolvent set* of $L$ is defined by

$$\rho(L) = \{ \mu \in \mathbb{C} \mid (\mu I - L) \text{ is bijective and } L_{\mu} \text{ is continuous} \}.$$

In fact, $\rho(L)$ contains all real positive numbers and $(\mu I - L)^{-1}$ is bounded. Here, we assume that $L$ has *compact resolvent*, i.e. $\rho(L) \neq \emptyset$ and there exists $\mu_0 \in \rho(L)$ such that $L_{\mu_0}$ is compact. Note that, through the resolvent identity, this implies the compactness of all resolvents $L_{\mu}$ for $\mu \in \rho(L)$. Let then $\mu \in \rho(L)$. As $(L, \mathrm{dom}(L))$ is assumed to be self-adjoint, so does $L_{\mu}$, so that $L_{\mu}$ is both compact and self-adjoint. Then, its *spectrum* $\mathrm{Sp}(L_{\mu}) = \mathbb{C} \setminus \rho(L_{\mu})$ is purely discrete and consists of isolated eigenvalues $(\lambda_i^{\mu})_{i \in \mathbb{N}}$ such that $|\lambda_i^{\mu}| \to 0$ associated with an orthonormal basis $(f_i)_{i \in \mathbb{N}}$ (see [53], chapter XI). In other words, the spectral decomposition of the resolvent writes

$$L_{\mu} = \sum_{i \in \mathbb{N}} \lambda_i^{\mu} f_i \otimes f_i$$

where the functions $f_i \in \mathcal{L}_{\pi}^2(\mathcal{X})$ are also eigenfunctions of the operator $L$ we get

$$L = \sum_{i \in \mathbb{N}} \lambda_i f_i \otimes f_i.$$

**Example 3** (Langevin). *Let* $(k_bT) \in \mathbb{R}$. *The* overdamped Langevin equation *driven by a* potential $V : \mathbb{R}^d \to \mathbb{R}$ *is given by* $dX_t = -\nabla V(X_t)dt + \sqrt{2(k_bT)}dW_t$ *and* $X_0 = x$. *Its invariant measure of the solution process* $X$ *is the* Boltzman distribution $\pi(dx) = z^{-1}e^{-(k_bT)^{-1}V(x)}dx$ *where* $z$ *denotes a normalizing constant. Its infinitesimal generator* $L$ *is defined by* $Lf = -\nabla V^\top \nabla f + (k_bT)\Delta f$, *for* $f \in \mathcal{W}_\pi^{1,2}(\mathcal{X})$, *and if* $\nabla V \in \mathcal{L}_\pi^2(\mathcal{X})$ *is positive and coercive, it has compact resolvent. Finally, since* $\int(-Lf)g\,d\pi = -\int \left[\nabla\left((k_bT)\nabla f(x)\frac{e^{-(k_bT)^{-1}V(x)}}{Z}\right)\right]g(x)dx = (k_bT)\int \nabla f^\top \nabla g\,d\pi = \int f(-Lg)\,d\pi$, *generator* $L$ *is self-adjoint and associated to a gradient Dirichlet form with* $s(x) = (k_bT)^{1/2}(\delta_{ij})_{i\in[d],j\in[p]}$.

**Example 4** (Cox-Ingersoll-Ross process). *Let* $d = 1$, $a, b \in \mathbb{R}$, $\sigma \in \mathbb{R}_+^*$. *The* Cox-Ingersoll-Ross *process is solution of the SDE* $dX_t = (a + bX_t)dt + \sigma\sqrt{X_t}dW_t$ *and* $X_0 = x$. *Its invariant measure* $\pi$ *is a Gamma distribution with shape parameter* $a/\sigma^2$ *and rate parameter* $b/\sigma^2$. *Its infinitesimal generator* $L$ *is defined for* $f \in \mathcal{L}_\pi^2(\mathcal{X})$ *by* $Lf = (a + bx)\nabla f + \frac{\sigma^2 x}{2}\Delta f$. *Note that by integration by parts, we can check that the generator* $L$ *satisfies* $\int(-Lf)g\,d\pi = \int \frac{\sigma^2 x}{2}f'(x)g'(x)\,\pi(dx) = \int f(-Lg)\,d\pi$, *and it is associated to a gradient Dirichlet form with* $s(x) = \sigma\sqrt{x}/\sqrt{2}$.

## A.2 Infinitesimal generator for diffusion processes

After defining the infinitesimal generator for Markov processes (see A.1), we provide its explicit form for solution processes of equations like(1). Given a smooth function $f \in \mathcal{C}^2(\mathcal{X}, \mathbb{R})$, Itô's formula (see for instance [4], B, p.495) provides for $t \in \mathbb{R}_+$,

$$f(X_t) - f(X_0) = \int_0^t \sum_{i=1}^d \partial_i f(X_s)dX_s^i + \frac{1}{2}\int_0^t \sum_{i,j=1}^d \partial_{ij}^2 f(X_s)d\langle X^i, X^j\rangle_s$$

$$= \int_0^t \nabla f(X_s)^\top dX_t + \frac{1}{2}\int_0^t \text{Tr}\left[X_s^\top(\nabla^2 f)(X_s)X_s\right]ds.$$

Recalling (1), we get

$$f(X_t) = f(X_0) + \int_0^t \left[\nabla f(X_s)^\top a(X_s) + \frac{1}{2}\text{Tr}\left[b(X_s)^\top(\nabla^2 f(X_s))b(X_s)\right]\right]ds$$

$$+ \int_0^t \nabla f(X_s)^\top b(X_s)dW_s. \tag{27}$$

Provided $f$ and $b$ are smooth enough, the expectation of the last stochastic integral vanishes so that we get

$$\mathbb{E}[f(X_t)|X_0 = x] = f(x) + \int_0^t \mathbb{E}\left[\nabla f(X_s)^\top a(X_s) + \frac{1}{2}\text{Tr}\left[b(X_s)^\top(\nabla^2 f(X_s))b(X_s)\right]\Big|X_0 = x\right]ds$$

Recalling that $L = \lim_{t\to 0^+}(A_t f - f)/t$, we get for every $x \in \mathcal{X}$,

$$Lf(x) = \lim_{t\to 0}\frac{\mathbb{E}[f(X_t)|X_0 = x] - f(x)}{t}$$

$$= \lim_{t\to 0}\frac{1}{t}\left[\int_0^t \mathbb{E}\left[\nabla f(X_s)^\top a(X_s) + \frac{1}{2}\text{Tr}\left[(X_s)^\top(\nabla^2 f(X_s))b(X_s)\right]\right]ds\Big|X_0 = x\right]$$

$$= \nabla f(x)^\top a(x) + \frac{1}{2}\text{Tr}\left[b(x)^\top(\nabla^2 f(x))b(x)\right], \tag{28}$$

which provides the closed formula for the IG associated with the solution process of (1).

## A.3 Spectral perturbation theory

Recalling that for a bounded linear operator $A$ on some Hilbert space $\mathcal{H}$ the *resolvent set* of the operator $A$ is defined as $\rho(A) = \{\lambda \in \mathbb{C}\,|\,A - \lambda I$ is bijective and $(A - \lambda I)^{-1}$ is continuous$\}$, and its *spectrum* $\text{Sp}(A) = \mathbb{C} \setminus \{\rho(A)\}$, let $\lambda \subseteq \text{Sp}(A)$ be isolated part of spectra, i.e. both $\lambda$ and $\mu = \text{Sp}(A) \setminus \lambda$ are closed in $\text{Sp}(A)$. Than, the *Riesz spectral projector* $P_\lambda : \mathcal{H} \to \mathcal{H}$ is defined by

$$P_\lambda = \frac{1}{2\pi}\int_\Gamma (zI - A)^{-1}dz, \tag{29}$$

where $\Gamma$ is any contour in the resolvent set $\mathrm{Res}(A)$ with $\lambda$ in its interior and separating $\lambda$ from $\mu$. Indeed, we have that $P_\lambda^2 = P_\lambda$ and $\mathcal{H} = \mathrm{Im}(P_\lambda) \oplus \mathrm{Ker}(P_\lambda)$ where $\mathrm{Im}(P_\lambda)$ and $\mathrm{Ker}(P_\lambda)$ are both invariant under $A$ and $\mathrm{Sp}(A_{|\mathrm{Im}(P_\lambda)}) = \lambda$, $\mathrm{Sp}(A_{|\mathrm{Ker}(P_\lambda)}) = \mu$. Moreover, $P_\lambda + P_\mu = I$ and $P_\lambda P_\mu = P_\mu P_\lambda = 0$.

Finally if $A$ is *compact* operator, then the Riesz-Schauder theorem, see e.g. [44], assures that $\mathrm{Sp}(T)$ is a discrete set having no limit points except possibly $\lambda = 0$. Moreover, for any nonzero $\lambda \in \mathrm{Sp}(T)$, then $\lambda$ is an *eigenvalue* (i.e. it belongs to the point spectrum) of finite multiplicity, and, hence, we can deduce the spectral decomposition in the form

$$A = \sum_{\lambda \in \mathrm{Sp}(A)} \lambda P_\lambda, \tag{30}$$

where geometric multiplicity of $\lambda$, $r_\lambda = \mathrm{rank}(P_\lambda)$, is bounded by the algebraic multiplicity of $\lambda$. If additionally $A$ is normal operator, i.e. $AA^* = A^*A$, then $P_\lambda = P_\lambda^*$ is orthogonal projector for each $\lambda \in \mathrm{Sp}(A)$ and $P_\lambda = \sum_{i=1}^{r_\lambda} \psi_i \otimes \psi_i$, where $\psi_i$ are normalized eigenfunctions of $A$ corresponding to $\lambda$ and $r_\lambda$ is both algebraic and geometric multiplicity of $\lambda$.

We conclude this section with well-known perturbation bounds for eigenfunctions and spectral projectors of self-adjoint compact operators.

**Proposition 3** ([11]). *Let $A$ be compact self-adjoint operator on a separable Hilbert space $\mathcal{H}$. Given a pair $(\widehat{\lambda}, \widehat{f}) \in \mathbb{C} \times \mathcal{H}$ such that $\|\widehat{f}\| = 1$, let $\lambda$ be the eigenvalue of $A$ that is closest to $\widehat{\lambda}$ and let $f$ be its normalized eigenfunction. If $\widehat{g} = \min\{|\widehat{\lambda} - \lambda| \,|\, \lambda \in \mathrm{Sp}(A) \setminus \{\lambda\}\} > 0$, then $\sin(\sphericalangle(\widehat{f}, f)) \leq \|A\widehat{f} - \widehat{\lambda}\widehat{f}\|/\widehat{g}$.*

**Proposition 4** ([57]). *Let $A$ and $\widehat{A}$ be two compact operators on a separable Hilbert space. For nonempty index set $J \subset \mathbb{N}$ let*

$$\mathrm{gap}_J(A) = \min\{|\lambda_i(A) - \lambda_j(A)| \,|\, i \in \mathbb{N} \setminus J, \, j \in J\}$$

*denote the spectral gap w.r.t $J$ and let $P_J$ and $\widehat{P}_J$ be the corresponding spectral projectors of $A$ and $\widehat{A}$, respectively. If $A$ is self-adjoint and for some $\|A - \widehat{A}\| < \mathrm{gap}_J(A)$, then*

$$\|P_J - \widehat{P}_J\| \leq \frac{\|A - \widehat{A}\|}{\mathrm{gap}_J(A)}.$$

# B   Representations in the RKHS

In section 2, we have defined the infinitesimal generator of a diffusion process and specified its form when associated with Dirichlet forms. These operators act on $\mathcal{L}_\pi^2(\mathcal{X})$ or specific subsets of it. To develop our learning procedure, we need to understand these operators' actions when embedding into the RKHS, and define their versions for feature maps.

**IG and Dirichlet operator in RKHS.** As a reminder, we consider $\mathcal{H}$ be an RKHS and let $k : \mathcal{X} \times \mathcal{X} \to \mathbb{R}$ be the associated kernel function. The canonical feature map is denoted by $\phi(x) = k(x, \cdot)$ for $x \in \mathcal{X}$ $k(x, x') = \langle \phi(x), \phi(x') \rangle$ for all $x, x' \in \mathcal{X}$. Assuming that $k$ is square-integrable with respect to the measure $\pi$, we define the injection operator $S_\pi : \mathcal{H} \hookrightarrow \mathcal{L}_\pi^2(\mathcal{X})$ and its adjoint $S_\pi^* : \mathcal{L}_\pi^2(\mathcal{X}) \to \mathcal{H}$ by $S_\pi^* f = \int_\mathcal{X} f(x)\phi(x)\pi(dx)$. As a preliminary step, we need to define the reproducing partial derivatives in RKHS, which we introduce via Mercer kernels.

**Definition 1** (Mercer kernel). *A kernel function $k : \mathcal{X} \times \mathcal{X} \to \mathbb{R}$ is called a Mercer kernel if it is a continuous and symmetric function such that for any finite set of points $\{x_1, \ldots, x_n\} \subset \mathcal{X}$, the matrix $(k(x_i, x_j))_{i,j=1}^n$ is positive semi-definite.*

Several standard kernels satisfy the Mercer property with $s \geq 1$, including the Gaussian kernel which we will consider subsequently.

For $s \in \mathbb{N}$ and $m \in \mathbb{N}$, we define the index set $I_s^m = \{\alpha \in \mathbb{N}^m : |\alpha| \leq s\}$ where $|\alpha| = \sum_{j=1}^s \alpha_j$, for $\alpha = (\alpha_1, \ldots, \alpha_m) \in \mathbb{N}^m$. For a function $f : \mathbb{R}^m \to \mathbb{R}$ and $x = (x_1, \ldots, x_m) \in \mathbb{R}^m$, we denote its partial derivative $D^\alpha f$ at point $x$ (if it exists) as

$$D^\alpha f(x) = \prod_{j=1}^m D_j^{\alpha_j} f(x) = \frac{\partial^{|\alpha|}}{\partial^{\alpha_1} x_1 \cdots \partial^{\alpha_m} x_m} f(x).$$

For a function $k \in \mathcal{C}^{2s}(\mathcal{X} \times \mathcal{X})$ with $\mathcal{X} \subset \mathbb{R}^d$ and $\alpha \in I_s^d$, we define

$$D^\alpha k(x,y) = D^{(\alpha,0)}k(x,y) = \frac{\partial^{|\alpha|}}{\partial^{\alpha_1} x_1 \cdots \partial^{\alpha_m} x_m} k(x,y). \tag{31}$$

and

$$D^{(0,\alpha)}k(x,y) = \frac{\partial^{|\alpha|}}{\partial^{\alpha_1} y_1 \cdots \partial^{\alpha_m} y_m} k(x,y).$$

**Theorem 3** (Theorem 1 in [56]). *Let $s \in \mathbb{N}$, $\mathcal{X} \subseteq \mathbb{R}^m$ and $k$ be a Mercer kernel such that $k \in \mathcal{C}^{2s}(\mathcal{X} \times \mathcal{X})$ with corresponding RKHS $\mathcal{H}$. Then the following hold:*

    *i. For any $x \in \mathcal{X}$, $\alpha \in I_s^m$,*

$$(D^\alpha k)_x(y) = D^\alpha k(x,y) \in \mathcal{H}. \tag{32}$$

    *ii. A partial derivative reproducing property holds for $\alpha \in I_s^m$*

$$(D^\alpha h)(x) = \langle (D^\alpha k)_x, h \rangle_{\mathcal{H}}, \quad \forall h \in \mathcal{H}. \tag{33}$$

Theorem 3 allows us to introduce the first and second order operators $D$ and $D^2$ that act on any feature map $\phi(x)$ as

$$D\phi(x) = ((D^{e_i}k)_x)_{i \in [d]} \quad \text{and} \quad D^2\phi(x) = ((D^{e_i+e_j}k)_x)_{i,j \in [d]}$$

where the $(D^{e_i}k)_x$ and $(D^{e_i+e_j}k)_x$ can be defined via (32). Then, we define the operator $d$ that maps any feature map $\phi(x)$ to $d\phi(x) = s(x)^\top D\phi(x)$. Denote $s^\top = [\bar{s}_1 | \ldots | \bar{s}_d] : x \in \mathcal{X} \mapsto s(x)^\top = [\bar{s}_1(x) | \ldots | \bar{s}_d(x)] \in \mathbb{R}^{p \times d}$. We have

$$s(x)^\top Dh(x) = \sum_{i=1}^d \bar{s}_i(x)\partial_i h(x) = \sum_{i=1}^d \bar{s}_i(x)\langle D^{e_i}\phi(x), h \rangle_{\mathcal{H}} = \langle d\phi(x), h \rangle_{\mathcal{H}},$$

so that we can define the embedding of the Dirichlet operator $B = s^\top \nabla : \mathcal{L}_\pi^2(\mathcal{X}) \to [\mathcal{L}_\pi^2(\mathcal{X})]^p$ into RKHS $d\phi : \mathcal{X} \to \mathcal{H}^p$ via the reproducing property as $\langle d\phi(x), h \rangle_{\mathcal{H}} = [BS_\pi h](x) = s(x)^\top Dh(x) \in \mathbb{R}^p$, $h \in \mathcal{H}$. In fact, $d\phi$ is a $p$-dimensional vector with components $d_k\phi : \mathcal{X} \to \mathcal{H}$ given via $\langle d_k\phi(x), h \rangle_{\mathcal{H}} = s_k(x)^\top Dh(x)$, $k \in [p]$. Then, we can define

$$T = -\mathbb{E}_{x \sim \pi}[d\phi(x) \otimes d\phi(x)] = -\sum_{k \in [p]} \mathbb{E}_{x \sim \pi}[d_k\phi(x) \otimes d_k\phi(x)]$$

which captures correlations between input and the outputs of the generator in the RKHS. Defining $w\phi : \mathcal{X} \to \mathcal{H}^{p+1}$ by $w\phi(x) = [\sqrt{\mu}\phi(x), d_1\phi(x), \ldots d_p\phi(x)]^\top \in \mathbb{R}^{d+1}$, we can consider

$$W_\mu = Z_\mu^* Z_\mu = S_\pi^*(\mu I - L)S_\pi$$
$$= \mathbb{E}_{x \sim \pi}[\mu\phi(x) \otimes \phi(x) + d\phi(x) \otimes d\phi(x)] = \mathbb{E}_{x \sim \pi}[w\phi(x) \otimes w\phi(x)]$$

which corresponds to the RKHS covariance operator w.r.t. energy space $\mathcal{W}$.

**Examples.** One way to ensure that the essential assumption (**(KE)**) holds is to show that $w\phi$ fulfils the boundedness condition (**(BK)**), i.e. $w\phi \in \mathcal{L}_\pi^\infty(\mathcal{X}, \mathcal{H}^{1+p})$. Let's show that the property holds true for the Damped Langevin (see Example 1) and the CIR process (see Example 2) if we consider the Radial Basis Function (RBF) kernel $k(x,y) = k_x(y) = \exp(-\kappa\|x-y\|^2)$ where $\kappa > 0$ is a free parameter that sets the "spread" of the kernel. As a reminder, for every $x \in \mathcal{X}$, we have

$$\|w\phi(x)\|^2 = \mu\|\phi(x)\|^2 + \sum_{k=1}^p \|d_k\phi(x)\|^2 \quad \text{with} \quad d_k = s_k(x)^\top D\phi(x).$$

Recalling that for the overdamped Langevin process $s(x) = (k_bT)^{1/2}(\delta_{ij})_{i \in [d], j \in [p]}$, $ss^\top$ is diagonal so that $\langle s(x)^\top Dk_x(\cdot), s(x)^\top Dk_x(\cdot) \rangle = 0$ for every $x \in \mathcal{X}$. As $\|\phi(x)\|^2 = 1$, we get that for every $x \in \mathcal{X}$, $\|w\phi(x)\|^2 \leq \mu =: c_{\mathcal{H}}$.

Consider now the CIR process. We have $d = p = 1$ and $s(x) = \sigma\sqrt{x}/\sqrt{2}$ for any $x \in \mathcal{X}$. For the very same reasons, $\langle s(x)^\top Dk_x(\cdot), s(x)^\top Dk_x(\cdot) \rangle = 0$ for every $x \in \mathcal{X}$, so that $\|w\phi(x)\|^2 \leq \mu =: c_{\mathcal{H}}$.

In both Langevin and CIR cases, we have $w\phi \in \mathcal{L}_\pi^\infty(\mathcal{X}, \mathcal{H}^{1+p})$ when considering an RBF kernel.

## C  Statistical learning framework

### C.1  Spectral perturbation bounds

In this section, we prove key perturbation result and discuss the properties of the metric distortion. We conclude this section with the approximation bound for arbitrary estimator $G \in \mathrm{B}_r(\mathcal{H})$ that is the basis of the statistical bounds that follow. This result is a direct consequence of [27] and Davis-Khan spectral perturbation result for compact self-adjoint operators, [11].

In the framework of Koopman operator learning [28], spectral bounds are expressed in terms of a distortion metric between the RKHS $\mathcal{H}$ and $\mathcal{L}^2_\pi(\mathcal{X})$, corresponding to the cost incurred from observing the operator's action on the $\mathcal{H}$ rather than on its domain $\mathcal{L}^2_\pi(\mathcal{X})$. Aligned with the risk definition (9), here we measure in a certain way the distortion between the $\mathcal{H}$ and $\mathcal{W}^\mu_\pi(\mathcal{X})$ as given in definitions in (10).

**Proposition 2.** *Let* $\widehat{G} = \sum_{i \in [r]} (\mu - \widehat{\lambda}_i)^{-1} \widehat{h}_i \otimes \widehat{g}_i$ *be the spectral decomposition of* $\widehat{G} \colon \mathcal{H} \to \mathcal{H}$, *where* $\widehat{\lambda}_i \geq \widehat{\lambda}_{i+1}$ *and let* $\widehat{f}_i = S_\pi \widehat{h}_i / \|S_\pi \widehat{h}_i\|_{\mathcal{L}^2_\pi}$, *for* $i \in [r]$. *Then for every* $\mu > 0$ *and* $i \in [r]$

$$\frac{|\lambda_i - \widehat{\lambda}_i|}{|\mu - \lambda_i||\mu - \widehat{\lambda}_i|} \leq \mathcal{E}(\widehat{G})\eta(\widehat{h}_i) \quad and \quad \|\widehat{f}_i - f_i\|^2_{\mathcal{L}^2_\pi} \leq \frac{2\,\mathcal{E}(\widehat{G})\eta(\widehat{h}_i)}{\mu\,[\mathrm{gap}_i - \mathcal{E}(\widehat{G})\eta(\widehat{h}_i)]_+}, \tag{11}$$

*where* $\mathrm{gap}_i$ *is the difference between $i$-th and $(i+1)$-th eigenvalue of* $(\mu I - L)^{-1}$.

*Proof.* We first remark that

$$\|((\mu I - L)^{-1} - (\mu - \widehat{\lambda}_i)^{-1} I_{\mathcal{L}^2_\pi(\mathcal{X})})^{-1}\|^{-1}_{\mathcal{H} \to \mathcal{W}} \leq \|((\mu I - L)^{-1} Z_\mu - Z_\mu \widehat{G})\widehat{h}_i\|_{\mathcal{H} \to \mathcal{W}}/\|Z_\mu \widehat{h}_i\| \leq \mathcal{E}(\widehat{G})\eta(\widehat{h}_i).$$

Then, from the first inequality, using that $(\mu I - L)^{-1}$ is self-adjoint as operator $\mathcal{W}^\mu_\pi(\mathcal{X}) \to \mathcal{W}^\mu_\pi(\mathcal{X})$, we obtain the first bound in (11).

So, observing that for every $(\mu - \lambda)^{-1} \in \mathrm{Sp}((\mu I - L)^{-1}) \setminus \{(\mu - \lambda_i)^{-1}\}$,

$$|(\mu - \widehat{\lambda}_i)^{-1} - (\mu - \lambda)^{-1}| \geq |(\mu - \lambda_i)^{-1} - (\mu - \lambda)^{-1}| - |(\mu - \widehat{\lambda}_i)^{-1} - (\mu - \lambda_i)^{-1}|$$

we conclude that $|(\mu - \widehat{\lambda}_i)^{-1} - (\mu - \lambda)^{-1}| \geq |(\mu - \lambda_i)^{-1} - (\mu - \lambda)^{-1}| - \mathcal{E}(\widehat{G})\,\eta(\widehat{h}_i)$, and

$$\min\{|(\mu - \widehat{\lambda}_i)^{-1} - (\mu - \lambda)^{-1}| \,|\, (\mu - \lambda)^{-1} \in \mathrm{Sp}((\mu I - L)^{-1}) \setminus \{(\mu - \lambda_i)^{-1}\}\} \geq \mathrm{gap}_i - \mathcal{E}(\widehat{G})\,\eta(\widehat{h}_i).$$

So, applying Proposition 3, we obtain

$$\sin(\sphericalangle(\widehat{f}_i, f_i)) \leq \frac{\|(\mu I - L)^{-1}\widehat{f}_i - (\mu - \widehat{\lambda}_i)^{-1}\,\widehat{f}_i\|}{[\mathrm{gap}_i - \mathcal{E}(\widehat{G})\,\eta(\widehat{h}_i)]_+} \leq \frac{\|((\mu I - L)^{-1}Z_\mu - Z_\mu \widehat{G})\widehat{h}_i\|/\|Z_\mu \widehat{h}_i\|}{[\mathrm{gap}_i - \mathcal{E}(\widehat{G})\,\eta(\widehat{h}_i)]_+}$$

$$\leq \frac{\mathcal{E}(\widehat{G})\,\eta(\widehat{h}_i)}{[\mathrm{gap}_i - \mathcal{E}(\widehat{G})\,\eta(\widehat{h}_i)]_+}.$$

Since, clearly $\|\widehat{f}_i - f_i\|^2 \leq 2(1 - \cos(\sphericalangle(\widehat{f}_i, f_i)) \leq 2\sin(\sphericalangle(\widehat{f}_i, f_i))$, the proof of the second bound is completed. $\square$

Next, we adapt the [28, Proposition 1] to our setting as follows.

**Proposition 5.** *Let* $\widehat{G} \in \mathrm{B}_r(\mathcal{H})$. *For all* $i \in [r]$ *the metric distortion of* $\widehat{h}_i$ *w.r.t. energy space* $\mathcal{W}^\mu_\pi(\mathcal{X})$ *can be tightly bounded as*

$$1 / \sqrt{\|W_\mu\|} \leq \eta(\widehat{h}_i) \leq \|\widehat{G}\| / \sigma^+_{\min}(Z_\mu \widehat{G}). \tag{34}$$

## D  Empirical estimation

The *Reduced Rank Regression* (RRR) estimator is the exact minimizer of (17) under fixed rank constraint. Specifically, RRR is the minimizer $\widehat{G}^r_{\mu,\gamma}$ of $\widehat{\mathcal{R}}_\gamma(G)$ within the set of bounded operators

$\mathrm{HS}_r(\mathcal{H})$ on $\mathcal{H}$ that have rank at most $r$. The *regularization* term $\gamma\|G\|_{\mathrm{HS}}^2$ is added to ensure stability. The closed form solution of the empirical RRR estimator is

$$\widehat{G}_{\mu,\gamma}^r = \widehat{H}_\gamma^{-1/2}[\![\widehat{H}_\gamma^{1/2}C]\!]_r, \tag{35}$$

while its population counterpart is given by $G_{\mu,\gamma}^r = W_{\mu,\gamma}^{-1/2}[\![W_{\mu,\gamma}^{1/2}C]\!]_r$.

In order to prove Theorem 1, recall kernel matrices in (19) and define

$$\mathbf{F}_\mu = \begin{bmatrix} \mu\mathbf{K} & \sqrt{\mu}\mathbf{N} \\ \sqrt{\mu}\mathbf{N}^\top & \mathbf{M} \end{bmatrix} \quad \text{and } \mathbf{F}_{\mu,\gamma} = \begin{bmatrix} \mu\mathbf{K}_\gamma & \sqrt{\mu}\mathbf{N} \\ \sqrt{\mu}\mathbf{N}^\top & \mathbf{M} + \gamma\mu I \end{bmatrix}. \tag{36}$$

Now we provide the explicit forms of the matrices N and M in the case of Langevin (see Example 1) and CIR (see Example 2) processes, considering an RBF kernel $k(x,y) = k_x(y) = \exp(-\kappa\|x-y\|^2)$. As a reminder (see (36)), $\mathbf{N} \in \mathbb{R}^{n \times pn}$ and $\mathbf{M} \in \mathbb{R}^{pn \times pn}$ are Gram matrices whose elements, for $k \in [1+p], i,j \in [n]$ are given by

$$\mathbf{N}_{i,(k-1)n+j} = n^{-1}\langle \phi(x_i), d_k\phi(x_j)\rangle_{\mathcal{H}} \text{ and } \mathbf{M}_{(k-1)n+i,(\ell-1)n+j} = n^{-1}\langle d_k\phi(x_i), d_\ell\phi(x_j)\rangle_{\mathcal{H}},$$

where $d_k\phi(x_j) = s_k(x_j)^\top D\phi(x_j)$ and $D\phi(x_j) = Dk(x_j,\cdot)$ is defined by (31). For $k \in [p], i,j \in [n]$, we have

$$\begin{aligned}
\langle\phi(x_i), d_k\phi(x_j)\rangle_{\mathcal{H}} &= \langle k_{x_i}, s_k(x_j)^\top Dk_{x_j}\rangle_{\mathcal{H}} \\
&= \left\langle k_{x_i}, s_k(x_j)^\top\left(\lim_{h\to 0}\frac{k(\cdot, x_j + he_l) - k(\cdot, x_j)}{h}\right)_{l\in[d]}\right\rangle \\
&= \lim_{h\to 0}\left\langle k_{x_i}, s_k(x_j)^\top\left(\frac{k(\cdot, x_j + he_l) - k(\cdot, x_j)}{h}\right)_{l\in[d]}\right\rangle \\
&= s_k(x_j)^\top\left(\lim_{h\to 0}\frac{k(x_i, x_j + he_l) - k(x_i, x_j)}{h}\right)_{l\in[d]} \\
&= s_k(x_j)^\top\left(D^{(0,e_l)}k(x_i, x_j)\right)_{l\in[d]} \\
&= 2\gamma s_k(x_j)^\top\left((x_i^{(l)} - x_j^{(l)})k(x_i, x_j)\right)_{l\in[d]}
\end{aligned}$$

where we have used the continuity of the inner product to get the third line and the reproducing property to obtain the following one. Similarly, for $k,\ell \in [p], j \in [n]$, we get

$$\begin{aligned}
\langle d_k\phi(x_i), d_\ell\phi(x_j)\rangle_{\mathcal{H}} &= \left\langle s_k(x_i)^\top\left(D^{(e_{l'},0)}k(x_i,\cdot)\right)_{l'\in[d]}, s_\ell(x_j)^\top\left(\lim_{h\to 0}\frac{k(\cdot, x_j + he_l) - k(\cdot, x_j)}{h}\right)_{l\in[d]}\right\rangle \\
&= \lim_{h\to 0}\left\langle s_k(x_i)^\top\left(D^{(e_{l'},0)}k(x_i,\cdot)\right)_{l'\in[d]}, s_\ell(x_j)^\top\left(\frac{k(\cdot, x_j + he_l) - k(\cdot, x_j)}{h}\right)_{l\in[d]}\right\rangle \\
&= \lim_{h\to 0}\left\langle\left(D^{(e_{l'},0)}k(x_i,\cdot)\right)_{l'\in[d]}, s_k(x_j)s_\ell(x_j)^\top\left(\frac{k(\cdot, x_j + he_l) - k(\cdot, x_j)}{h}\right)_{l\in[d]}\right\rangle \\
&= s_k(x_i)s_\ell(x_j)^\top\left(\lim_{h\to 0}\frac{D^{(e_{l'},0)}k(x_i, x_j + he_l) - D^{(e_{l'},0)}k(x_i, x_j)}{h}\right)_{l',l\in[d]} \\
&= s_k(x_i)s_\ell(x_j)^\top\left(\mathfrak{D}_{l',l}k(x_i, x_j)\right)_{l',l\in[d]},
\end{aligned}$$

where we have used the partial derivative reproducing (32) property and where we define for $l' \neq l$,

$$\begin{aligned}
\mathfrak{D}_{l',l}k(x_i, x_j) &= \lim_{h\to 0}\frac{D^{(e_{l'},0)}k(x_i, x_j + he_l) - D^{(e_{l'},0)}k(x_i, x_j)}{h} \\
&= -2\gamma\lim_{h\to 0}\frac{(x_i^{(l')} - x_j^{(l')})k(x_i, x_j + he_l) - (x_i^{(l')} - x_j^{(l')})k(x_i, x_j)}{h} \\
&= -4\gamma^2(x_i^{(l')} - x_j^{(l')})(x_i^{(l)} - x_j^{(l)})k(x_i, x_j), \tag{37}
\end{aligned}$$

and for $l \in [d]$,

$$\begin{aligned}
\mathfrak{D}_{l,l}k(x_i, x_j) &= \lim_{h\to 0}\frac{D^{(e_l,0)}k(x_i, x_j + he_l) - D^{(e_l,0)}k(x_i, x_j)}{h} \\
&= -2\gamma\lim_{h\to 0}\frac{(x_i^{(l)} - x_j^{(l)} - h)k(x_i, x_j + he_l) - (x_i^{(l)} - x_j^{(l)})k(x_i, x_j)}{h} \\
&= \left[2\gamma - 4\gamma^2(x_i^{(l)} - x_j^{(l)})^2\right]k(x_i, x_j) = 2\gamma[1 - 2\gamma(x_i^{(l)} - x_j^{(l)})^2]k(x_i, x_j). \tag{38}
\end{aligned}$$

For the overdamped Langevin (see Example 1), for $k \leq d$, $s_k(x_i) = (k_bT)^{1/2}e_k$ and $s_k(x_i)s_\ell(x_j)^\top = (k_bT)I$ so that

$$\mathrm{N}_{i,(k-1)n+j} = n^{-1}\langle\phi(x_i), d_k\phi(x_j)\rangle_{\mathcal{H}} = 2\gamma(k_bT)^{1/2}n^{-1}(x_i^{(k)} - x_j^{(k)})k(x_i, x_j)$$

and

$$\mathrm{M}_{(k-1)n+i,(\ell-1)n+j} = n^{-1}\langle d_k\phi(x_j), d_\ell\phi(x_j)\rangle_{\mathcal{H}} = (k_bT)n^{-1}(\mathfrak{D}_{l',l}k(x_j, x_j))_{l',l\in[d]},$$

where the elements of $\mathfrak{D}$ are given by (37) and (38). For the CIR process (see Example 2) in dimension $d = 1$, we have $s(x) = \sigma\sqrt{x}/\sqrt{2}$. Then,

$$\mathrm{N}_{i,(k-1)n+j} = n^{-1}\langle\phi(x_i), d_k\phi(x_j)\rangle_{\mathcal{H}} = \sqrt{2}\sigma\gamma n^{-1}\sqrt{x_j}(x_i - x_j)\exp(-\gamma|x_i - x_j|^2)$$

and

$$\mathrm{M}_{(k-1)n+i,(\ell-1)n+j} = n^{-1}\langle d_k\phi(x_i), d_\ell\phi(x_j)\rangle_{\mathcal{H}} = \sigma^2\gamma n^{-1}\sqrt{x_i}\sqrt{x_j}\exp(-\gamma|x_i - x_j|^2).$$

Based on the previous formulas, using Theorem 1, which we prove next, one can estimate generator's eigenpairs in practice.

**Theorem 1.** *Given $\mu > 0$ and $\gamma > 0$, let $\mathrm{J}_{\mu,\gamma} = \mathrm{K} - \mathrm{N}(\mathrm{M}+\gamma\mu I)^{-1}\mathrm{N}^\top + \gamma I$. Let $(\widehat{\sigma}_i^2, v_i)_{i\in[r]}$ be the leading eigenpairs of the following generalized eigenvalue problem*

$$\mu^{-1}(\mathrm{J}_{\mu,\gamma} - \gamma I)\mathrm{K}v_i = \widehat{\sigma}_i^2\mathrm{J}_{\mu,\gamma}v_i, \quad v_i^\top\mathrm{K}v_j = \delta_{ij}, \ i,j \in [r]. \tag{20}$$

*Denoting $\mathrm{V}_r = [v_1|\ldots|v_r] \in \mathbb{R}^{n\times r}$ and $\Sigma_r = \mathrm{diag}(\widehat{\sigma}_1,\ldots,\widehat{\sigma}_r)$, if $(\nu_i, w_i^\ell, w_i^r)_{i\in[r]}$ are eigentriplets of matrix $\mathrm{V}_r^\top\mathrm{V}_r\Sigma_r^2 \in \mathbb{R}^{r\times r}$, then the eigenvalue decomposition the RRR estimator $\widehat{G}_{\mu,\gamma}^r = \widehat{Z}_\mu\mathrm{U}_r\mathrm{V}_t^\top\widehat{S}$ is given by $\widehat{G}_{\mu,\gamma}^r = \sum_{i\in[r]}(\mu - \widehat{\lambda}_i)^{-1}\widehat{h}_i \otimes \widehat{g}_i$, where $\widehat{\lambda}_i = \mu - 1/\nu_i$, $\widehat{g}_i = \nu_i^{-1/2}\widehat{S}^*\mathrm{V}_rw_i^\ell$ and $\widehat{h}_i = \widehat{Z}_\mu^*\mathrm{U}_rw_i^r$ for $\mathrm{U}_r = (\mu\gamma)^{-1}[\mu^{-1/2}I \mid -\mathrm{N}(\mathrm{M}+\gamma\mu I)^{-1}]^\top(\mathrm{K}\mathrm{V}_r - \mu\mathrm{V}_r\Sigma_r^2) \in \mathbb{R}^{(1+p)n\times r}$.*

*Proof.* First, note that $\mu\mathrm{J}_{\mu,\gamma} \succ 0$ is exactly Schurs's complement w.r.t. second diagonal block of $\mathrm{F}_{\mu,\gamma} \succ 0$, and that, due to block inversion lemma [18], we have that

$$\mathrm{F}_{\mu,\gamma}^{-1} = \begin{bmatrix} \mu^{-1}\mathrm{J}_{\mu,\gamma} & \mu^{-1/2}\mathrm{J}_{\mu,\gamma}^{-1}\mathrm{N}(\mathrm{M}+\gamma\mu I)^{-1} \\ \mu^{-1/2}(\mathrm{M}+\gamma\mu I)^{-1}\mathrm{N}^\top\mathrm{J}_{\mu,\gamma}^{-1} & \mathrm{A} \end{bmatrix}. \tag{39}$$

where $\mathrm{A}$ is some $np \times np$ matrix. The first step in computing the RRR estimator lies in computing a truncating SVD of $\widehat{W}_{\mu,\gamma}^{-1/2}\widehat{C}$, that is $\widehat{C}\widehat{W}_{\mu,\gamma}^{-1}\widehat{C}q_i = \widehat{\sigma}_i^2q_i$, $i \in [r]$. Now, using the low-rank eigenvalue problem formulation [18], we have that $q_i = \widehat{S}^*v_i$ and $\widehat{S}\widehat{W}_{\mu,\gamma}^{-1}\widehat{C}\widehat{S}^*v_i = \widehat{\sigma}_i^2v_i$. Now, recalling that $\widehat{S} = [\mu^{-1/2}\mid 0]\widehat{Z}_\mu$ we obtain and that $\widehat{Z}_\mu(\widehat{Z}_\mu^*\widehat{Z}_\mu + \mu\gamma I)^{-1} = (\widehat{Z}_\mu\widehat{Z}_\mu^* + \mu\gamma I)^{-1}\widehat{Z}_\mu$, we obtain

$$[\mu^{-1/2}I \mid 0]\mathrm{F}_{\mu,\gamma}^{-1}\mathrm{F}_\mu[\mu^{-1/2}I \mid 0]^\top\mathrm{K}v_i = \widehat{\sigma}_i^2v_i,$$

which after some algebra, using (39)

$$\mu^{-1}(I - \gamma\mathrm{J}_{\mu,\gamma}^{-1})\mathrm{K}v_i = \widehat{\sigma}_i^2v_i,$$

i.e. $\mu^{-1}(I - \gamma\mathrm{J}_{\mu,\gamma}^{-1})\mathrm{K}\mathrm{V}_r = \mathrm{V}_r\Sigma_r^2$.

By normalizing right singular value functions $q_i$ of $\widehat{W}_{\mu,\gamma}^{-1/2}\widehat{C}$, that is by asking that $\langle q_i, q_i\rangle_{\mathcal{H}} = v_i^\top\mathrm{K}v_i = 1$, we obtain that $[\![\widehat{W}_{\mu,\gamma}^{-1/2}\widehat{C}]\!]_r = \widehat{W}_{\mu,\gamma}^{-1/2}\widehat{C}Q_rQ_r^*$, for $Q_r = [q_1|\ldots|q_r]$. In other words, we have

$$\widehat{G}_{\mu,\gamma} = \widehat{W}_{\mu,\gamma}^{-1}\widehat{Z}_\mu^*[\mu^{-1/2}I \mid 0]^\top\widehat{S}\widehat{S}^*\mathrm{V}_r\mathrm{V}_r^\top\widehat{S} = \widehat{Z}_\mu^*\mathrm{F}_{\mu,\gamma}^{-1}[\mu^{-1/2}I \mid 0]^\top\mathrm{K}\mathrm{V}_r\mathrm{V}_r^\top\widehat{S} = \widehat{Z}_\mu^*\mathrm{U}_r\mathrm{V}_r^\top\widehat{S},$$

where we used that $\mathrm{J}_{\mu,\gamma}^{-1}\mathrm{K}\mathrm{V}_r = \gamma^{-1}[\mathrm{K}\mathrm{V}_r - \mu\mathrm{V}_r\Sigma_r^2]$. Once with this form, we apply [27, Theorem 2] to obtain the result. $\qquad\square$

Finally, next result provides the reasoning for using empirical metric distortion of Theorem 23.

**Proposition 6.** *Under the assumptions of Theorem 1, for every $i \in [r]$*

$$\widehat{\eta}_i = \frac{\|\widehat{h}_i\|}{\|\widehat{Z}_\mu \widehat{h}_i\|} = \sqrt{\frac{(w_i^r)^* U_r^\top \mathbf{F}_\mu U_r w_i^r}{\|\mathbf{F}_\mu U_r w_i^r\|^2}}, \tag{40}$$

*and*

$$\left| \widehat{\eta}_i - \eta(\widehat{h}_i) \right| \leq \left( \eta(\widehat{h}_i) \wedge \widehat{\eta}_i \right) \eta(\widehat{h}_i) \, \widehat{\eta}_i \, \|\widehat{W}_\mu - W_\mu\|. \tag{41}$$

*Proof.* First, note that (40) follows directly from Theorem 1. Next, since for every $i \in [r]$,

$$(\widehat{\eta}_i)^{-2} - (\eta(\widehat{h}_i))^{-2} = \frac{\langle \widehat{h}_i, (\widehat{W}_\mu - W_\mu) \widehat{h}_i \rangle}{\|\widehat{h}_i\|^2} \leq \|\widehat{W}_\mu - W_\mu\|,$$

we obtain

$$\left| \widehat{\eta}_i^{-1} - (\eta(\widehat{h}_i))^{-1} \right| \leq \frac{\left| \widehat{\eta}_i^{-2} - (\eta(\widehat{h}_i))^{-2} \right|}{(\eta(\widehat{h}_i))^{-1} \vee \widehat{\eta}_i^{-1}} \leq \left( \eta(\widehat{h}_i) \wedge \widehat{\eta}_i \right) \|\widehat{W}_\mu - W_\mu\|.$$

$\square$

# E  Learning bounds

## E.1  Main assumptions

Recalling the form of IG in Equation 3, the Dirichlet form in **(DF)** for a self-adjoint $L$ exists whenever the positive definite diffusion part satisfies uniform ellipticity conditions and the drift term allows integration by parts, leading to $s(x) = b(x)/\sqrt{2}$. Thus, to obtain the partial knowledge we need, it is enough to estimate the inexact Dirichlet coefficient $s_\epsilon$ that satisfies the relative error bound in **(DF)**. When the sample paths are observed continuously, such $s_\epsilon$ can be directly identified from these observations, making it a non-statistical problem. For discrete realizations of the process, *the diffusion coefficient can be estimated non-parametrically* using various methods, such as pathwise estimation by computing the variance of the increments over small intervals [20], kernel-based methods [15], local polynomial regression [13]. Remark, however, that *the estimation of the drift (needed for the full knowledge) is a much more demanding task.* Different methods are reviewed in [30] and references therein. A more recent approach [9] drawing inspiration from particle systems, consists in constructing estimates from several i.i.d. paths of the solution process.

Next, observe that **(BK)** implies $Z_\mu \in \mathrm{HS}\,(\mathcal{H}, \mathcal{W}_\pi^\mu(\mathcal{X}))$, which according to the spectral theorem for positive self-adjoint operators, has an SVD, i.e. there exists at most countable positive sequence $(\sigma_j)_{j \in J}$, where $J = \{1, 2, \ldots, \} \subseteq \mathbb{N}$, and ortho-normal systems $(z_j)_{j \in J}$ and $(h_j)_{j \in J}$ of $\mathrm{cl}(\mathrm{Im}(Z_\mu))$ and $\mathrm{Ker}(Z_\mu)^\perp$, respectively, such that $Z_\mu h_j = \sigma_j z_j$ and $Z_\mu^* z_j = \sigma_j h_j$, $j \in J$.

Now, given $\alpha \geq 0$, let us define scaled injection operator $Z_{\mu,\alpha} \colon \mathcal{H} \to \mathcal{W}_\pi^\mu(\mathcal{X})$ as

$$Z_{\mu,\alpha} = \sum_{j \in J} \sigma_j^\alpha z_j \otimes h_j. \tag{42}$$

Clearly, we have that $Z_\mu = Z_{\mu,1}$, while $\mathrm{Im}\, Z_{\mu,0} = \mathrm{cl}(\mathrm{Im}(Z_\mu))$. Next, we equip $\mathrm{Im}(Z_{\mu,\alpha})$ with a norm $\|\cdot\|_{\mathcal{H},\alpha}$ to build an interpolation space

$$[\mathcal{H}]_\alpha = \left\{ f \in \mathrm{Im}(Z_{\mu,\alpha}) \mid \|f\|_{\mathcal{H},\alpha}^2 = \sum_{j \in J} \sigma_j^{-2\alpha} \langle f, z_j \rangle_{\mathcal{W}}^2 < \infty \right\},$$

noting that the inner product in $\mathcal{W}_\pi^\mu(\mathcal{X})$ is given by bilinear energy functional

$$\langle f, g \rangle_{\mathcal{W}} = \mu \langle f, g \rangle_{\mathcal{L}_\pi^2} - \langle f, Lg \rangle_{\mathcal{L}_\pi^2}.$$

We remark that for $\alpha = 1$ the space $[\mathcal{H}]_\alpha$ is just an RKHS $\mathcal{H}$ seen as a subspace of $\mathcal{W}_\pi^\mu(\mathcal{X})$. Moreover, we have the following injections

$$[\mathcal{H}]_{\alpha_1} \hookrightarrow [\mathcal{H}]_1 \hookrightarrow [\mathcal{H}]_{\alpha_2} \hookrightarrow [\mathcal{H}]_0 = \mathcal{W}_\pi^\mu(\mathcal{X}),$$

where $\alpha_1 \geq 1 \geq \alpha_2 \geq 0$.

In addition, from **(BK)** we also have that RKHS $\mathcal{H}$ can be embedded into
$$W_\pi^{\mu,\infty}(\mathcal{X}) = \{f \in \mathcal{W}_\pi^\mu(\mathcal{X}) \mid \|f\|_{W_\pi^{\mu,\infty}} = \operatorname*{ess\,sup}_{x \sim \pi}[|f(x)|^2 - f(x)[Lf](x)] < \infty\}$$
that is, for some $\tau \in (0,1]$
$$[\mathcal{H}]_1 \hookrightarrow [\mathcal{H}]_\tau \hookrightarrow W_\pi^{\mu,\infty}(\mathcal{X}) \hookrightarrow \mathcal{W}_\pi^\mu(\mathcal{X}).$$
Now, according to [14], if $Z_{\mu,\tau,\infty} : [\mathcal{H}]_\tau \hookrightarrow L_\pi^\infty(\mathcal{X})$ denotes the injection operator, its boundedness implies the polynomial decay of the singular values of $Z_\mu$, i.e. $\sigma_j^2(Z_\mu) \lesssim j^{-1/\tau}$, $j \in J$, and the condition **(KE)** is assured.

Assumption **(SD)** allows one to quantify the effective dimension of $\mathcal{H}$ in ambient space $\mathcal{W}_\pi^\mu(\mathcal{X})$, while the kernel embedding property **(KE)** allows one to estimate the norms of whitened feature maps, in our generator setting vector-valued since they define rank$(1+p)$ operators on $\mathcal{H}$,
$$\xi(x) := W_{\mu,\gamma}^{-1/2} w\phi(x) \in \mathcal{H}^{1+p}. \tag{43}$$
This object plays a key role in deriving the learning rates for regression problems (see [29]) and the following result is bounding it.

**Lemma 1.** *Let* **(KE)** *hold for some* $\tau \in [\beta,1]$ *and* $c_\tau \in (0,\infty)$. *Then,*

$$\mathbb{E}_{x \sim \pi}\|\xi(x)\|_{\mathcal{H}^{1+p}}^2 \leq \begin{cases} \frac{c_\beta^\beta}{1-\beta}(\mu\gamma)^{-\beta} & ,\beta < 1, \\ c_\tau(\mu\gamma)^{-1} & ,\beta = 1. \end{cases} \quad and \quad \|\xi\|_\infty^2 = \operatorname*{ess\,sup}_{x \sim \pi}\|\xi(x)\|_{\mathcal{H}^{1+p}}^2 \leq c_\tau(\mu\gamma)^{-\tau}.$$
$$\tag{44}$$

*Proof.* W.l.o.g. set $\mu = 1$, observing that the only change in the proof is in scaling $\gamma > 0$. We first observe that for every $j \in J$ from definition of $w\phi$ and fact that $h_j(x) = [Z_\mu h_j](x)$ $\pi$-a.e., it holds that
$$\sum_{i \in [1+p]} \langle w_i\phi(x), h_j \rangle^2 = \mu|h_j(x)|^2 - h_j(x)[LZ_\mu h_j](x) = \mu|[Z_\mu h_j](x)|^2 - [Z_\mu h_j](x)[LZ_\mu h_j](x), \ \pi\text{-a.e.},$$
implying that $\sum_{i \in [1+p]} \langle w_i\phi(x), h_j \rangle^2 \leq \sigma_j\mu|z_j(x)|^2 - z_j(x)[Lz_j](x)$. So, for every $\tau > 0$,

$$\|\xi(x)\|_{\mathcal{H}^{1+p}}^2 = \sum_{j \in J}\sum_{i \in [1+p]} \langle W_{\mu,\gamma}^{-1/2}w_i\phi(x), h_j \rangle^2 = \sum_{j \in J}\sum_{i \in [1+p]} \frac{1}{\sigma_j^2 + \gamma}\langle w_i\phi(x), h_j \rangle^2$$

$$= \sum_{j \in J}\sum_{i \in [1+p]} \frac{\sigma_j^{2(1-\tau)}}{\sigma_j^2 + \gamma}\frac{\langle w_i\phi(x), h_j \rangle^2}{\sigma_j^2}\sigma_j^{2\tau} = \gamma^{-\tau}\sum_{j \in J}\sum_{i \in [1+p]} \frac{(\sigma_j^2\gamma^{-1})^{1-\tau}}{\sigma_j^2\gamma^{-1} + 1}\frac{\langle w_i\phi(x), h_j \rangle^2}{\sigma_j^2}\sigma_j^{2\tau}$$

$$\leq \gamma^{-\tau}\sum_{j \in J}\frac{\mu|h_j(x)|^2 - h_j(x)[LZ_\mu h_j](x)}{\sigma_j^2}\sigma_j^{2\tau} = \gamma^{-\tau}\sum_{j \in J}(\mu|z_j(x)|^2 - z_j(x)[Lz_j](x))\sigma_j^{2\tau},$$

and, due to (21), we obtain $\|\xi\|_\infty^2 \leq \gamma^{-\tau}c_\tau$. On the other hand, we also have that
$$\operatorname{tr}(\mathbb{E}_{x \sim \pi}[\xi(x) \otimes \xi(x)]) = \operatorname{tr}(W_{\mu,\gamma}^{-1/2}W_\mu W_{\mu,\gamma}^{-1/2}) = \operatorname{tr}(W_{\mu,\gamma}^{-1}W_\mu),$$
which is an effective dimension of the RKHS $\mathcal{H}$ in $\mathcal{W}_\pi^\mu(\mathcal{X})$. Therefore, following the proof of Fischer and Steinwart [14, Lemma 11] for classical covariances in $\mathcal{L}_\pi^2$, we show that the bound on the effective dimension is

$$\operatorname{tr}(W_{\mu,\gamma}^{-1}W_\mu) = \sum_{j \in N}\frac{\sigma_j^2}{\sigma_j^2 + \gamma} \leq \begin{cases} \frac{c_\beta^\beta}{1-\beta}\gamma^{-\beta} & ,\beta < 1, \\ c_\tau\gamma^{-1} & ,\beta = 1. \end{cases} \tag{45}$$

For the case $\beta = 1$, it suffices to see that
$$\|z_j\|_{\mathcal{W}} = \mathfrak{E}_\mu[z_j] \leq \|z_j\|_{W_\pi^{\mu,\infty}} = \operatorname*{ess\,sup}_{x \sim \pi}[|z_j(x)|^2 - z_j(x)[LZ_\mu z_j](x)],$$
and, hence
$$\operatorname{tr}(W_{\mu,\gamma}^{-1}W_\mu) \leq \gamma^{-1}\sum_{j \in N}\sigma_j^2\|z_j\|_{\mathcal{W}}^2 = \gamma^{-1}\sum_{j \in N}\sigma_j^2\mathfrak{E}_\mu[z_j] \leq \gamma^{-1}c_\tau.$$
For $\beta < 1$ we can apply the same classical reasoning as in the proof of Proposition 3 of [8]. $\qquad\square$

**Proposition 7.** *If the eigenfunctions of $L$ belong to $[\mathcal{H}]_\alpha$, then Condition (RC) is satisfied.*

*Proof.* Note first that the resolvent $(\mu I - L)^{-1}$ admits the same eigenfunctions as the generator $L$, meaning that $\mathrm{Im}((\mu I - L)^{-1} Z_\mu) \subseteq \mathrm{cl}(\mathrm{Im}(Z_{\mu,\alpha}))$. But according to [54, Theorem 2.2], this last condition is equivalent to Condition (RC). $\qquad\square$

Finally we prove here Proposition 8.

**Proposition 8.** *Given $\mu > 0$, let $\mathcal{H} \subseteq \mathcal{W}_\pi^\mu(\mathcal{X})$ be the RKHS associated to kernel $k \in \mathcal{C}^2(\mathcal{X} \times \mathcal{X})$ such that $Z_\mu \in \mathrm{HS}\,(\mathcal{H}, \mathcal{W}_\pi^\mu(\mathcal{X}))$, and let $P_\mathcal{H}$ be the orthogonal projector onto the closure of $\mathrm{Im}(Z_\mu) \subseteq \mathcal{W}_\pi^\mu(\mathcal{X})$. Then, for every $\varepsilon > 0$, there exists a finite rank operator $G\colon \mathcal{H} \to \mathcal{H}$ such that $\mathcal{R}(G) \le \|(I - P_\mathcal{H})(\mu I - L)^{-1} Z_\mu\|_{\mathrm{HS}(\mathcal{H},\mathcal{W})}^2 + \varepsilon$. Consequently, when $k$ is universal, $\mathcal{R}(G) \le \varepsilon$.*

*Proof.* Recall first that since $Z_\mu \in \mathrm{HS}\,(\mathcal{H}, \mathcal{W}_\pi^\mu(\mathcal{X}))$, according to the spectral theorem for positive self-adjoint operators, $Z_\mu$ admits an SVD. Its form is provided in (42) taking $\alpha = 1$.

Now, recalling that $[\![\cdot]\!]_r$ denotes the $r$-truncated SVD, i.e. $[\![Z_\mu]\!]_r = \sum_{j\in[r]} \sigma_j z_j \otimes h_j$, since $\|Z_\mu - [\![Z_\mu]\!]_r\|_{\mathrm{HS}}^2 = \sum_{j>r} \sigma_j^2$, for every $\delta > 0$ there exists $r \in \mathbb{N}$ such that $\|Z_\mu - [\![Z_\mu]\!]_r\|_{\mathrm{HS}} < \mu\delta/3$. Consequently since all the eigenvalues of $L$ are non-positive, $\|(\mu I - L)^{-1}(Z_\mu - [\![Z_\mu]\!]_r)\|_{\mathrm{HS}} \le \|Z_\mu - [\![Z_\mu]\!]_r\|_{\mathrm{HS}}/\mu \le \delta/3$. Next since $\mathrm{Im}(P_\mathcal{H}(\mu I - L)^{-1} Z_\mu) \subseteq \mathrm{cl}(\mathrm{Im}(Z_\mu))$, for any $j \in [r]$, there exists $g_j \in \mathcal{H}$ s.t. $\|P_\mathcal{H}(\mu I - L)^{-1} z_j - Z_\mu g_j\| \le \frac{\delta}{3r}$, and, denoting $B_r := \sum_{j\in[r]} \sigma_j g_j \otimes h_j$ we conclude $\|P_\mathcal{H}(\mu I - L)^{-1}[\![Z_\mu]\!]_r - Z_\mu B_r\|_{\mathrm{HS}} \le \delta/3$. Finally we recall that the set of non-defective matrices is dense in the space of matrices [47], implying that the set of non-defective rank-$r$ linear operators is dense in the space of rank-$r$ linear operators on a Hilbert space. Therefore, there exists a non-defective $G \in \mathrm{B}_r(\mathcal{H})$ such that $\|G - B_r\|_{\mathrm{HS}} < \delta/(3\sigma_1(Z_\mu))$. So, we conclude

$$
\begin{aligned}
\|(\mu I - &L)^{-1} Z_\mu - Z_\mu G\|_{\mathrm{HS}} \\
&\le \|(I - P_\mathcal{H})(\mu I - L)^{-1} Z_\mu\|_{\mathrm{HS}} + \|(\mu I - L)^{-1} Z_\mu - [\![(\mu I - L)^{-1} Z_\mu]\!]_r\|_{\mathrm{HS}} \\
&\quad + \|[\![(\mu I - L)^{-1} Z_\mu]\!]_r - Z_\mu B_r\|_{\mathrm{HS}} + \|Z_\mu(G - B_r)\|_{\mathrm{HS}} \\
&\le \|(I - P_\mathcal{H})(\mu I - L)^{-1} Z_\mu\|_{\mathrm{HS}} + \delta.
\end{aligned}
$$

$\qquad\square$

**Example 5.** *These three conditions depend on the process $X$ (through its generator $L$) as well as the chosen RKHS. They are satisfied, for example, by choosing for $k$ as a Gaussian kernel. Indeed, the sub-linearity conditions on $A$ and $B$ required to ensure the existence and uniqueness of the solution of the process of* (1)*, also ensure that $A$ and $B$ are sufficiently 'nice' to fulfil, notably, condition ((BK)).*

### E.2 Bounding the Bias

Recalling the error decomposition and passing to the $\mathcal{H}$ and $\mathcal{L}_\pi^2$-norms, we have that

$$
\mathcal{E}(\widehat{G}) \le \underbrace{\|(\mu I - L)^{-1} Z_\mu - Z_\mu G_{\mu,\gamma}\|_{\mathcal{H}\to\mathcal{W}}}_{\text{regularization bias}} + \underbrace{\|W_\mu^{1/2}(G_{\mu,\gamma} - G_{\mu,\gamma}^r)\|}_{\text{rank reduction bias}} + \underbrace{\|W_\mu^{1/2}(G_{\mu,\gamma}^r - \widehat{G}_{\mu,\gamma}^r)\|}_{\text{estimator's variance}}, \quad (46)
$$

and continue to prove the bound of the first term. Note that, while this proof technique is standard for operator learning [29, 35], we present it here for the sake of completeness.

**Proposition 9.** *Let $G_{\mu,\gamma} = W_{\mu,\gamma}^{-1} C$ for $\gamma > 0$, and $P_\mathcal{H}\colon \mathcal{W}_\pi^\mu(\mathcal{X}) \to \mathcal{W}_\pi^\mu(\mathcal{X})$ be the orthogonal projector onto $\mathrm{cl}(\mathrm{Im}(Z_\mu))$. If the assumptions (BK), (SD) and (RC) hold, then $\|G_{\mu,\gamma}\| \le c_\alpha c_\mathcal{W}^{(\alpha-1)/2}$ for $\alpha \in [1,2]$, $\|G_{\mu,\gamma}\| \le c_\alpha\,(\mu\gamma)^{(\alpha-1)/2}$ for $\alpha \in (0,1]$, and*

$$
\|(\mu I - L)^{-1} Z_\mu - Z_\mu G_{\mu,\gamma}\|_{\mathcal{H}\to\mathcal{W}} \le c_\alpha\,(\mu\gamma)^{\frac{\alpha}{2}} + \|(I - P_\mathcal{H})(\mu I - L)^{-1} Z_\mu\|_{\mathcal{H}\to\mathcal{W}}. \quad (47)
$$

*Proof.* Recalling that $P_\mathcal{H} := \sum_{j\in J} z_j \otimes z_j$, start by denoting the orthogonal projectors on the subspace of $k$ leading left singular functions of $Z_\mu$ as $P_k := \sum_{j\in[k]} z_j \otimes z_j$, respectively. Next, observe

that $C=Z_\mu^*(\mu I-L)^{-1}Z_\mu$ and $Z_\mu^* W_{\mu,\gamma}^{-1}=Z_\mu^*(Z_\mu^* Z_\mu + \mu\gamma I)^{-1}=(Z_\mu Z_\mu^* + \mu\gamma I)^{-1}Z_\mu$. Therefore,

$$(\mu I-L)^{-1}Z_\mu - Z_\mu G_{\mu,\gamma}=(I-Z_\mu W_{\mu,\gamma}^{-1}Z_\mu^*)(\mu I-L)^{-1}Z_\mu$$
$$=(I-(Z_\mu Z_\mu^* + \mu\gamma I)^{-1}Z_\mu Z_\mu^*)(\mu I-L)^{-1}Z_\mu$$
$$=\mu\gamma(Z_\mu Z_\mu^* + \mu\gamma I)^{-1}(\mu I-L)^{-1}Z_\mu$$

and, hence

$$(\mu I-L)^{-1}Z_\mu - Z_\mu G_{\mu,\gamma}= \left(\sum_{j\in J}\frac{\mu\gamma}{\sigma_j^2 + \mu\gamma}z_j \otimes z_j\right)(\mu I-L)^{-1}Z_\mu$$
$$= \left(\sum_{j\in J}\frac{\mu\gamma}{(\sigma_j^2 + \mu\gamma)\sigma_j}z_j \otimes (Z_\mu h_j)\right)(\mu I-L)^{-1}Z_\mu$$
$$= \left(\sum_{j\in J}\frac{\mu\gamma}{(\sigma_j^2 + \mu\gamma)\sigma_j}z_j \otimes h_j\right)C.$$

Therefore, for every $k \in J$, $\|P_k((\mu I-L)^{-1}Z_\mu - Z_\mu G_{\mu,\gamma})\|_{\mathcal{H}\to\mathcal{W}}^2$ becomes

$$\left\|\left(\sum_{j\in[k]}\frac{\mu\gamma}{(\sigma_j^2+\mu\gamma)\sigma_j}z_j \otimes h_j\right)C^2\left(\sum_{j\in[k]}\frac{\gamma}{(\sigma_j^2+\mu\gamma)\sigma_j}h_j \otimes z_j\right)\right\|_{\mathcal{H}\to\mathcal{W}},$$

which, due to **(RC)**, implies that

$$\|P_k((\mu I-L)^{-1}Z_\mu - Z_\mu G_{\mu,\gamma})\|_{\mathcal{H}\to\mathcal{W}} \leq c_\alpha \left\|\sum_{j\in[k]}\frac{\mu\gamma\,\sigma_j^\alpha}{\sigma_j^2 + \mu\gamma}z_j \otimes z_j\right\|_{\mathcal{W}\to\mathcal{W}}.$$

On the other hand,

$$\sum_{j\in[k]}\frac{\mu\gamma\,\sigma_j^\alpha}{\sigma_j^2 + \mu\gamma}z_j \otimes z_j=\gamma^{\frac{\alpha}{2}}\sum_{j\in[k]}\frac{(\sigma_j^2(\mu\gamma)^{-1})^{\frac{\alpha}{2}}}{\sigma_j^2(\mu\gamma)^{-1} + 1}z_j \otimes z_j \preceq (\mu\gamma)^{\frac{\alpha}{2}}\sum_{j\in[k]}z_j \otimes z_j,$$

where the inequality holds due to $x^s \leq x + 1$ for all $x \geq 0$ and $s \in [0,1]$. Since the norm of the projector equals one, we get $\|P_k((\mu I-L)^{-1}Z_\mu - Z_\mu G_{\mu,\gamma})\| \leq c_\alpha(\mu\gamma)^{\frac{\alpha}{2}}$.

Next, observe that

$$\|(P_\mathcal{H}-P_k)((\mu I-L)^{-1}Z_\mu - Z_\mu G_{\mu,\gamma})\|_{\mathcal{H}\to\mathcal{W}}^2=\left\|\sum_{j\in J\setminus[k]}\frac{\mu^2\gamma^2}{(\sigma_j^2+\mu\gamma)^2}(Z_\mu^* z_j) \otimes (Z_\mu^* z_j)\right\|_{\mathcal{H}\to\mathcal{H}}$$

which is bounded by

$$\sum_{j\in J\setminus[k]}\frac{\mu^2\gamma^2}{(\sigma_j^2+\mu\gamma)^2}\|Z_\mu^* z_j\|^2 \leq \sum_{j\in J\setminus[k]}\frac{\mu^2\gamma^2\,\sigma_j^{2\alpha}}{(\sigma_j^2+\mu\gamma)^2} \leq \sum_{j\in J\setminus[k]}\sigma_j^{2\alpha}.$$

Using triangular inequality, for every $k \in J$, we have that $\|P_\mathcal{H}((\mu I-L)^{-1}Z_\mu - Z_\mu G_{\mu,\gamma})\|_{\mathcal{H}\to\mathcal{W}}$ is bounded by

$$\|P_k((\mu I-L)^{-1}Z_\mu - Z_\mu G_{\mu,\gamma})\|_{\mathcal{H}\to\mathcal{W}}+\|(P_\mathcal{H}-P_k)((\mu I-L)^{-1}Z_\mu - Z_\mu G_{\mu,\gamma})\|_{\mathcal{H}\to\mathcal{W}}$$

and, therefore,

$$\|P_\mathcal{H}((\mu I-L)^{-1}Z_\mu - Z_\mu G_{\mu,\gamma})\|_{\mathcal{H}\to\mathcal{W}} \leq c_\alpha(\mu\gamma)^{\frac{\alpha}{2}} + \sum_{j\in J\setminus[k]}(\sigma_j^{2\beta})^{\frac{\alpha}{\beta}},$$

and, hence, letting $k \to \infty$ we obtain $\|P_{\mathcal{H}}(\mu I - L)^{-1} Z_\mu - Z_\mu G_{\mu,\gamma}\| \leq c_\alpha (\mu\gamma)^{\frac{\alpha}{2}}$. Hence, (47) follows from triangular inequality.

To estimate $\|G_{\mu,\gamma}\|$, note that **(RC)** implies $\|G_{\mu,\gamma}\| \leq c_\alpha \|W_{\mu,\gamma}^{-1} W_\mu^{\frac{1+\alpha}{2}}\|$ and consider two cases. First, if **(RC)** holds for some $\alpha \in [1, 2]$, then, clearly $\|G_{\mu,\gamma}\| \leq c_\alpha c_{\mathcal{W}}^{(\alpha-1)/2}$. On the other hand, if $\alpha \in (0, 1]$, then

$$\frac{\sigma_j^{1+\alpha}}{\sigma_j^2 + \mu\gamma} = (\mu\gamma)^{-1} \frac{\left(\sigma_j^2 (\mu\gamma)^{-1}\right)^{\frac{1+\alpha}{2}}}{\sigma_j^2 (\mu\gamma)^{-1} + 1} \leq (\mu\gamma)^{\frac{\alpha-1}{2}},$$

and, thus, $\|G_{\mu,\gamma}\| \leq c_\alpha \gamma^{(\alpha-1)/2}$.

$\square$

**Remark 2.** *Inequality* (47) *says that the regularization bias is comprised of a term depending on the choice of $\gamma$, and on a term depending on the "alignment" between $\mathcal{H}$ and $\mathrm{Im}((\mu I - L)^{-1} Z_\mu)$. The term $\|(I - P_{\mathcal{H}})(\mu I - L)^{-1} Z_\mu\|$ can be set to zero by two different approaches. One option is to choose a kernel that, in some way, minimizes $\|(I - P_{\mathcal{H}})(\mu I - L)^{-1} Z_\mu\|$. Another is to choose a universal kernel [45, Chapter 4], for which $\mathrm{Im}((\mu I - L)^{-1} S_\pi) \subseteq \mathrm{cl}(\mathrm{Im}(S_\pi))$. While we here develop theory for universal kernels, deep learning approaches that leverage on our approach can be developed, which is the direction to pursue in future.*

In order to proceed with bounding the bias due to rank reduction for both considered estimators, we first provide auxiliary result.

**Proposition 10.** *Let $B := W_{\mu,\gamma}^{-1/2} C$, let* **(RC)** *hold for some $\alpha \in (0, 2]$, and for $j \in \mathbb{N}$ denote*

$$\lambda_j^\star = \sigma_j^2((\mu I - L)^{-1} Z_\mu) = \lambda_j(S_\pi^*(\mu I - L)^{-1} S_\pi). \tag{48}$$

*Then for every $j \in \mathbb{N}$,*

$$\lambda_j^\star - c_\alpha^2 c_{\mathcal{W}}^{\alpha/2} (\mu\gamma)^{\alpha/2} \leq \sigma_j^2(B) \leq \lambda_j^\star. \tag{49}$$

*Proof.* Start by observing that

$$B^* B = [(\mu I - L)^{-1} Z_\mu]^* Z_\mu W_{\mu,\gamma}^{-1} Z_\mu^* (\mu I - L)^{-1} Z_\mu$$
$$= [(\mu I - L)^{-1} Z_\mu]^* (\mu I - L)^{-1} Z_\mu - \mu\gamma [(\mu I - L)^{-1} Z_\mu]^* (Z_\mu Z_\mu^* + \mu\gamma I)^{-1} (\mu I - L)^{-1} Z_\mu,$$

implies that, using $[(\mu I - L)^{-1} Z_\mu]^* = S_\pi$ and $[(\mu I - L)^{-1} Z_\mu]^* (\mu I - L)^{-1} Z_\mu = S_\pi^*(\mu I - L)^{-1} S_\pi$,

$$S_\pi^*(\mu I - L)^{-1} S_\pi - \sum_{j \in J} \frac{\mu\gamma}{\sigma_j^2 + \mu\gamma} (S_\pi^* z_j) \otimes (S_\pi^* z_j) = B^* B \preceq Z_\mu^* Z_\mu.$$

Next, similarly to the above, for every $k \in J$, we have

$$\left\| \sum_{j \in [k]} \frac{\mu\gamma}{\sigma_j^2 + \mu\gamma} (S_\pi^* z_j) \otimes (S_\pi^* z_j) \right\| \leq c_\alpha^2 \left\| \sum_{j \in [k]} \frac{\sigma_j^{2\alpha}}{\sigma_j^2 (\mu\gamma)^{-1} + 1} z_j \otimes z_j \right\|$$

$$= c_\alpha^2 \left\| \sum_{j \in [k]} \frac{(\sigma_j^2 (\mu\gamma)^{-1})^{\alpha/2} \sigma_j^\alpha (\mu\gamma)^{\alpha/2}}{\sigma_j^2 (\mu\gamma)^{-1} + 1} z_j \otimes z_j \right\| \leq c_\alpha^2 \gamma^{\alpha/2} \|W_\mu\|^{\alpha/2},$$

and

$$\left\| \sum_{j \in J \setminus [k]} \frac{\gamma}{\sigma_j^2 + \mu\gamma} (S_\pi^* z_j) \otimes (S_\pi^* z_j) \right\| \leq c_\alpha^2 \sum_{j \in J \setminus [k]} \frac{\gamma}{\sigma_j^2 + \mu\gamma} \sigma_j^{2\alpha} \leq c_\alpha^2 \sum_{j \in J \setminus [k]} (\sigma_j^{2\beta})^{\alpha/\beta}.$$

So, as before, letting $k \to \infty$ we get the result. $\square$

As a consequence, we obtain the bound for the rank reduction bias of the RRR method.

**Proposition 11** (RRR)**.** *Let* **(RC)** *hold for some* $\alpha \in (0, 2]$*. Then the bias of* $G^r_{\mu,\gamma}$ *due to rank reduction, recalling* (48)*, is bounded as*

$$\sqrt{\lambda^\star_{r+1}} - c_\alpha \, c_{\mathcal{W}}^{\alpha/4} (\mu\gamma)^{\alpha/4} - 2 \, c_\alpha \, (\mu\gamma)^{(1 \wedge \alpha)/2} \leq \|Z_\mu(G_{\mu,\gamma} - G^r_{\mu,\gamma})\| \leq \sqrt{\lambda^\star_{r+1}}. \tag{50}$$

*Proof.* Observe that

$$\|Z_\mu(G_{\mu,\gamma} - G^r_{\mu,\gamma})\| \leq \|W^{1/2}_{\mu,\gamma}(G_{\mu,\gamma} - G^r_{\mu,\gamma})\| = \|B - [\![B]\!]_r\| = \sigma_{r+1}(B) \leq \sigma_{r+1}((\mu I - L)^{-1} Z_\mu)$$

while

$$\|Z_\mu(G_{\mu,\gamma} - G^r_{\mu,\gamma})\| \geq \|W^{1/2}_{\mu,\gamma}(G_{\mu,\gamma} - G^r_{\mu,\gamma})\| - (\mu\gamma)^{1/2}\|G_{\mu,\gamma} - G^r_{\mu,\gamma}\|$$
$$\geq \sigma_{r+1}((\mu I - L)^{-1} Z_\mu) - c_\alpha \|W_\mu\|^{\alpha/4}(\mu\gamma)^{\alpha/4} - 2c_\alpha(\mu\gamma)^{(1 \wedge \alpha)/2}.$$

$\square$

### E.3  Bounding the Variance

#### E.3.1  Concentration Inequalities

All the statistical bounds we present will relay on two versions of Bernstein inequality. The first one is Pinelis and Sakhanenko inequality for random variables in a separable Hilbert space, see [8, Proposition 2].

**Proposition 12.** *Let* $A_i$*,* $i \in [n]$ *be i.i.d copies of a random variable* $A$ *in a separable Hilbert space with norm* $\|\cdot\|$*. If there exist constants* $\Lambda > 0$ *and* $\sigma > 0$ *such that for every* $m \geq 2$ $\mathbb{E}\|A\|^m \leq \frac{1}{2} m! \Lambda^{m-2} \sigma^2$*, then with probability at least* $1 - \delta$*,*

$$\left\| \frac{1}{n} \sum_{i \in [n]} A_i - \mathbb{E}A \right\| \leq \frac{4\sqrt{2}}{\sqrt{n}} \log \frac{2}{\delta} \sqrt{\sigma^2 + \frac{\Lambda^2}{n}}. \tag{51}$$

On the other hand, we recall that in [39], a dimension-free version of the non-commutative Bernstein inequality for finite-dimensional symmetric matrices is proposed (see also Theorem 7.3.1 in [48] for an easier to read and slightly improved version) as well as an extension to self-adjoint Hilbert-Schmidt operators on a separable Hilbert spaces.

**Proposition 13.** *Let* $A_i$*,* $i \in [n]$ *be i.i.d copies of a Hilbert-Schmidt operator* $A$ *on the separable Hilbert space. Let* $\|A\| \leq c$ *almost surely,* $\mathbb{E}A = 0$ *and let* $\mathbb{E}[A^2] \preceq V$ *for some trace class operator* $V$*. Then with probability at least* $1 - \delta$*,*

$$\left\| \frac{1}{n} \sum_{i \in [n]} A_i \right\| \leq \frac{2c}{3n} \mathcal{L}_A(\delta) + \sqrt{\frac{2\|V\|}{n} \mathcal{L}_A(\delta)}, \tag{52}$$

*where*

$$\mathcal{L}_A(\delta) = \log \frac{4}{\delta} + \log \frac{\mathrm{tr}(V)}{\|V\|}.$$

**Proposition 14.** *Given* $\delta > 0$*, with probability in the i.i.d. draw of* $(x_i)^n_{i=1}$ *from* $\pi$*, it holds that*

$$\mathbb{P}\{\|\widehat{W}_\mu - W_\mu\| \leq \varepsilon_n(\delta)\} \geq 1 - \delta,$$

*where*

$$\varepsilon_n(\delta) = \frac{2c_{\mathcal{W}}}{3n} \mathcal{L}(\delta) + \sqrt{\frac{2\|W_\mu\|}{n} \mathcal{L}(\delta)} \quad \text{and} \quad \mathcal{L}(\delta) = \log \frac{4 \, \mathrm{tr}(W_\mu)}{\delta \, \|W_\mu\|}. \tag{53}$$

*Proof.* Proof follows directly from Proposition 13 applied to rank-$(1+p)$ operators $w\phi(x_i) \otimes w\phi(x_i)$ using the fact that $W_\mu = \mathbb{E}[w\phi(x_i) \otimes w\phi(x_i)]$. $\square$

**Proposition 15.** *Let* **(KE)** *hold for* $\tau \in [\beta, 1]$. *Given* $\delta > 0$, *with probability in the i.i.d. draw of* $(x_i)_{i=1}^n$ *from* $\pi$, *it holds that*

$$\mathbb{P}\left\{\|W_{\mu,\gamma}^{-1/2}(\widehat{W}_\mu - W_\mu)W_{\mu,\gamma}^{-1/2}\| \leq \varepsilon_n^1(\gamma, \delta)\right\} \geq 1 - \delta, \tag{54}$$

*where*

$$\varepsilon_n^1(\gamma, \delta) = \frac{2c_\tau \mu^{-\tau}}{3n\gamma^\tau}\mathcal{L}^1(\gamma, \delta) + \sqrt{\frac{2c_\tau \mu^{-\tau}}{n\gamma^\tau}\mathcal{L}^1(\gamma, \delta)}, \tag{55}$$

*and*

$$\mathcal{L}^1(\gamma, \delta) = \ln\frac{4}{\delta} + \ln\frac{\mathrm{tr}(W_{\mu,\gamma}^{-1}W_\mu)}{\|W_{\mu,\gamma}^{-1}W_\mu\|}.$$

*Moreover,*

$$\mathbb{P}\left\{\|W_{\mu,\gamma}^{1/2}\widehat{W}_{\mu,\gamma}^{-1}W_{\mu,\gamma}^{1/2}\| \leq \frac{1}{1 - \varepsilon_n^1(\gamma, \delta)}\right\} \geq 1 - \delta. \tag{56}$$

*Proof.* The idea is to apply Proposition 13 for operator $\xi(x) \otimes \xi(x)$, where $\xi(x)$ is defined in (43). Due to (1), we have that $\|A\| \leq \|\xi\|_\infty^2 \leq (\mu\gamma)^{-\tau}c_\tau$. On the other hand, we have that

$$\mathbb{E}_{x\sim\pi}[\xi(x) \otimes \xi(x)]^2 \preceq \|\xi\|_\infty^2 \mathbb{E}_{x\sim\pi}[\xi(x) \otimes \xi(x)] = \|\xi\|_\infty^2 W_{\mu,\gamma}^{-1/2}W_\mu W_{\mu,\gamma}^{-1/2},$$

and, hence (54) follows. To complete the proof, observe that

$$\|I_\mathcal{H} - W_{\mu,\gamma}^{-1/2}\widehat{W}_{\mu,\gamma}W_{\mu,\gamma}^{-1/2}\| = \|W_{\mu,\gamma}^{-1/2}(W_\mu - \widehat{W}_\mu)W_{\mu,\gamma}^{-1/2}\| \leq \varepsilon_n^1(\gamma, \delta),$$

and, hence for $\varepsilon_n^1(\gamma, \delta)$ smaller than one we obtain

$$\|W_{\mu,\gamma}^{1/2}\widehat{W}_{\mu,\gamma}^{-1}W_{\mu,\gamma}^{1/2}\| = \|(W_{\mu,\gamma}^{-1/2}\widehat{W}_{\mu,\gamma}W_{\mu,\gamma}^{-1/2})^{-1}\| \leq \frac{1}{1 - \|I_\mathcal{H} - W_{\mu,\gamma}^{-1/2}\widehat{W}_{\mu,\gamma}W_{\mu,\gamma}^{-1/2}\|}.$$

$\square$

**Proposition 16.** *Let* **(RC)**, **(SD)** *and* **(KE)** *hold for some* $\alpha \in (0, 2]$, $\beta \in (0, 1]$ *and* $\tau \in [\beta, 1]$. *Given* $\delta > 0$, *with probability in the i.i.d. draw of* $(x_i)_{i=1}^n$ *from* $\pi$, *it holds*

$$\mathbb{P}\left\{\|W_{\mu,\gamma}^{-1/2}(\widehat{W}_\mu - W_\mu)W_{\mu,\gamma}^{-1}C\|_{\mathrm{HS}} \leq \varepsilon_n^2(\gamma, \delta)\right\} \geq 1 - \delta,$$

*where*

$$\varepsilon_n^2(\gamma, \delta) = 4\sqrt{2c_\alpha c_\tau}\ln\frac{2}{\delta}\sqrt{\frac{c_\beta\mu^{-\beta}}{n\gamma^\beta} + \frac{c_\tau\mu^{-\tau}}{n^2\gamma^\tau}}\begin{cases}(\mu\gamma)^{-(\tau-\alpha)/2} & , \alpha \leq \tau, \\ c_\mathcal{W}^{(\alpha-\tau)/2} & , \alpha \geq \tau.\end{cases} \tag{57}$$

*Proof.* As before, w.l.o.g. set $\mu = 1$. First, recall that $\mathrm{HS}(\mathcal{H})$ equipped with $\|\cdot\|_{\mathrm{HS}}$ is separable Hilbert space. Hence, we will apply Proposition 12 for $A = \xi(x) \otimes \psi(x)$, where $\psi(x) = CW_{\mu,\gamma}^{-1}w\phi(x) \in \mathcal{H}^{1+p}$. To that end, observe that $\mathbb{E}\|A\|_{\mathrm{HS}}^m = \mathbb{E}\left[\|\xi(x)\|_{\mathcal{H}^{1+p}}^m \|\psi(x)\|_{\mathcal{H}^{1+p}}^m\right]$, and that

$$\mathbb{E}[\|\xi(x)\|_{\mathcal{H}^{1+p}}^m] \leq \frac{1}{2}m!\left(\gamma^{-\tau/2}\sqrt{c_\tau}\right)^{m-2}\left(\sqrt{\mathrm{tr}(W_{\mu,\gamma}^{-1}W_\mu)}\right)^2.$$

Recalling Lemma 1, the task is to bound $\|\psi(x)\|_{\mathcal{H}^{1+p}}^2 = \sum_{i\in[1+p]}\|CW_{\mu,\gamma}^{-1}w_i\phi(x)\|_\mathcal{H}^2$. Using **(RC)**, we have that

$$\|\psi(x)\|_{\mathcal{H}^{1+p}}^2 \leq c_\alpha\sum_{i\in[1+p]}\|W_\mu^{(1+\alpha)/2}W_{\mu,\gamma}^{-1}w_i\phi(x)\|_\mathcal{H}^2 = c_\alpha\sum_{j\in J}\sum_{i\in[1+p]}\langle W_\mu^{(1+\alpha)/2}W_{\mu,\gamma}^{-1}w_i\phi(x), h_j\rangle_\mathcal{H}^2.$$

But, since

$$\langle W_\mu^{(1+\alpha)/2}W_{\mu,\gamma}^{-1}w_i\phi(x), h_j\rangle_\mathcal{H} = \frac{\sigma_j^{1+\alpha}}{\sigma_j^2 + \gamma}\langle w_i\phi(x), h_j\rangle_\mathcal{H},$$

expanding as in proof of Lemma 1 and using **(KE)**, we have that

$$\|\psi(x)\|_{\mathcal{H}^{1+p}}^2 \leq c_\alpha\sum_{j\in J}\sum_{i\in[1+p]}\left[\frac{\sigma_j^{(2+\alpha-\tau)}}{\sigma_j^2 + \gamma}\right]^2\frac{\langle w_i\phi(x), h_j\rangle^2}{\sigma_j^2}\sigma_j^{2\tau} \leq \begin{cases}\gamma^{-(\tau-\alpha)} & , \alpha \leq \tau, \\ c_\mathcal{W}^{(\alpha-\tau)} & , \alpha \geq \tau.\end{cases}$$

Therefore, we can set

$$\Lambda^2 = c_\alpha c_\tau^2 \begin{cases} \gamma^{-(2\tau-\alpha)} & , \alpha \le \tau, \\ c_{\mathcal{W}}^{(\alpha-\tau)} \gamma^{-\tau} & , \alpha \ge \tau. \end{cases} \quad \text{and} \quad \sigma^2 = c_\alpha c_\tau c_\beta \begin{cases} \gamma^{-(\beta+\tau-\alpha)} & , \alpha \le \tau, \\ c_{\mathcal{W}}^{(\alpha-\tau)} \gamma^{-\beta} & , \alpha \ge \tau. \end{cases}$$

in Proposition 12 to obtain (58). $\qquad\square$

**Proposition 17.** *Let* **(KE)** *hold for* $\tau \in [\beta, 1]$. *Given* $\delta > 0$, *with probability in the i.i.d. draw of* $(x_i)_{i=1}^n$ *from* $\pi$, *it holds*

$$\mathbb{P}\left\{ \|W_{\mu,\gamma}^{-1/2}(\widehat{C} - C)\|_{\mathrm{HS}} \le \varepsilon_n^3(\gamma, \delta) \right\} \ge 1 - \delta,$$

*where*

$$\varepsilon_n^3(\gamma, \delta) = \frac{4\sqrt{2\,c_{\mathcal{W}}}}{\mu} \ln \frac{2}{\delta} \sqrt{\frac{c_\beta \mu^{-\beta}}{n\gamma^\beta} + \frac{c_\tau \mu^{-\tau}}{n^2\gamma^\tau}}. \tag{58}$$

*Proof.* First, note that since $\|\phi(x)\|_{\mathcal{H}}^2 \le \mu^{-1}\|w\phi(x)\|_{\mathcal{H}^{1+p}}^2$ and $C \preceq \mu^{-1}W_\mu$, **(KE)** and **(SD)** for $W_\mu$ imply analogous assumptions for $C$. Hence, we can apply Proposition 13 from [29], which using the observation that $\|W_{\mu,\gamma}^{-1/2}C_\gamma^{1/2}\|^2 = \|C_\gamma^{1/2}(\mu C_\gamma - T)^{-1}C_\gamma^{1/2}\| \le \mu^{-1}\|C_\gamma C_\gamma^{-1}\| = \mu^{-1}$, completes the proof. $\qquad\square$

Next, we develop concentration bounds of some key quantities used to build RRR empirical estimator.

### E.3.2  Variance and Norm of KRR Estimator

**Proposition 18.** *Let* **(RC)**, **(SD)** *and* **(KE)** *hold for some* $\alpha \in (0, 2]$, $\beta \in (0, 1]$ *and* $\tau \in [\beta, 1]$. *Given* $\delta > 0$ *if* $\varepsilon_n^1(\gamma, \delta) < 1$, *then with probability at least* $1 - \delta$ *in the i.i.d. draw of* $(x_i, y_i)_{i=1}^n$ *from* $\rho$

$$\mathbb{P}\left\{ \|W_{\mu,\gamma}^{1/2}(\widehat{G}_{\mu,\gamma} - G_{\mu,\gamma})\| \le \frac{\varepsilon_n^2(\gamma, \delta/3) + \varepsilon_n^3(\gamma, \delta/3)}{1 - \varepsilon_n^1(\gamma, \delta/3)} \right\} \ge 1 - \delta.$$

*Proof.* Note that $W_{\mu,\gamma}^{1/2}(\widehat{G}_{\mu,\gamma} - G_{\mu,\gamma}) = W_{\mu,\gamma}^{1/2}(\widehat{W}_{\mu,\gamma}^{-1}\widehat{C} - W_{\mu,\gamma}^{-1}C)$, and, hence,

$$W_{\mu,\gamma}^{1/2}(\widehat{G}_{\mu,\gamma} - G_{\mu,\gamma}) = W_{\mu,\gamma}^{1/2}\widehat{W}_{\mu,\gamma}^{-1}(\widehat{C} - \widehat{W}_{\mu,\gamma}W_{\mu,\gamma}^{-1}C \pm C)$$
$$= W_{\mu,\gamma}^{1/2}\widehat{W}_{\mu,\gamma}^{-1}W_{\mu,\gamma}^{1/2}\left(W_{\mu,\gamma}^{-1/2}(\widehat{C} - C) - W_{\mu,\gamma}^{-1/2}(\widehat{W}_\mu - W_\mu)W_{\mu,\gamma}^{-1}C\right). \tag{59}$$

Thus, taking the norm and using $\|C_\gamma^{-1}T\| \le c_\alpha\,\sigma_1^{\alpha-1}(S_\pi)$ with the Propositions 16 and 15 we prove the first bound. $\qquad\square$

### E.3.3  Variance of Singular Values

In this section we prove concentration of singular values computed in Theorem 1, a necessary step to derive learining rates for RRR estimator.

**Proposition 19.** *Let* **(RC)**, **(SD)** *and* **(KE)** *hold for some* $\alpha \in (0, 2]$, $\beta \in (0, 1]$ *and* $\tau \in [\beta, 1]$. *Let* $B = W_{\mu,\gamma}^{-1/2}C$ *and* $\widehat{B} = \widehat{W}_{\mu,\gamma}^{-1/2}\widehat{C}$. *Given* $\delta > 0$ *if* $\varepsilon_n^1(\gamma, \delta/5) < 1$, *then with probability at least* $1 - \delta$ *in the i.i.d. draw of* $(x_i)_{i=1}^n$ *from* $\pi$

$$|\sigma_i^2(\widehat{B}) - \sigma_i^2(B)| \le \|\widehat{B}^*\widehat{B} - B^*B\| \le \varepsilon_n^4(\gamma, \delta/3), \tag{60}$$

*where*

$$\varepsilon_n^4(\gamma, \delta/3) = (\varepsilon_n^2(\gamma, \delta/3) + \varepsilon_n^3(\gamma, \delta/3))\left(\frac{1}{\mu} + \frac{\varepsilon_n^2(\gamma, \delta/3) + \varepsilon_n^3(\gamma, \delta/3)}{1 - \varepsilon_n^1(\gamma, \delta/3)}\right). \tag{61}$$

*Proof.* We start from the Weyl's inequalities for the square of singular values

$$|\sigma_i^2(\widehat{B}) - \sigma_i^2(B)| \le \|\widehat{B}^*\widehat{B} - B^*B\|, \ i \in [n].$$

But, since,

$$\widehat{B}^*\widehat{B} - B^*B = \widehat{C}^*\widehat{W}_{\mu,\gamma}^{-1}\widehat{C} - C^*W_{\mu,\gamma}^{-1}C = (\widehat{C}-C)^*\widehat{W}_{\mu,\gamma}^{-1}\widehat{C} + C^*W_{\mu,\gamma}^{-1}(\widehat{C}-C) + C^*(\widehat{W}_{\mu,\gamma}^{-1} - W_{\mu,\gamma}^{-1})\widehat{C}$$

denoting $M = W_{\mu,\gamma}^{-1/2}(\widehat{C}-C)$, $N = W_{\mu,\gamma}^{-1/2}(\widehat{W}_\mu - W_\mu)$ and $R = W_{\mu,\gamma}^{1/2}(\widehat{G}_{\mu,\gamma} - G_{\mu,\gamma})$, we have

$$
\begin{aligned}
\widehat{B}^*\widehat{B} - B^*B &= B^*M + M^*C_\gamma^{1/2}\widehat{G}_{\mu,\gamma} - B^*N\widehat{G}_{\mu,\gamma} = B^*M + (M^*C_\gamma^{1/2} - B^*N)(\widehat{G}_{\mu,\gamma} \pm G_{\mu,\gamma}) \\
&= B^*M + M^*B - B^*NG_{\mu,\gamma} + (M^* - B^*NW_{\mu,\gamma}^{-1/2})R \\
&= (G_{\mu,\gamma})^*(\widehat{C}-C) + (\widehat{C}-C)G_{\mu,\gamma} - (G_{\mu,\gamma})^*(\widehat{W}_\mu - W_\mu)G_{\mu,\gamma} + (M^* + (G_{\mu,\gamma})^*N^*)R.
\end{aligned}
$$

Therefore, recalling that, due to (59), $R = W_{\mu,\gamma}^{1/2}\widehat{W}_{\mu,\gamma}^{-1}W_{\mu,\gamma}^{1/2}(M - NG_{\mu,\gamma})$, we conclude

$$
\begin{aligned}
\widehat{B}^*\widehat{B} - B^*B =& (G_{\mu,\gamma})^*(\widehat{C}-C) + (\widehat{C}-C)G_{\mu,\gamma} - (G_{\mu,\gamma})^*(\widehat{W}_\mu - W_\mu)G_{\mu,\gamma} \\
&+ (M - NG_{\mu,\gamma})^*W_{\mu,\gamma}^{1/2}\widehat{W}_{\mu,\gamma}^{-1}W_{\mu,\gamma}^{1/2}(M - NG_{\mu,\gamma}).
\end{aligned}
\tag{62}
$$

Next, observe that

- $\|(\widehat{C}-C)G_{\mu,\gamma}\| \leq \|(\widehat{C}-C)W_{\mu,\gamma}^{1/2}\|\|W_{\mu,\gamma}^{1/2}C\| \leq \mu^{-1}\|W_{\mu,\gamma}^{1/2}(\widehat{C}-C)\|$ is bounded by Proposition 17,

- $\|M - NG_{\mu,\gamma}\| \leq \|W_{\mu,\gamma}^{-1/2}(\widehat{C}-C)\| + \|W_{\mu,\gamma}^{-1/2}(\widehat{W}_\mu - W_\mu)W_{\mu,\gamma}^{-1}C\|$ is bounded by Propositions 16 and 17,

- $\|G_{\mu,\gamma}^*(\widehat{W}_\mu - W_\mu)G_{\mu,\gamma}\| \leq \mu^{-1}\|W_{\mu,\gamma}^{-1/2}(\widehat{W}_\mu - W_\mu)W_{\mu,\gamma}^{-1}C\|$ is bounded by Proposition 16.

Therefore, using additionally Proposition 15 result follows. □

Remark that to bound singular values we can rely on the fact

$$|\sigma_i(\widehat{B}) - \sigma_i(B)| = \frac{|\sigma_i^2(\widehat{B}) - \sigma_i^2(B)|}{\sigma_i(\widehat{B}) + \sigma_i(B)} \leq \frac{|\sigma_i^2(\widehat{B}) - \sigma_i^2(B)|}{\sigma_i(\widehat{B}) \vee \sigma_i(B)}.$$

### E.3.4 Variance of RRR Estimator

Recalling the notation $B := W_{\mu,\gamma}^{-1/2}C$ and $\widehat{B} := \widehat{W}_{\mu,\gamma}^{-1/2}\widehat{C}$, let denote $P_r$ and $\widehat{P}_r$ denote the orthogonal projector onto the subspace of leading $r$ right singular vectors of $B$ and $\widehat{B}$, respectively. Then we have $[\![B]\!]_r = BP_r$ and $[\![\widehat{B}]\!]_r = \widehat{B}\widehat{P}_r$, and, hence $G_{\mu,\gamma}^r = G_{\mu,\gamma}P_r$ and $\widehat{G}_{\mu,\gamma}^r = \widehat{G}_{\mu,\gamma}\widehat{P}_r$.

**Proposition 20.** *Let* **(RC)**, **(SD)** *and* **(KE)** *hold for some* $\alpha \in (0,2]$, $\beta \in (0,1]$ *and* $\tau \in [\beta, 1]$. *Given* $\delta > 0$ *and* $\gamma > 0$, *if* $\varepsilon_n^1(\gamma, \delta) < 1$, *then with probability at least* $1 - \delta$ *in the i.i.d. draw of* $(x_i)_{i=1}^n$ *from* $\pi$,

$$\|Z_\mu(G_{\mu,\gamma}^r - \widehat{G}_{\mu,\gamma}^r)\| \leq \frac{\varepsilon_n^2(\gamma, \delta/3) + \varepsilon_n^3(\gamma, \delta/3)}{1 - \varepsilon_n^1(\gamma, \delta/3)} + \frac{\sigma_1(B)}{\sigma_r^2(B) - \sigma_{r+1}^2(B)}\varepsilon_n^4(\gamma, \delta/3). \tag{63}$$

*Proof.* Start by observing that $\|Z_\mu(G_{\mu,\gamma}^r - \widehat{G}_{\mu,\gamma}^r)\| \leq \|W_{\mu,\gamma}^{1/2}(G_{\mu,\gamma}^r - \widehat{G}_{\mu,\gamma}^r)\|$ and

$$
\begin{aligned}
W_{\mu,\gamma}^{1/2}(G_{\mu,\gamma}^r - \widehat{G}_{\mu,\gamma}^r) =& (W_{\mu,\gamma}^{1/2}\widehat{W}_{\mu,\gamma}^{-1}W_{\mu,\gamma}^{1/2}) \cdot \\
&\left( W_{\mu,\gamma}^{-1/2}(\widehat{W}_\mu - W_\mu)G_{\mu,\gamma}P_r + W_{\mu,\gamma}^{-1/2}(\widehat{C} - C)\widehat{P}_r + B(\widehat{P}_r - P_r) \right).
\end{aligned}
$$

Using that the norm of orthogonal projector $\widehat{P}$ is bounded by one and applying Propositions 15 and 16 together with Propositions 4 and 19 completes the proof. □

## E.4 Proof of Theorem 2

First, observe that **(DF)** implies that

$$\epsilon \widehat{W}_\mu \preceq \widehat{W}_\mu - \widehat{W}_\mu^\epsilon \preceq \epsilon \widehat{W}_\mu,$$

where the empirical covariance with the inexact Dirichlet coefficient $s_\epsilon$ is denoted by $\widehat{W}_\mu^\epsilon$. Since RRR algorithm now uses $\widehat{W}_\mu^\epsilon$, when $\epsilon > 0$ the only change to our proof technique lies in the analysis of variance in section E.3. In particular, we only need to adapt Propositions 15 and 16, which is straightforward. Indeed, we have

$$\begin{aligned}
\|F(W_\mu - \widehat{W}_\mu^\epsilon)G\| &= \|F(W_\mu \pm \widehat{W}_\mu - \widehat{W}_\mu^\epsilon)FG\| \\
&\leq \|F(W_\mu - \widehat{W}_\mu)FG\| + \|F(\widehat{W}_\mu^\epsilon - \widehat{W}_\mu)F\|\|G\| \\
&\leq \|F(W_\mu - \widehat{W}_\mu)FG\| + \epsilon\|F\widehat{W}_\mu F\|\|G\|
\end{aligned}$$

for $F = W_{\mu,\gamma}^{-1/2}$ and $G$ being either $W_{\mu,\gamma}^{-1/2}C$ or $I$. Thus, since $\|F\widehat{W}_\mu F\| \leq \frac{1}{1-\varepsilon^1(\gamma,\delta)}$ and $\|G\|$ is either 1 or bounded by $\sqrt{c_\mathcal{H}/\mu}$, in conclusion the relative error $\epsilon$ of imperfect knowledge simply appears as the additive term in the final guarantees. Thus, in what follows, we present that case $\epsilon = 0$.

### E.4.1 Operator Norm Error Bounds

Summarising previous sections, in order to prove Theorem 2, we just need to analyse the bounds $\varepsilon_n^1$, $\varepsilon_n^2$ and $\varepsilon_n^3$. Not that we fix the hyperparameter $\mu > 0$, which affects the constants, but need to chose the decay rate of Tikhonov regularization parameter $\gamma > 0$ to obtain balancing of bias and variance in the generalization bounds.

Let us first assume regime $\alpha \geq \tau$, which covers "well-specified learning" when non-trivial eigenfunctions of $L$ are inside $\mathcal{H}$. Since $\alpha \geq \tau$ and $\beta \leq \tau$, we have that for large enough $n$ one has

$$\varepsilon_n^1(\gamma,\delta) \lesssim \frac{n^{-1/2}}{\gamma^{\tau/2}}\ln\delta^{-1} \quad \text{and} \quad \varepsilon_n^i(\gamma,\delta) \lesssim \left(\frac{n^{-1/2}}{\gamma^{\beta/2}} \vee \frac{n^{-1}}{\gamma^{\tau/2}}\right)\ln\delta^{-1}, \; i = 2,3,4. \tag{64}$$

But, since the bias term is $\lesssim \gamma^{\alpha/2}$ and the slow term from Proposition 20 is $1/\sqrt{n\gamma^\beta}$, we can set $\gamma_n = n^{-\frac{1}{\alpha+\beta}}$ and obtain

$$\gamma_n^{\alpha/2} = \frac{n^{-1/2}}{\gamma_n^{\beta/2}} = n^{-\frac{\alpha}{2(\alpha+\beta)}}, \quad \text{and} \quad \frac{n^{-1/2}}{\gamma_n^{\tau/2}} = n^{-\frac{\alpha+\beta-\tau}{2(\alpha+\beta)}},$$

which, due to $\alpha \geq \tau$, implies

$$\lim_{n\to\infty} \varepsilon_n^1(\gamma_n,\delta/3) = \lim_{n\to\infty} \varepsilon_n^2(\gamma_n,\delta/3) = \lim_{n\to\infty} \varepsilon_n^3(\gamma,\delta/3) = \lim_{n\to\infty} \varepsilon_n^4(\gamma,\delta/3) = 0.$$

Therefore, for this choice of regularization parameter Equation (46) with Propositions 9, 11 and 20 assure that

$$\mathcal{E}(\widehat{G}_{\mu,\gamma}^r) - \sigma_{r+1}(B) \leq \mathcal{E}(\widehat{G}_{\mu,\gamma}^r) - \sqrt{\lambda_{r+1}^\star} \lesssim n^{-\frac{\alpha}{2(\alpha+\beta)}}. \tag{65}$$

Now, let us consider the more difficult to learn case $\alpha < \tau$ when eignefunctions of the generator have only weaker norms than the RKHS one. Then, for large enough $n$ bounds in (64) hold for $i = 3$, but

$$\varepsilon_n^2(\gamma,\delta) \vee \varepsilon_n^4(\gamma,\delta) \lesssim \left(\frac{n^{-1/2}}{\gamma^{(\beta+\tau-\alpha)/2}} \vee \frac{n^{-1}}{\gamma^{(2\tau-\alpha)/2}}\right)\ln\delta^{-1}. \tag{66}$$

Then, by balancing the slow terms with bias $\gamma^{\alpha/2}$, we set $\gamma_n = n^{-\frac{1}{\tau+\beta}}$ and obtain

$$\gamma_n^{\alpha/2} = \frac{n^{-1/2}}{\gamma_n^{(\tau+\beta-\alpha)/2}} = n^{-\frac{\alpha}{2(\tau+\beta)}}, \quad \text{and} \quad \frac{n^{-1/2}}{\gamma_n^{\tau/2}} = n^{-\frac{\beta}{2(\tau+\beta)}},$$

which, since $\tau \geq \beta$, implies

$$\lim_{n\to\infty} \varepsilon_n^1(\gamma_n,\delta/3) = \lim_{n\to\infty} \varepsilon_n^2(\gamma_n,\delta/3) = \lim_{n\to\infty} \varepsilon_n^3(\gamma,\delta/3) = \lim_{n\to\infty} \varepsilon_n^4(\gamma,\delta/3) = 0,$$

and we obtain

$$\mathcal{E}(\widehat{G}^r_{\mu,\gamma}) - \sigma_{r+1}(B) \leq \mathcal{E}(\widehat{G}^r_{\mu,\gamma}) - \lambda^\star_{r+1} \lesssim n^{-\frac{\alpha}{2(\tau+\beta)}}. \tag{67}$$

Therefore, denoting

$$\gamma_n \asymp \begin{cases} n^{-\frac{1}{\tau+\beta}} & , \alpha \leq \tau, \\ n^{-\frac{1}{\alpha+\beta}} & , \alpha \geq \tau, \end{cases} \quad \text{and} \quad \varepsilon^\star_n = \begin{cases} n^{-\frac{\alpha}{2(\tau+\beta)}} & , \alpha \leq \tau, \\ n^{-\frac{\alpha}{2(\alpha+\beta)}} & , \alpha \geq \tau. \end{cases} \tag{68}$$

as a consequence the operator norm error bound in Theorem 2 holds, and We have the following result on the estimation of the singular values of $(\mu I - L)^{-1} Z_\mu$, denoted by $\lambda^\star_i$, using the singular values of $\widehat{B}$, denoted by $\widehat{\sigma}_i$, from Theorem 1.

**Proposition 21.** *Let* **(RC)***,* **(SD)** *and* **(KE)** *hold for some $\alpha \in (0, 2]$ and $\beta \in (0, 1]$ and $\tau \in [\beta, 1]$, and define* (68)*. Then, there exists a constant $c > 0$ such that for every given $\delta \in (0, 1)$, large enough $n > r$ and with probability at least $1 - \delta$ in the i.i.d. draw of $(x_i)_{i=1}^n$ from $\pi$ for all $i \in [r]$,*

$$|\widehat{\sigma}_i^2 - \lambda^\star_i| \lesssim \varepsilon^\star_n \ln \delta^{-1} \tag{69}$$

*Proof.* The proof is direct consequence of Propositions 10 and 19 using (64)-(66). □

### E.5 Spectral Learning Rates

Finally, we conclude the proof of Theorem 2 showing the concentration of eigenpairs. Recalling Proposition 2, we need to combine the operator norm bound and metric distortion. Since, as indicated in Proposition 9, the population KRR estimator can grow in the operator norm whenever the regularity condition is violated $\alpha < 1$, leading to possibly unbounded metric distortions w.r.t. increasing sample size, we restrict to the case $\alpha \geq 1$.

First, combining (65)-(67) and Proposition 21, we have that

$$\mathcal{E}(\widehat{G}^r_{\mu,\gamma}) \leq (\widehat{\sigma}_{r+1} \wedge \lambda^\star_{r+1}) + c \varepsilon^\star_n \ln \delta^{-1}.$$

On the other hand, from Propositions 40 and 14, we have that with failure probability $\delta$,

$$(\widehat{\eta}_i)^2 - (\eta(\widehat{h}_i))^{-2} \leq \varepsilon_n(\delta),$$

which, if we can prove that $\widehat{\eta}_i$ and $\eta(\widehat{h}_i)$ are bounded, concludes the proof for empirical spectral biases. To that end, recall that, c.f. Proposition 5, for RRR estimator we have

$$\eta(\widehat{h}_i) \leq \frac{\|\widehat{G}^r_{\mu,\gamma}\|}{\sigma_r(Z_\mu \widehat{G}^r_{\mu,\gamma})} \leq \frac{\|\widehat{G}_{\mu,\gamma}\|}{\sigma_r((\mu I - L)^{-1} Z_\mu) - \mathcal{E}(\widehat{G}^r_{\mu,\gamma})} \leq \frac{\|G_{\mu,\gamma}\| + \|\widehat{G}_{\mu,\gamma} - G_{\mu,\gamma}\|}{\sqrt{\lambda^\star_r} - \mathcal{E}(\widehat{G}^r_{\mu,\gamma})}$$

$$\leq \frac{\|G_{\mu,\gamma}\| + (\mu\gamma)^{-1/2}\|W^{1/2}_{\mu,\gamma}(\widehat{G}_{\mu,\gamma} - G_{\mu,\gamma})\|}{\sqrt{\lambda^\star_r} - \mathcal{E}(\widehat{G}^r_{\mu,\gamma})} \lesssim \frac{1 + n^{-1/2}\gamma^{-(\beta+1)/2}}{\sqrt{\lambda^\star_r} - n^{-1/2}\gamma^{-\beta/2}} = \frac{1 + n^{-\frac{\alpha-1}{\alpha+\beta}}}{\sqrt{\lambda^\star_r} - n^{-\frac{\alpha}{\alpha+\beta}}}$$

where in the last inequality we have applied Propositions 9 and 18. Thus, using Proposition 40, the proof of Theorem 2 is concluded.

### E.6 Discussion of the learning rates

We first discuss the learning rates reported in Table 1. Notice that although the papers we compare to employ a different risk, our comparison remains meaningful because our energy-based risk measure provides an upper bound on their risk measure. This ensures that any upper bound derived with our risk also applies to theirs.

- The learning bound for IG obtained is [1] covers only pure diffusion processes (Laplacian with constant weights). Their learning rate is non-parametric and depends on the state space dimension $d$ in a counter-intuitive way $O(n^{-\frac{d}{2(d+1)}})$ in [1, Theorem 3], highlighting a potential limitation of their approach. In comparison, when we specify our RRR method with an RBF kernel (i.e. $\beta = 0$), we achieve a much faster parametric learning rate $O(n^{-1/2})$.

- The recent work of [43] covers Langevin processes via a kernel approach, but they derive a sub-optimal learning bound for IG of order $O(n^{-1/4})$ in [43, Theorem 4.4]. For Langevin diffusions, our RRR method with an RBF kernel achieves a faster parametric rate $O(n^{-1/2})$. Moreover, the computational complexity of their method is $O(n^3 d^3)$, which limits its application in realistic molecular dynamics scenarios.

- As for [19, Theorem 7], although they considered general diffusions, they only derived a suboptimal bound for the variance component of their risk, with an explicit dependence on the dimension of the state space.

Additionally, note that the mentioned works lack learning guarantees for eigenfunctions and eigenvalues. Notably, their methods are prone to the spurious eigenvalue phenomenon, requiring expert manual review of each eigenpair to select plausible ones.

Next, we contrast our IG learning and well established TO learning. First, note that TO methods apply only to equally spaced data and the sampling frequency $1/\Delta t$ must be high enough to distinguish all relevant time-scales. Otherwise, since TO eigenvalues are $e^{\lambda_i \Delta t}$, small spectral gaps complicate learning (see [28, Thm. 3]). Conversely, our IG method, which uses gradient information, is time-scale independent, handles irregularly spaced measurements, and does not rely on time discretizations. Indeed, recalling the risk functional in Eq. (8), we see that the "label" of the model $\chi_\mu(x) \approx G^* \phi(x)$ is the action of the resolvent. Since this "label" is not computable, we "*fight fire (resolvent) with fire (generator)*", i.e. we use the energy norm of Eq. (9) to rewrite regularized problem (8). Crucially, this allows us to obtain estimators via energy covariance $W_\mu$ in (14) that *completely captures infinitesimal nature* of the learning problem without needing time-lagged observations. This contrasts with TO methods, where choosing the time-lag $\Delta t$ is the major bottleneck in real applications. Indeed, to obtain data from an invariant distribution $\pi$ one uses the trajectory data after some burn-in time needed to ensure that ergodic mean approximates well $\pi$. Then, the problem is reduced to studying only the dependence as is done e.g. in [27] using standard tools of $\beta$-mixing and the method of blocks. This allows one to obtain non-parametric learning bounds for TO methods where the effective sample size suffers from the multiplicative effect of the time-lag to achieve approximate independence. So, TO methods "waste" a lot of data negatively impacting statistical accuracy. Contrary to this, our method can be applied to data with larger time-lags (even irregularly spaced) so that effective sample size is close to the true one. All this results in better generalization, as shown in Fig. 1 g)-h)-i) where our IG estimator captures ground truth significantly better than TO for the same sample size, this generalization across all time scales incurs quadratic computational complexity w.r.t. state dimension and not statistical accuracy. Lastly, with our additional discussion on imperfect knowledge, our IG method can also be applied in a fully data-driven regime as TO methods.

To conclude this section, we contrast our method with classical numerical methods for the spectral decomposition of differential operators, such as finite-elements (FEM). For FEM, the approximation error is $|\lambda_k - \widehat{\lambda}_k| \leq c h^{2p} \lambda_k$, where $p$ is the polynomial degree used to construct finite elements, $h$ is the mesh size, and $c$ depends on the eigenfunctions' smoothness. As the number of mesh elements grows exponentially with $d$ (i.e., $\sim h^{-d}$), reducing $h$ is the major bottleneck mitigated by computationally demanding adaptive higher order methods. On the other hand, our IG method that requires less or no knowledge has a quadratic impact of the $d$ only on the computational complexity. Indeed, sample complexity depends on the effective dimension of the equilibrium distribution on the domain that can be much lower than $d$. Therefore, RRR for IG's resolvent doesn't suffer from the curse of dimensionality as the FEM does.

## F  Experiments

All the experiments were performed on a workstation with 125.6 GiB of memory and AMD® Ryzen threadripper pro 3975wx 32-cores × 64 processor, no graphics card was used. The version of python used is Python 3.9.18. The choice of the kernel in all experiments was Gaussian RBF with specified length-scales, and the hyperparameters were chosen via cross-validation. The code to reproduce the results of the experiments can be found in the following repository: https://github.com/DevergneTimothee/GenLearn_kernel

**Four well potential** For this experiment, we used an in-house code to simulate the system. The equations of motions were discretized using the Euler-Maruyama scheme with a timestep of $10^{-4}$.

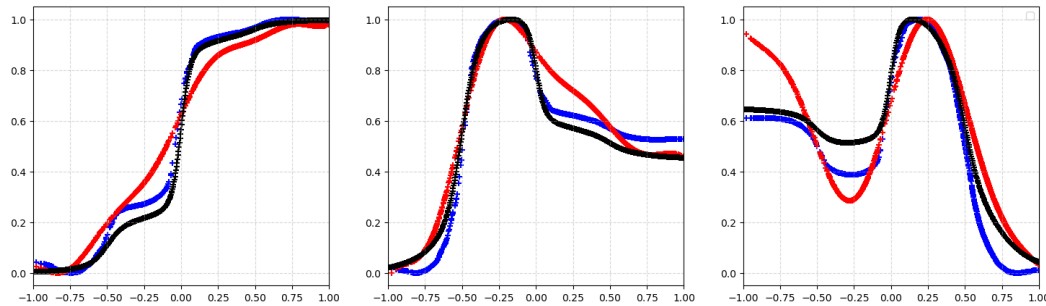

Figure 2: Results of the RRR given by our method for two different length scales (blue and red) compared with ground truth (black) for the Langevin dynamics driven by a four well one dimensional potential.

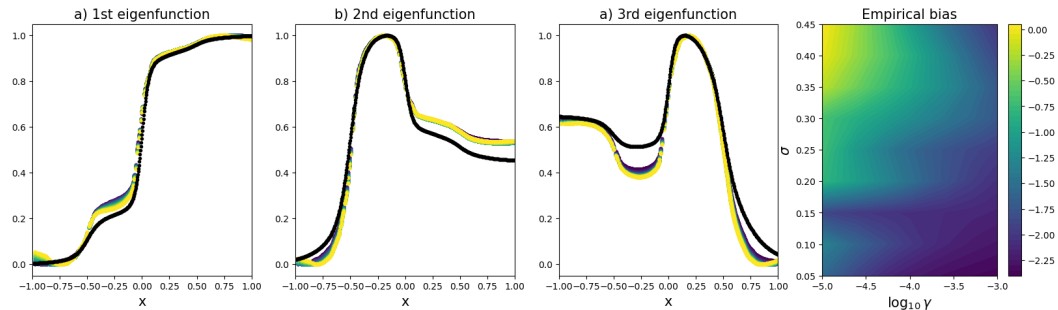

Figure 3: Panels a)-c): Test of the model's robustness with respect to the hyperparameter $\mu$, tested for 30 different values between $10^{-3}$ and 5, compared to the ground truth result. Panel d): logarithm of the empirical bias as a function of the kernel length scale $\sigma$ and the logarithm of regularization parameter $\gamma$.

RRR was fitted using 1000 points, $\mu = 5$ and $\gamma = 10^{-5}$. The length scales used were 0.05 and 0.5. This experiment was reproduced 100 times leading to very small change in the estimation of the eigenfunctions. In Figure 2 we report the result of one of them. The reduced rank regression was performed with a rank of 5. Further, in Figure 3 we show the robustness of eigenfunctions w.r.t. choice of shift hyper parameter $\mu$, as well as values of empirical bias for different values of length-scale and regularization hyperparameters. Concerning panel **c)** of Figure 1 where we show the consistency of our model with the true Boltzmann distribution. Namely, we use our model to forecast the conditional probability density function (pdf) of the system being in one bin, given it started at some point and after 100000 steps (so that it relaxes towards the equilibrium distribution). We bin the space into 50 bins and approximate the pdf via Eq. (12), where $h$ is characteristic function of a bin. In order to achieve an accurate estimation, we use the knowledge that the leading eigenvalue is zero. Additionally, we perform the same procedure with imperfect diffusion coefficient estimated from data ($k_b T = 0.45 \pm 0.01$ instead of true value 0.5 a.u.) by looking at the variance of increments over 10 steps. Note that the result (green dotted lines) is unchanged and compares well with the analytical Boltzmann distribution (black lines). Finally, we report that if the same approach is used with the method described in [19, 43], no dynamical quantity can be forecasted due to many spurious eigenpairs (see Figure 1a) which prevents the system from relaxing towards the Boltzmann distribution.

**Muller Brown** For this experiment, we used an in-house code to simulate the system. The equations of motions were discretized using the Euler-Maruyama scheme with a timestep of $10^{-3}$ and a temperature of 2 (arbitrary units). RRR was fitted using 2000 points, $\mu = 1$ and $\gamma = 10^{-5}$. The length scale used was 0.6. The reduced rank regression was performed with a rank of 5

**CIR model** For this experiment, we reproduced 100 times the simulations in order to obtain statistical uncertainties. We used a length scale of 0.5, $\mu = 1$ and $\gamma = 10^{-6}$ . The reduced rank regression was performed with a rank of 2.

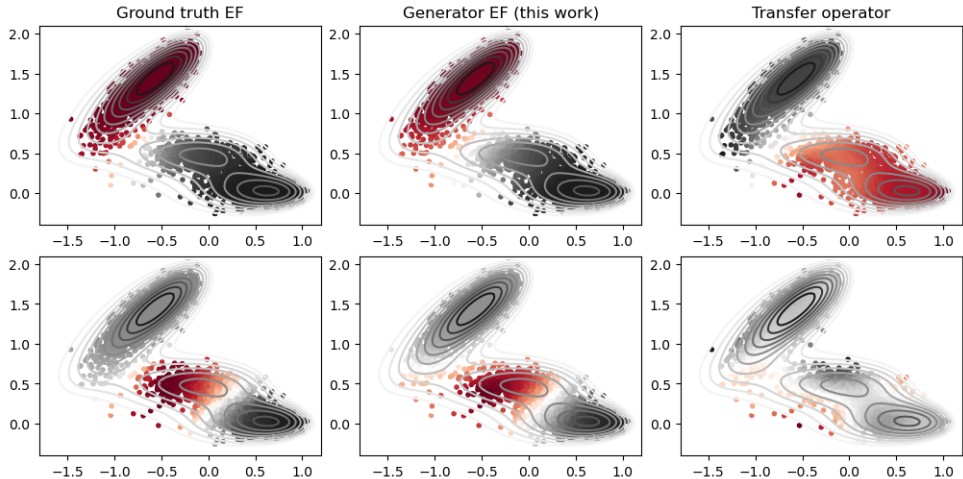

Figure 4: Results of the RRR given by our method (second column) compared to ground truth (first column) and transfer operator RRR (last column). Points are colored according to the value of the eigenfunction

**US mortgage rates** We have trained our method on a real 30-year US mortgage rates dataset and contrasted it with the fitted CIR model using continuous ranked probability scores that are estimated from the forecasts obtained by of each of them. Each model has been trained using data from January 2009 to December 2016. The initial condition was the last week of December 2016 and the predictions were made for the years 2017 and 2018. Since the dataset is real, we used the imperfect partial knowledge, that is, for our method, we estimated the diffusion coefficient only via a least squares calibration of a CIR model over the training set. This allows more flexibility on the drift term in our model.

