# OpenReview forum: "Learning the Infinitesimal Generator of Stochastic Diffusion Processes"
_NeurIPS.cc/2024/Conference — NeurIPS 2024 poster_

### Official Review · Reviewer_D2P3 · 2024-07-11

**Soundness:** 4
**Presentation:** 4
**Contribution:** 3
**Rating:** 8
**Confidence:** 4

**Summary:**

This paper proposes a relevant and sound approach for learning self-adjoint SDE generators via operator learning techniques.

The paper includes a compactification, a novel prior knowledge inclusion, and first-of-its-kind statistical learning guarantees that extend the known ones from discrete Markov processes.

**Strengths:**

1. The paper is easy to read, with suitable mediation throughout.

2. Inserting known diffusion effects into the Dirichlet forms and using the resolvent, provides an elegant way of including prior knowledge and well-posed estimation.

3. These are the most complete statistical learning guarantees for spectra of self-adjoint generators (albeit reliant on partial knowledge).

**Weaknesses:**

1. Unclear extensions to entirely data-driven regimes (no partial knowledge) and requiring sampling from an invariant distribution.

2. Experiments only consider toy examples that validate theory but do not showcase the practicality of the approach for more general cases.

3. I am not sure that contribution **4)** is well supported by the current presentation, primarily due to hard-to-parse figures. A revised comparison and legends with sharper plots could make the results much easier to understand.

**Questions:**

1. *Are Dirichlet forms commonly deduced from the diffusion part of the SDE?* It would help assess the ease of knowing these forms beforehand for readers unfamiliar with the notion.

2. *Is it reasonable to think you could deal with a bias of the Dirichlet form using imperfect diffusion knowledge?*

3. *Could you highlight what practical gains/losses are introduced w.r.t. discrete-time setting?* I would guess you need less data due to diffusion priors and not requiring as small of a discretization. On the other hand, IG regression complexity seems higher for high-dimensional systems.

4. *Can a sample complexity gain be recognized using prior knowledge in this form for your theoretical results?* Could the performance gain w.r.t. TO learning be recognized using the derived bounds?

5. *How arbitrary is the choice of $\mu$?*

6. *How does the time-sampled data come into play, and does the sampling rate of the data influence any aspect of the approach?* Does sampling the invariant distribution + Dirichlet form allow you to avoid time-derivative or "1-step" dynamics observations (e.g., compared to [A])? Based on the current writing, it is not immediately clear to me (but I might have missed something).

7. *How do you compare (using full knowledge) to classical numerical methods using known drift and diffusion? Would your sample efficiency be better than classical FEM for a given precision?* You mention how, for known operators (via drift and diffusion knowledge), there would be no spuriousness in estimating eigenpairs. This is perhaps of separate interest.


[A] - Meng, Y., Zhou, R., Ornik, M., & Liu, J. (2024). Koopman-Based Learning of Infinitesimal Generators without Operator Logarithm. http://arxiv.org/abs/2403.15688

**Limitations:**

1. *Possibly strong prior knowledge requirements.*
2. *Limited to self-adjoint generators.*
3. *Experiments are on 1D toy examples.*

---

> ### Author Rebuttal · Authors · 2024-08-06
>
> We appreciate the reviewer's insightful evaluation and valuable comments. In what follows, we aim to address the highlighted weaknesses and respond to the reviewer's questions.
>
> ## Weaknesses:
> 1. Thank you for emphasizing the realistic setting of imperfect partial knowledge. This motivated us to __show theoretically and empirically__ that indeed our IG learning method with estimated Dirichlet form __is guaranteed to work__. Concerning the sampling, our assumption to have i.i.d. data from the invariant distribution $\pi$ can be relaxed as we reply in Q6.
> 2. Concerning the experiments, please see our general reply.
> 3. We agree that more discussion of Fig. 1 is needed to enhance the understanding of the experiment section. If accepted, an extra page will allow us to improve it, as well as to  include additional content from the attached PDF and this rebuttal in the revision.
>
> ## Questions:
> - Answers to both __Q1__ and __Q2__ are positive. Please see our general reply for the discussion.
> - __Q3:__ TO methods apply only to equally spaced data and the sampling frequency $1/\Delta t$ must be high enough to distinguish all relevant time-scales. Otherwise, since TO eigenvalues are $e^{\lambda_i \Delta t}$, small spectral gaps complicate learning (see Thm. 3 [22]). Conversely, our IG method, which uses gradient information, is time-scale independent, handles irregularly spaced measurements, and does not rely on time discretizations (see reply to Q6). While it results in better generalization, as shown in Fig. 1 c)-d)-e) where our IG estimator captures ground truth significantly better than TO for the same sample size, this generalization across all time scales incurs quadratic computational complexity w.r.t. state dimension and not statistical accuracy (see also reply to Q7). Lastly, with our additional discussion on imperfect knowledge, our IG method can be safely applied in a fully data-driven regime.
> - __Q4:__ For i.i.d. data from $\pi$, at this point, it is hard to answer this question, while for trajectory data, important improvement can be expected, see reply to Q6. While the standard assumptions (SD) and (KE) are the same for TO and IG error bounds, regularity assumptions in these two settings are not easily comparable. This motivates the development of non-standard regularity assumptions for TO and IG learning that could expose the benefit of the prior knowledge. This is an interesting problem in its own right, and it will be the subject of our future work.
> - __Q5:__  As illustrated in Fig. 3 in the attached pdf, estimators are quite robust w.r.t. value of the shift parameter $\mu$. Theoretically, it is important that $\mu$ is not too small.
> - __Q6:__ Thank you for raising this question, we will emphasize these important aspects of our method in the revision.
>   - Recalling the risk functional in eq. (8), we see that the “label” of the model $\chi_{\mu}(x) \approx G^*\phi(x)$ is the action of the resolvent. Since this “label” is not computable, we “__fight fire__ (resolvent) __with fire__ (generator)”, i.e. we use the energy norm of eq. (9) to rewrite regularized problem (8) as in line 222. Crucially, this allows us to obtain estimators via energy covariance $W_{\mu}$ in (14) that __completely captures infinitesimal nature__ of the learning problem without needing time-lagged observations. This contrasts with TO methods, where choosing the time-lag $\Delta t$ is the major bottleneck in real applications, [18,42].
>   - Thank you for this reference (Meng et al. 2024)  that we will cite in the revision. Note that this method is exactly the Galerkin projection of the Yoshida approximation of IG on a finite-dimensional RKHS (dictionary of functions). It essentially corresponds to solving (8) where $\phi$ is finite-dimensional, expectation is used instead of energy and $\chi_{\mu}$ is replaced with $\mu^2\chi_{\mu}(x)-\mu\phi(x)$. To do this, authors estimate resolvent via equation in line 167 which requires approximating an operator integral via __equally spaced data with high sampling frequency (small $\Delta t$)__, essentially suffering from the same bottleneck as TO methods. Since in (Meng et al. 2024) the provided learning theory and the empirical evaluation is limited, guarantees and limitations of this method are not fully clear to us.
>   - In practical applications to obtain data from an invariant distribution $\pi$ one uses the trajectory data after some burn-in time needed to ensure that ergodic mean approximates well $\pi$. Then, the problem is reduced to studying only the dependence as is done e.g. in [21] using standard tools of $\beta$-mixing and the method of blocks. This allows one to obtain non-parametric learning bounds for TO methods, c.f.  [22], where the effective sample size suffers from the multiplicative effect of the time-lag to achieve approximate independence. Recalling Q3, TO methods “waste” a lot of data negatively impacting statistical accuracy (we expect similar limitations for the method of  (Meng et al. 2024)). Contrary to this, our method can be applied to data with larger time-lags (even irregularly spaced) so that effective sample size is close to the true one.
> - __Q7:__ Compared to FEM, the key difference lies in the approximation error $ |\lambda_k - \widehat{\lambda}_k| \leq c h^{2p} |\lambda_k|$, where $p$ is the polynomial degree used to construct finite elements, $h$ is the mesh size, and $c$ depends on the eigenfunctions' smoothness. As the number of mesh elements grows exponentially with $d$,  i.e., $\sim h^{-d}$, reducing $h$ is the major bottleneck mitigated by computationally demanding adaptive higher order methods. On the other hand, our IG method that requires less or no knowledge has a quadratic impact of the $d$ only on the computational complexity. Indeed, sample complexity depends on the effective dimension of the equilibrium distribution on the domain that can be much lower than $d$.

---

> ### Comment · Reviewer_D2P3 · 2024-08-11
> **Rebuttal Reply for Submission18874 by Reviewer D2P3**
>
> I thank the authors for the extensive rebuttal in the general reply and comprehensively addressing my review comments. Also, it is great to see the authors put in the effort to deliver adjusted theory as well as run new experiments.
>
> Many important aspects of the paper got clearer: limits of FEM and TO approaches, fully-data driven regime, energy functional rationale and shift parameter.
> Accordingly, I have increased my overall score.
>
> I expect this rebuttal content to be incorporated as part of a final submission.

---

> > ### Author Response · Authors · 2024-08-12
> > **Acknowledgement to the reviewer**
> >
> > We would like once more  to sincerely thank the reviewer for their deep questions, which inspired us to make these additional steps and improve our work.
> >
> > We are happy that our rebuttal is appreciated and we commit to incorporate it in the revised manuscript.

---

### Official Review · Reviewer_PUzn · 2024-07-13

**Soundness:** 3
**Presentation:** 3
**Contribution:** 3
**Rating:** 7
**Confidence:** 3

**Summary:**

The paper considers a time-homogeneous Stochastic Differential Equation (SDE) with known diffusion part and known or unknown drift. The problem is to find properties of this equation from known data, in particular, to find a (low-rank) representation of its infinitesimal generator (IG). For this purpose, the resolvent of this operator is considered and the reduced-rank estimator in reproducing kernel Hilbert spaces (RKHS) together with the energy-based risk metric are used. Accuracy estimates for the found approximation to IG and time-complexity are given. Several model numerical experiments are conducted for proof-of-concept.

**Strengths:**

- Detailed appendix with necessary reference material

- Development of SDE theory with exact estimates on the result obtained

**Weaknesses:**

- The paper uses quite complex concepts and may be difficult to understand in a 9 page format.

- There are not enough practical examples of the application of these results to machine learning


Minor

The x-axis in Figure 1 (a) is not signed

L120 comma is not needed

L124 space after dot is missing

L206: colon missing at the end of the line

L215 A new line  is redundant (`this work we focus on the case when`)

L275 space after dot is missing

L915 "anan" -> "and"

**Questions:**

- Can your method be extended to the case when neither drift nor diffusion coefficients are known?

**Limitations:**

Limitation are described in the paper. The paper is mostly theoretical.

---

> ### Author Rebuttal · Authors · 2024-08-06
>
> We appreciate the reviewer's insightful evaluation and valuable comments. In what follows, we aim to address the highlighted weaknesses and respond to the reviewer's questions.
>
> ## Weaknesses:
>
> ### Major:
>
> - Indeed, the topic of learning IG of a stochastic process with kernel-based methods, and, in particular, development of nonparametric learning rates contains several complex concepts. More so, since the paper is the first one to formalize a consistent way to learn IG from data. We tried our best to present the sophisticated theory and mitigate the complexities, believing that this work paves the path to efficient and reliable algorithms for important applications in ML (see also general reply).
> - In addition to our general reply, we would like to emphasise here that, from molecular dynamics to weather models, the ability to reliably learn eigenfunctions of differential operators that generate the dynamics is of capital importance for building trustworthy and physics informed AI. Hence, we strongly believe this work is of __high relevance__, if not for the most general ML community, definitely __for the AI-for-Science community__.  With that respect, we would like to further stress that our method is the first of its kind in machine learning to have error bounds that theoretically demonstrate the superiority of an ML approach for spectral estimation of differential operators over classical numerical methods like FEM in high-dimensional settings. While empirical evidence supporting this has been well documented in practice, particularly in fields such as molecular dynamics [42], to the best of our knowledge, this had never been theoretically proven until now. Following reviewer's suggestions, in the revised manuscript we will better emphasize possible applications and further improve readability and accessibility of our results.
>
>
>
> ### Minor:
>
> - Thank you for spotting the typos, we will correct them in the revision.
>
>
> ## Questions:
>
> As we elaborate in the general reply, indeed we can show, both theoretically and empirically (Figures 1 and 2 in the attached pdf), that our method works in a fully data-driven regime. While empirically this might not come as a big surprise, we believe that __adding a theoretical proof__ motivated by the comments of reviewers is a very nice __additional contribution__. If the reviewer wishes to have more details on this aspect, we are happy to elaborate.

---

> > ### Comment · Reviewer_PUzn · 2024-08-12
> >
> > I thank the authors for their explanations. After reading the global response and the discussion with other reviewers, I believe that in general my concerns were addressed. I raise my score to 7 "Accept".

---

> > > ### Author Response · Authors · 2024-08-12
> > > **Acknowledgement to the reviewer**
> > >
> > > We would like once more to thank the reviewer for their comments, which inspired us to make additional steps and improve our work.We are happy that our rebuttal was helpful and we commit to incorporate it in the revised manuscript.

---

### Official Review · Reviewer_6wyC · 2024-07-15

**Soundness:** 3
**Presentation:** 3
**Contribution:** 2
**Rating:** 6
**Confidence:** 2

**Summary:**

In this paper, the authors consider the problem of learning the infinitesimal generator a Stochastic Diffusion Process (SDP). Compared to existing approaches such as [1] they tackle the unbounded nature of the generator by introducing a novel statistical framework which is based on the Dirichlet form associated with the SDP. In this framework, the authors estimate the resolvent of the generator which can be approximated with finite-rank operators. They consider a regularized and rank-truncated version of the regression loss. Theorem 1 provides a way to compute the eigenvalues and eigenvectors of the estimated operator. Spectral learning bounds are derived in Theorem 2. The theoretical results of the paper are completed with three examples: learning the overdamped Langevin generator for a one-dimensional four well potential, the overdamped Langevin generator for a Muller Brown potential and finally for a Cox-Ingersoll-Ross process (CIR process).

[1] Cabannes et al. (2023) -- The Galerkin method beats Graph-Based Approaches for Spectral Algorithms

**Strengths:**

* Even though the paper is mathematically heavy and contains a lot of notation, I think it is well-written. I appreciate that (almost) all the assumptions needed before stating Theorem 2 are clearly laid out and explained. I also appreciate the rigour shown in the paper.

* The obtained results regarding the spectral bounds are interesting and provide a strong theoretical grounding for the method.

* From a methodological point of view the time complexity can be reduced compared to [1,2,3] if the rank is low. I think this is one of the strength of the method, although I would have appreciated more details on the choice of the hyperparameters and their interactions (see below).

[1] Cabannes et al. (2023) -- The Galerkin method beats Graph-Based Approaches for Spectral Algorithms

[2] Hou et al. (2023) -- Sparse learning of dynamical systems in RKHS: An operator-theoretic approach

[3] Pillaud-Vivien et al. (2023) -- Kernelized Diffusion Maps

**Weaknesses:**

Before expressing my main concerns regarding this paper, I want to emphasize that I'm not an expert in this domain and therefore my understanding of the main competitors and methods used in the paper is lacking.

* My main concern is regarding the experiments. I know this is a theoretical paper but a new statistical framework and a new learning procedure should be clearly validated against competing methods like [1,2,3]. Even though some qualitative conclusions are drawn I would have liked to see a more extensive study (for different choices of hyperparameters for both the target Langevin diffusion and the method introduced by the authors).

* Indeed, there are little details in the paper about how to choose the crucial parameters of the algorithm like $\mu$, $r$ and $\gamma$. Establishing the robustness of the procedure regarding these hyperparameters seem crucial to validate the methodology.

* As mentioned before, even though I appreciate the rigour of the paper it is notation heavy. Having to constantly rely on the (massive) table of page 13 to remember the notation hindered my understanding of the paper. Is there a way to either reduce the notational load or to incorporate a lightweight version of Table 2 in the main paper? This would greatly help the reader.

* All assumptions are commented except (KE). What are the main limitations of this assumption? I am not familiar with the domain so a light explanation in the main text would also be appreciated.

* Is the assumption of l.188 that there exists a Dirichlet operator realistic? It is true for Langevin operator and CIR (which already covers quite a lot of ground) but apart from these models? Does this impose anything on the diffusion?

Some minor remarks:

* l.178 "Importantly, this energy functional can be empirically estimated from data sampled from π, whenever full knowledge, that is drift and diffusion coefficients of the SDE (1), or partial knowledge," -- This sentence is not really clear to me. How can one leverage the drift and diffusion coefficient here?

* l.215 typo (spurious new line)

[1] Cabannes et al. (2023) -- The Galerkin method beats Graph-Based Approaches for Spectral Algorithms

[2] Hou et al. (2023) -- Sparse learning of dynamical systems in RKHS: An operator-theoretic approach

[3] Pillaud-Vivien et al. (2023) -- Kernelized Diffusion Maps

**Questions:**

See weaknesses

**Limitations:**

Limitations are addressed in Section 7 ("Conclusion")

---

> ### Author Rebuttal · Authors · 2024-08-06
>
> We appreciate the reviewer's insightful evaluation and valuable comments. In what follows, we aim to address the highlighted weaknesses and respond to the reviewer's questions.
>
> ## Weaknesses:
>
> __Main contributions and broader impact.__ In the general reply we tried our best to provide more details on positioning of our main contributions in the larger context of solving SDEs, and help the reviewer evaluate the possible impact of the presented results.
>
> __Experiments.__ While we would like to stress out that we originally provided quantitative comparison to the cited methods ([1] is in fact just a special case of [2], while [3] is not applicable to Langevin dynamics) in Fig. 1 a) which implies the failure of this method to capture the true dynamics,  we follow the reviewer’s suggestion and provide additional experiments that make this point more clear for the reader.
>
> __Choice of hyper-parameters.__ As suggested, we provide the  empirical evaluation of the robustness of hyper-parameter choice, as indicated in the general answer. In summary, the robustness of the choice of kernel’s hyper-parameters and Tikhonov regularization is similar to general kernel methods for (vector-valued) regression. On the other hand when estimating IG via its resolvent the choice of additional shift hyper-parameter $\mu$, as expected from the theory, has a much smaller impact on the performance, see Fig. 3 of the attached pdf.
> Finally, we wish to stress out that we do go beyond classical validation via prediction by introducing novel quantities, the empirical spectral biases $\widehat s_i$ of eq. (24), which are (theoretically and empirically) good metrics to use for fine-tuning the model.
>
> __Notations.__ Thank you for your suggestion. We will prepare a smaller summary table that only introduces necessary notations for presenting the IG learning framework and the method, while omitting extensive notation related to proving the generalization bounds. We believe this is feasible, since, if the manuscript is accepted, the additional page can be used to present this table in the main body of the paper.
>
> __(KE) assumption.__ The kernel embedding (KE) assumption originates from the study of mini-max optimal excess risk bounds for classical kernel regression [12]. Whenever the kernel is bounded, (SD) and (KE) do _not impose any additional constraints_, they are just used to _describe the interplay between the data distribution and the chosen RKHS_. This is a formal way to quantify the impact of the kernel choice to the learning rate. We will additionally clarify this in the revised manuscript.
>
> __Dirichlet form.__ To complement the discussion on this issue from the general reply, we would like to stress out that many diffusion processes have an infinitesimal generator that can be expressed in the form (IG). In addition to the Langevin and the CIR processes already studied, we can mention:
> -  the Wright-Fisher diffusion (in dimension one), which can be defined in the context of population genetics and can be adapted to model interest rates, see [G],
> -  the geometric Brownian motion which models the price process of a financial asset,
> - the multi-dimensional Brownian motion (a=0) that corresponds to the heat equation,
> - the transport processes associated with advection-diffusion equation, see [H]
> - the process associated with Poisson's equation in electrostatics, see [I].
>
> __Minor remarks.__ Thank you for pointing out the typo, we will fix it. Concerning the claim in line 178, it relates to the first equality in Equation (7). There, one can see that knowing drift and diffusion terms, we can evaluate $[L f] (x) $ and approximate the energy as empirical mean of $ \mu |f (x) |^2 + f(x)[L f] (x) $ via samples $x$ from the invariant distribution.
>
>
> ## Additional references:
> [A] Fukushima, Masatoshi, Yoichi Oshima, and Masayoshi Takeda. Dirichlet forms and symmetric Markov processes. Vol. 19. Walter de Gruyter, 2011
>
> [B] Jean Jacod. Discretization of processes. Springer, 2012
>
> [C] Danielle Florens-Zmirou. Approximate discrete-time schemes for statistics of diffusion processes. Statistics: A Journal of Theoretical and Applied Statistics 20.4, 1989
>
> [D] Fan, J., & Zhang, C. (2003). A reexamination of diffusion estimators with applications to financial model validation. Journal of the American Statistical Association, 98(461), 118-134.
>
> [E] Yuri Kutoyants. Statistical inference for ergodic diffusion processes. Springer Science & Business Media, 2013.
> Yuri Kutoyants. Parameter estimation for stochastic processes, 1984.
>
> [F] Fabienne Comte,Valentine Genon-Catalot Nonparametric drift estimation for iid paths of stochastic differential equations. The Annals of Statistics, 48(6), 2020
>
> [G] Martin Grothaus, Max Sauerbrey. Dirichlet form analysis of the Jacobi process. Stochastic Processes and their Applications, 157, 2023.
>
> [H] Antoine Lejay,  Lionel Lenôtre, Géraldine Pichot. Analytic expressions of the solutions of advection-diffusion problems in one dimension with discontinuous coefficients. SIAM Journal on Applied Mathematics, 79(5), 2019.
>
> [I] Sethu Hareesh Kolluru. Preliminary Investigations of a Stochastic Method to solve Electrostatic and Electrodynamic Problems. Phd Thesis, 2008.

---

> > ### Comment · Reviewer_6wyC · 2024-08-12
> >
> > Thank you for your thorough  answer and rebuttal. I have increased my score.

---

> > > ### Author Response · Authors · 2024-08-12
> > > **Acknowledgement to the reviewer**
> > >
> > > We would like once more to thank the reviewer for their comments, which inspired us to make additional steps and improve our work.We are happy that our rebuttal was helpful and we commit to incorporate it in the revised manuscript.

---

### Official Review · Reviewer_nFzp · 2024-07-26

**Soundness:** 3
**Presentation:** 4
**Contribution:** 3
**Rating:** 5
**Confidence:** 3

**Summary:**

This paper discusses learning the generator of stochastic diffusion processes in reproducing kernel Hilbert space. In particular,
section 2: background on the generator, Dirichlet form, energy, learning in RKHS, empirical risk in the Hilbert-Schmidt norm
section 3: introduce an energy-based risk functional for resolvent estimation which leads to spectral estimation
section 4: empirical risk minimization with Tikhonov regularization and rank constraints.

**Strengths:**

The paper is well-written and has a thorough discussion on the background and the comparison of previous work. It proposes a novel energy risk functional and a reduced-rank estimator with dimension-free learning bounds. It also establishes spectral learning bounds for generator learning.

**Weaknesses:**

The weaknesses are mainly in the motivation (please see question 1 below) and the experiment (see questions 3 and 4 below). The authors did a great job in the theoretical comparison with previous work on TO and IG learning, however, the empirical comparison focuses on the spuriousness of eigenvalues and the recovery capability of the metastable states. To sufficiently demonstrate the practical advantage of the proposed method, other comparisons of the learned dynamics, phase transition, and invariant measure should also be included.

**Questions:**

1. In the paper, 'full knowledge' refers to given drift and diffusion coefficients, and 'partial knowledge' refers to the given diffusion coefficient and energy/Dirichlet operator. In practice, is it necessarily easier to obtain diffusion coefficient and energy/Dirichlet operator than to obtain drift and diffusion coefficients?
In a fully data-driven scenario, one can first estimate drift and diffusion coefficients, and then use physics-informed method with 'full knowledge' obtained. Does this make the 'full knowledge' methods cited in the paper more flexible than the 'partial knowledge' method proposed?

2. As transfer operator A_t and the infinitesimal generator L satisfy the relation A_t = e^{Lt}, is there a consistency between the learned TO from cited TO methods with the learned IG from the proposed method?

3. What are the kernels used for the 3 examples in experiments? How sensitive is the experimental result to the kernel selection, regularization, and rank parameter?

4. Can the authors provide the learned dynamics of the 3 examples compared to the true dynamics, the invariant measure, and time scales?

**Limitations:**

Limitations have been discussed.

---

> ### Author Rebuttal · Authors · 2024-08-06
>
> We appreciate the reviewer's insightful evaluation and valuable comments. In what follows, we aim to address the highlighted weaknesses and respond to the reviewer's questions.
>
> ## Weaknesses:
>
> Thank you for pointing this out. Indeed, clearer motivation beyond the Dirichlet form should help the reader understand our approach better. While some aspects of this are addressed in the general reply, in the following we try to address your concerns in detail.
>
> ## Questions:
>
>  1. As we have discussed in the general reply, a large class of processes can be written in Dirichlet form and, importantly, for many of them statistical estimation of the Dirichlet coefficient (related to the diffusion coefficient) from data is much easier than recovering the drift term. This means that our method is possible to apply even when the challenging drift term is not estimated. That said, we are, to our knowledge, __the only existing method__ that is able to learn the infinitesimal generator model from __a single trajectory__ that is __theoretically consistent with the true dynamics__. Finally, as we show in the general reply, note that our method is robust under imperfectly estimated partial knowledge. This is not only proven, but also empirically demonstrated in Fig. 1 and 2 of the attached pdf.
>
> 2. Indeed, there is the consistency of our IG method and the properly executed TO methods. This is theoretically backed by Thm. 2 in [22] for TO and our Thm. 2 for IG. However, note that TO methods apply only to equally spaced data and the sampling frequency  $1/\Delta t$ must be high enough to distinguish all relevant time-scales. Otherwise, since TO eigenvalues are $e^{\lambda_i \Delta t}$, small spectral gaps complicate learning (see Thm. 3 [22]). Conversely, our IG method, which uses gradient information, is time-scale independent, handles irregularly spaced measurements, and does not rely on time discretizations. This results in better generalization, as shown in Fig. 1 c)-d)-e) where our IG estimator captures true phase transitions between meta-stable states significantly better than TO for the same sample size. Lastly, with our additional discussion on imperfect knowledge, our IG method can be safely applied in a fully data-driven regime as TO methods.
>
> 3. Thank you for spotting this. Indeed, in App. F we didn’t specify that the RBF Gaussian kernel with specified length-scales was used in all cases. As reported in the generall reply, the robustness of kernel hyperparameters and Tikhonov regularization is essentially the same as for TO kernel estimators, while the IG estimator is very robust against the new shift hyperparameter, see Fig. 3 of the attached pdf.
>
> 4. Thank you for this question.  Besides original comparisons of the time-scales (IG eigenvalues) and meta-stable states (IG eigenfunctions), providing the additional experiments, we demonstrate our method’ consistency with true dynamics from the perspective of equilibrium distribution recovery and  long term  forecasting of conditional cumulative density functions in Fig. 1 and 2 of the attached pdf, respectively. This might come as a surprise, but after a closer look, the cited works,  only showcase empirical evaluation of the leading eigenfunctions, which due to the reported effect of spuriousness in Fig, 1a requires picking proper eigenpairs from an abundance of wrongly estimated ones. Indeed, if one builds such models without expert knowledge needed to remove spurious estimation, the obtained dynamics is inconsistent with the true one, see caption of Fig. 1 of the attached pdf.

---

### Author Rebuttal · Authors · 2024-08-06

We wish to thank all reviewers for their insightful evaluation of our paper. We appreciate all their comments and remarks, which we will incorporate in our revision. Before addressing each review in detail, we would like to point out some general remarks that apply to all of them.

## __Assumptions on the process and availability of partial knowledge.__
The existence of Dirichlet forms for diffusion processes satisfying a stochastic differential equation like (1) is not too restrictive an assumption, as it encompasses many processes beyond Langevin and CIR such as advection-diffusion, Wright-Fisher diffusion, multidimensional Brownian motion, etc. (see reply to reviewer __6wyC__ for details and the list of additional references, there ref. [A] gives in-depth review of this topic). For the reviewers convenience, we briefly elaborate. Recalling the form of IG in eq. (3), the Dirichlet form in eq. (5) for a self-adjoint $L$ exists whenever the positive definite diffusion part $b b^\top$ satisfies uniform ellipticity conditions and the drift term $a$ allows integration by parts, leading to $s(x)=b(x)/\sqrt{2}$.  Thus, to obtain the partial knowledge we need, it is enough to estimate the diffusion function. When the sample paths are observed continuously, the diffusion coefficient $b$ can be directly identified from these observations, making  it a non-statistical problem. For discrete realizations of the process, __the diffusion coefficient can be estimated non-parametrically using various methods__. These methods include pathwise estimation by computing the variance of the increments over small intervals [B], kernel-based methods [C], local polynomial regression [D]. On the other hand, __the estimation of the drift is a much more demanding task__. Different methods are reviewed in [E] and references therein. A more recent approach [F] drawing inspiration from particle systems, consists in constructing estimates from $m$ i.i.d. paths of the solution process.

 ## __Imperfect partial knowledge and fully data-driven method.__
Inspired by the reviewers' questions, we show how to make our method fully data-driven. This theoretical discussion will be presented as a remark in the main body of the revision with detailed proofs in the appendix. In practice, whenever classical modeling from first principles is not feasible, we can first estimate the Dirichlet form and then apply our method. Hence, for a fully data-driven method, we need to analyze the impact of the imperfect knowledge. To this end, assume that we apply our model with some $\tilde s$ that incurs relative estimation error $\tilde \varepsilon>0$ of the form
$$
\mathbf{(RE)} \quad \tilde \varepsilon s(x)^\top s(x) \preceq  \tilde s(x)^\top \tilde s(x) - s(x)^\top s(x) \preceq  \tilde \varepsilon s(x)^\top s(x),
$$
where $\preceq$ is the standard Loewner ordering. Note that whenever $s$ is constant (e.g. Langevin) or linear (e.g. CIR), the estimation can be done from a single trajectory by estimating the variance of increments, and then __(RE)__ reduces to the standard relative error. So, our algorithm now uses empirical covariance $\widetilde W$ built from $\tilde s$, and, hence, the only change to our proof technique lies in the analysis of variance in App. E.3. In particular, we only need to adapt Prop. 15 and 16, which is straightforward since (RE) implies $\tilde\varepsilon \widehat W \preceq \widetilde W-\widehat W \preceq  \tilde\varepsilon \widehat W$. Indeed, we have
$$
\Vert F(W-\widetilde W)FG \Vert = \Vert F(W\pm\widehat W-\widetilde W)F G \Vert \leq \Vert F(W-\widehat W)FG \Vert  + \tilde\varepsilon \Vert F(\widehat W \pm W)F\Vert \Vert G \Vert \leq \Vert F(W-\widehat W)FG \Vert + \tilde\varepsilon (\Vert F(W-\widehat W)F\Vert +1)\Vert G \Vert,
$$
for $F= W_{\mu,\gamma}^{-1/2}$ and $G$ being either $W_{\mu,\gamma}^{-1/2}C$ or $I$. So, in conclusion, __the relative error $\widetilde\varepsilon$ of imperfect knowledge simply appears as the additive term in the final guarantees.__ Namely, (23) and (24) hold upon replacing $\varepsilon_n^\star \ln \delta^{-1}$ with $\varepsilon_n^\star \ln \delta^{-1} + \tilde \varepsilon$.

## __Experiments.__
Given the theoretically challenging nature of the problems addressed in this paper, we have intentionally limited the role of experiments to effectively highlight the main theoretical contributions. While we include two additional experiments in the attached pdf and test robustness on hyper-parameters choice, significantly extending empirical studies provided in the most related works on IG  [1, 15, 36], our focus remains consistent. We believe that developing realistic applications requires extensive work and contextual analysis, which  would constitute contribution in its own right.

## __Contributions.__
To conclude, let us clarify our contributions:
 1. We propose a __fundamentally new idea to estimate the spectrum of self-adjoint generators of stable Ito SDE from a single trajectory__. In contrast to all existing works, we exploit the geometry of the process via a novel energy (risk) functional. In a certain sense we “fight fire (resolvent ) with fire (generator) ” to
2.  derive __a new efficient learning method__ that is able to learn the best approximation of the resolvent of IG on the RKHS __independently of the time-sampling__, (see reply Q6 to __D2P3__);
3. We derive __the first IG spectral estimation finite sample bounds__ using (imperfect) partial knowledge, which __notably, overcome the curse of dimensionality present in classical numerical methods__ (see reply to __D2P3__, item Q7 for a quantitative discussion);
4. __each important aspect of our learning method__, especially in relation to the most relevant existing works, __is empirically demonstrated__ to complement our theoretical analysis.

We once again thank all the reviewers whose comments inspired the discussion above that significantly strengthens our paper and better demonstrates its broader impact.

---

### Decision · Program_Chairs · 2024-09-25

**Decision:**

Accept (poster)

**Comment:**

The reviewers agree that this is an interesting paper that addresses a different and important problem from a typical NeurIPS submissions. Counterbalancing its strengths as a theoretically grounded estimation framework, the reviewers list some concerns about motivation, hyperparameter tuning, experimental evaluation and presentation. I think the authors have addressed the first two well enough (including additional results). I think there is some truth to the latter two points, and while I personally though the  presentation in the main body was fine, the over submission crams a lot of theory material into the supplementary material, and I do not think the reviewers have or can be expected to go carefully over it. As such, I think this paper sits just at the borderline.